# GNN Explanations that do not Explain and How to find Them

**Steve Azzolin** [†]   **Stefano Teso**[†]   **Bruno Lepri**[‡]   **Andrea Passerini**[†]   **Sagar Malhotra**[§]

## Abstract

Explanations provided by Self-explainable Graph Neural Networks (SE-GNNs) are fundamental for understanding the model's inner workings and for identifying potential misuse of sensitive attributes. Although recent works have highlighted that these explanations can be suboptimal and potentially misleading, a characterization of their failure cases is unavailable. In this work, we identify a critical failure of SE-GNN explanations: explanations can be unambiguously unrelated to how the SE-GNNs infer labels. We show that, on the one hand, many SE-GNNs can achieve optimal true risk while producing these degenerate explanations, and on the other, most faithfulness metrics can fail to identify these failure modes. Our empirical analysis reveals that degenerate explanations can be maliciously planted (allowing an attacker to hide the use of sensitive attributes) and can also emerge naturally, highlighting the need for reliable auditing. To address this, we introduce a novel faithfulness metric that reliably marks degenerate explanations as unfaithful, in both malicious and natural settings.

## 1 Introduction

Self-Explainable GNNs (SE-GNNs) combine the predictive power of standard GNNs with ante-hoc explainability (Miao et al., 2022a; Chen et al., 2024). They couple an explanation extractor, which identifies explanatory subgraphs, with a classifier that uses these subgraphs to generate predictions. Being designed with explainability in mind, SE-GNNs promise to be genuinely interpretable and therefore suitable for high-risk use cases, e.g., power grid analysis (Varbella et al., 2024), health forecast (Hu et al., 2024), and drug discovery (Wong et al., 2024).

Despite their promise, previous work has highlighted weaknesses in SE-GNNs' explanations. For instance, they can be redundant (Tai et al., 2025), ambiguous (Azzolin et al., 2025b), may not fully reflect the individual importance of each of their sub-components (Chen et al., 2024), and can be affected by spurious correlations (Wu et al., 2022). Nonetheless, little research has focused on whether there exist cases in which SE-GNNs catastrophically fail to provide meaningful explanations.

In this work, we identify a critical failure case of SE-GNN's explanations: *explanations can be unrelated to the SE-GNN's inner workings* (see Fig. 1 for an example). Such explanations are unambiguously unfaithful, undermining the original purpose of SE-GNNs, raising the risk of explanations concealing the use of protected attributes, and impairing model debugging and scientific discovery. First, we theoretically show that, under mild assumptions, several SE-GNNs can output these unfaithful explanations while achieving optimal loss. We then exploit this result to answer the following questions positively:

- **RQ1:** Can SE-GNN be manipulated to output maliciously defined unfaithful explanations?
- **RQ2:** Can these unfaithful explanations go undetected by faithfulness metrics?
- **RQ3:** Can these unfaithful explanations also emerge without manipulation?

To address the undetectability of these unfaithful explanations, we present a *benchmark* for faithfulness metrics and a novel *evaluation metric*, EST, that is shown to be more robust. More generally, our work warns practitioners from blindly trusting SE-GNN explanations, highlights avenues to make these models more reliable, and provides users with more reliable tools to audit their explanations.

---

Corresponding author: `steve.azzolin@unitn.it`
[†] University of Trento, Italy   [‡] Fondazione Bruno Kessler, Italy   [§] TU Wien, Austria

## 2 BACKGROUND

**Graph Classification**. We consider the standard supervised graph classification setting, where the goal is to learn a deterministic graph classifier $f$ mapping from the set of graphs $\mathcal{G}$ to the set of labels $\mathcal{Y}$, typically implemented as a GNN (Bacciu et al., 2020). A graph $G = (V, E)$ is defined as a set $V$ of nodes connected by edges $E$, and both can have features associated with them. We write $R \subseteq G$ to denote that $R$ is a subgraph of $G$, and $|G|$ to denote the size of the graph in terms of nodes, edges, or both, as will be clear by the context. Subgraphs may preserve all, some, or none of the features.

**Self-explainable GNNs.** SE-GNNs aim to enhance the interpretability of graph classifiers by generating explanations during inference (Miao et al., 2022a;b; Chen et al., 2024). An SE-GNN consists of two components: (1) an **explanation extractor** $e$ that maps the input graph $G$ to a subgraph $e(G) = R \subseteq G$, and (2) a **classifier** $g$ that takes $R$ as input and produces the final prediction:

$$f(G) = g(e(G)). \tag{1}$$

The explanation extractor $e$ can either produce per-edge relevance scores $p_{uv} = e(G)_{(u,v)} \in [0,1]$, or per-node relevance scores $p_u = e(G)_u \in [0,1]$, which are thresholded or top-$k$ selected to form the explanatory subgraph $R$ (Wu et al., 2022; Tai et al., 2025). We will focus on per-node relevance scores, but our analysis also applies to per-edge scores. Also, we say the explanation extractor is *hard* if it outputs scores in $\{r, 1\}$, where $r$ is either 0 or an SE-GNN-specific hyperparameter; the precise value will be clear from the context. We will assume $|R| > 0$ to prove our theoretical result.

**Faithfulness metrics.** At a high level, faithfulness metrics assess whether the predictor's output changes upon perturbing the input (Azzolin et al., 2025a). Metrics that estimate an explanation's *sufficiency* perturb the complement of the explanation subgraph while keeping the rest unchanged, while metrics that evaluate its *necessity* do the opposite. An explanation is considered highly faithful if perturbing the complement does not induce a large shift, and if perturbing the explanation does. Different metrics also differ based on the set of perturbations $\mathcal{I}$ they allow: *i & ii)* **Complement removal** and **Explanation removal** metrics erase the complement and the explanation altogether, respectively; *iii)* **Edge removal** metrics erase edges at random; *iv)* **Complement swap** metrics replace the complement of an explanation with that of another graph. With a slight abuse of notation, we use $G' \in \mathcal{I}$ if $G'$ is obtainable from $G$ by applying the perturbations given by $\mathcal{I}$. We summarize existing metrics in Table 1.

Table 1: **Taxonomy of popular faithfulness metrics**, grouped by whether they estimate *necessity* vs. *sufficiency* of explanations, and by the family of perturbations $\mathcal{I}$ they employ.

| Type | Perturbations $\mathcal{I}$ | Metrics |
|---|---|---|
| *Nec.* | Explanation removal | Fid+ (Yuan et al., 2022) PN (Tan et al., 2022) |
| *Nec.* | Edge removal | RFid+ (Zheng et al., 2023) Nec (Azzolin et al., 2025a) |
| *Suff.* | Complement removal | Fid- (Yuan et al., 2022) PS (Tan et al., 2022) GEF (Agarwal et al., 2023) |
| *Suff.* | Edge removal | RFid- (Zheng et al., 2023) SimOAR (Fang et al., 2023) |
| *Suff.* | Complement swap | Suf (Azzolin et al., 2025a) |
| *Suff.* | All | EST (**Ours**, Definition 1) |

## 3 SE-GNNS'S EXPLANATIONS THAT DO NOT EXPLAIN

In this section, we study SE-GNNs through the lens of their loss functions, and provide a sufficient condition to achieve optimal true risk for several representative SE-GNNs, namely, `GSAT` (Miao et al., 2022a), `LRI` (Miao et al., 2022b), `CAL` (Sui et al., 2022), `GMT-lin` (Chen et al., 2024), and `SMGNN` (Azzolin et al., 2025b). Surprisingly, *this condition is met by degenerate explanations highlighting recurrent patterns with no class-discriminative power*, like background pixels for object recognition or punctuation for text classification.

We formalize a class of such recurrent patterns for graphs as an **anchor set**: a set of single-node subgraphs $\mathcal{Z} = \{z_i\}_{i=1}^m$ where $m = |\mathcal{Y}|$ and $z_i \subseteq G$, for all instances $G \in \mathcal{G}$ and for all $z_i \in \mathcal{Z}$. Anchor sets can be seen as a collection of prototypes, one for each class, appearing in every graph. As an example, green (◯) and violet (◯) nodes form an anchor set $\mathcal{Z}$ in Fig. 1. Nodes in an anchor

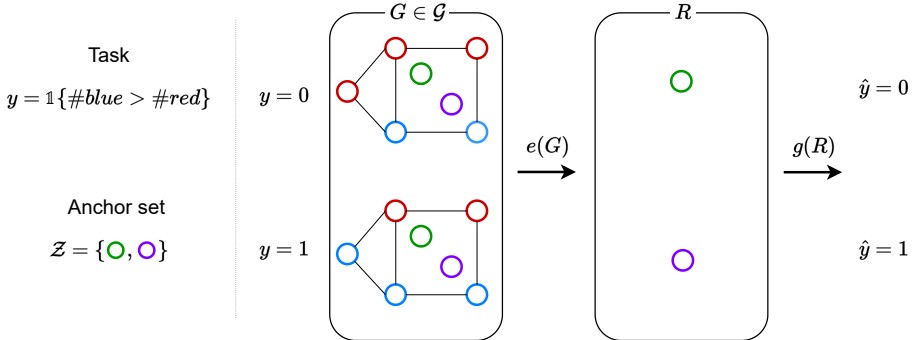

Figure 1: SE-GNNs couple an explanation extractor $e$ producing an explanation $R$, and a classifier $g$ using $R$ to infer the prediction. **We identify a critical failure case of SE-GNNs, i.e., degenerate explanations – explanations that encode the label but are unrelated to how the model actually infers it.** In this example, green (◯) and violet (◯) nodes appear identically in every graph and form an anchor set $\mathcal{Z}$, hence they have no class-discriminative power by construction (see Section 3 and Example 1 for details). Yet the explanation extractor $e$ can exploit them to secretly encode the predicted label while hiding the use of red and blue nodes. Such explanations are highly unfaithful and mislead users by falsely suggesting that green and violet nodes are relevant for the label.

set have no class-discriminative power, as they appear in every sample. Nonetheless, we show that picking them as explanations can be an optimal solution for SE-GNNs.

**Theorem 1.** *Let $\mathcal{D}_{\mathcal{G} \times \mathcal{Y}}$ be a data distribution with deterministic ground truth labeling function $\phi : \mathcal{G} \mapsto \mathcal{Y}$, $e$ be a hard explanation extractor, and $\mathcal{Z} = \{z_y\}_{y \in \mathcal{Y}}$ be an anchor set. Then, there exists an explanation extractor $e(G) := z_{\phi(G)}$ and a classifier $g(z_y) := y$, such that the SE-GNN $g \circ e$ implemented by* `GSAT`, `LRI`, `CAL`, `GMT-lin` *or* `SMGNN` *achieves optimal true-risk.*

The proof is provided in Appendix B. Theorem 1 shows that the explanation extractor can use any recurrent set of nodes as label-encoding explanations for the classifier. These explanations are **unambiguously unfaithful**. To see this, consider a highly accurate SE-GNN. If it relied solely on explanations from the anchor set $\mathcal{Z}$, it would be unable to predict the label at all, as these nodes are constant across graphs. In order to attain high accuracy, it *must* be looking at other parts of the input graph, yet these are not revealed by the explanation. Hence, Theorem 1 shows that SE-GNNs with optimal true risk can provide completely unfaithful explanations. This gives rise to rather counterintuitive degenerate explanations, as depicted in the following example.

**Example 1.** *Let us consider the binary graph classification dataset* `RBGV`, *where graphs are composed of red and blue nodes randomly connected (see Fig. 1). Positive graphs are those where the number of blue nodes ($\#blue$) exceeds the number of red nodes ($\#red$). Every instance further contains two single isolated nodes, one green ($u_{green}$) and one violet ($u_{violet}$). Therefore, green and violet nodes form a valid anchor set $\mathcal{Z}$, and are completely irrelevant to the task that only depends on red and blue nodes. However, the following SE-GNN's explanation extractor $e$ and classifier $g$ can achieve perfect task accuracy while providing explanations consisting of green and violet nodes:*

$$R = e(G) = \begin{cases} u_{green} & \text{if } \#red \geq \#blue \\ u_{violet} & \text{if } \#blue > \#red \end{cases} \qquad g(R) = \begin{cases} 0 & \text{if } R = u_{green} \\ 1 & \text{if } R = u_{violet} \\ 0.5 & \text{otherwise} \end{cases} \qquad (2)$$

Notably, Theorem 1 can be extended beyond sets of nodes appearing in all graphs (see Appendix B.2 for a formal analysis): First, if members of mutually-exclusive sets of nodes appear across graphs, the explanation extractor can partition them into class-specific blocks $\{\{z_0, \dots, z_n\}_y\}_{y \in \mathcal{Y}}$, which the classifier then maps to the respective labels. Second, the explanation extractor can encode predictions inside subgraphs if no node-level anchor set (or no class-specific partition) exists. In fact, we discuss in Section 6 an SE-GNN where the optimization avoids single-node degenerate explanations but still highlights task-irrelevant subgraphs.

A consequence of Theorem 1 is that the same explanation can appear for opposite classes for different optimal SE-GNNs, i.e., explanations fail to be consistent (Dasgupta et al., 2022). Specifically, given

an optimal SE-GNN as per Theorem 1 and a permutation $\pi : \mathcal{Y} \mapsto \mathcal{Y}$ of class indices, it is possible to construct another SE-GNN, say SE-GNN$_\pi$, composed of

$$e_\pi(G) := z_{\pi(\phi(G))} \quad \text{and} \quad g_\pi(z_y) := \pi^{-1}(y)$$

that also achieves optimal true risk. This is made intuitive by Example 1, where it is possible to swap $u_{green}$ with $u_{violet}$ across classes in Eq. (2) without altering the final predictions.

As a final remark, although Theorem 1 covers a broad range of SE-GNNs, alternative formulations outside its scope exist, such as DIR (Wu et al., 2022) and SUNNY (Deng & Shen, 2024). These models enforce compact explanations through an explanation-selection mechanism retaining a fixed ratio $K$ of nodes, with $K$ a hyperparameter. We will empirically show in Section 6 that an inappropriate choice of this hyperparameter can, however, still force the label to be encoded in degenerate explanations.

**Takeaways.** Theorem 1 shows that SE-GNN explanation extractor $e$ can output perfect label-encoding subgraphs which, with an appropriate classifier $g$, provide optimal true-risk. Yet such label-encoding subgraphs can be completely degenerate. These explanations are highly unfaithful, as they only need to convey an encoding of the label to the classifier, without giving any insight into how the overall model arrives at the decision. Next, we show that a malicious attacker can exploit this failure case to make SE-GNNs output fabricated explanations concealing the model's inner workings.

## 4   RQ1: CAN SE-GNN EXPLANATIONS BE MANIPULATED?

We introduce an attack to SE-GNNs aimed at training accurate models that, however, provide explanations unambiguously unrelated to how SE-GNNs infer their predictions. This is significant in two distinct ways: First, it shows how an attacker can manipulate explanations without impairing accuracy, potentially misleading users and concealing reliance on protected features. Second, it offers a controlled, reproducible setup for evaluating faithfulness metrics, as we will discuss in Section 5.

**Setup** Our attack comprises two main steps: First, the attacker defines a designated malicious explanation for each class; Second, they encourage an SE-GNN to output the designated explanation while optimizing for downstream performance. In practice, it amounts to training the SE-GNN using a regular classification loss $\mathcal{L}_{clf}$ paired with a binary cross-entropy loss $\mathcal{L}_{expl}$ designed to push the relevance scores of nodes belonging to the designated explanation $p_u^y$ to 1 and the rest to 0, that is:

$$\min_{g,e} \sum_{G,y} \mathcal{L}_{clf}(g, e, G, y) + \mathcal{L}_{expl}(e, G, y), \qquad \mathcal{L}_{expl}(e, G, y) := \frac{1}{|V|} \sum_{u \in V} \text{BCE}\big(e(G)_u, p_u^y\big). \quad (3)$$

To ensure $\mathcal{L}_{expl}$ does not conflict with other regularization terms, it is preferable to deactivate these. We provide further details in Appendix D.1.2.

**Datasets and designated explanations** We empirically evaluate this attack on one synthetic dataset – RBGV, introduced in Example 1 – and three real-world datasets: MNISTsp (Knyazev et al., 2019), MUTAG (Debnath et al., 1991), and SST2P.[1] To underscore the counterintuitive nature of our results, we will focus on designated explanations belonging to an anchor set, which are therefore *unrelated to the task being solved*. Specifically, our designated explanations include only nodes that *i)* are not predictive of the label, and *ii)* can be used by the SE-GNN to encode its prediction. Examples include green/violet nodes for RBGV, background pixels for MNISTsp, and punctuation for SST2P. We report in Appendix D.4.1 the full list of the designated explanations for each dataset in use in our experiments. Under ideal circumstances, a model that always and only relies on these unrelated subgraphs should be weakly predictive of the true label. Surprisingly, however, SE-GNNs can be trained to achieve high accuracy even when using them as explanations. As it turns out, they can do so by, in essence, *encoding the predicted label in the (relevance scores of the) explanation itself*, even when the highlighted subgraph alone is uninformative for the label.

**Evaluation metrics** We assess the success of the attack by comparing the task accuracy of the attacked model with that of an unmanipulated baseline. Also, to evaluate how well the predicted explanations align with the designated ones, we compute the $F_1$ score[2] between predicted and designated relevance

---

[1] SST2P is a variant of SST2 (Yuan et al., 2022) where we ensure a single ',' and a single '.' to be present in every graph. To avoid the emergence of unwanted correlations, we append them as isolated nodes and with fixed embeddings $\vec{0}$ (for ',') and $\vec{1}$ (for '.'). More details available in Appendix D.4.

[2] We prefer $F_1$ over AUCROC (Luo et al., 2020), which ignores non-sparse yet accurate relevance scores.

scores. Specifically, for each node, we binarize relevance scores – as $\mathbb{1}\{p_u > 0.9\}$ if $p_u^y = 1$ and $\mathbb{1}\{p_u > 0.1\}$ if $p_u^y = 0$ – to penalize uncertain relevance scores, and then report the average across classes. The $F_1$ score is computed only on nodes from correctly classified graphs, as designated explanations cannot be learned if the underlying model fails to infer the right label.

**Results** Table 2 lists the results of attacking three representative SE-GNN architectures, namely GSAT (Miao et al., 2022a), DIR (Wu et al., 2022), and SMGNN (Azzolin et al., 2025b), averaged over five random seeds. Crucially, the attack always succeeds: all attacked models output explanations similar to the designated ones ($F_1 \geq 92\%$). The exception is SMGNN on SST2P ($F_1 \approx 59\%$). This is because the SST2P test set includes Out-Of-Distribution (OOD) graphs (Wu et al., 2022), which are inherently more difficult to recognize for models not designed for OOD generalization. In fact, computing the $F_1$ on the in-distribution validation set yields a much higher value ($F_1 = 96.5 \pm 0.8$), supporting this observation. We report several example explanations for each dataset in Appendix E.2.

Yet, attacked models perform comparably to their unmanipulated counterparts. The exception is SST2P, where attacked models slightly underperformed: the difference is however within one standard deviation for two models out of three. Another interesting case is MNISTsp: while the manipulated GSAT performs slightly worse ($-1\%$ Acc), DIR and SMGNN perform surprisingly better ($+50\%$ and $+5\%$ Acc, respectively). This suggests that encoding the predicted label into the explanation can help models improve their own prediction accuracy during training. We will expand on this in Section 6.

For transparency, we report in Appendix E.3 some cases in which the attack failed to match the designated explanation exactly. Nonetheless, even in these cases, we show that it still managed to force the models into concealing the features they are truly relying on.

These results allow us to answer **RQ1** in the affirmative: in the vast majority of cases, *our attack successfully teaches SE-GNNs to output degenerate explanations unrelated to the task at hand, all without degrading (and sometimes improving) predictive performance*.

Table 2: **SE-GNNs can be successfully attacked to output arbitrary explanations while making accurate predictions.** We report the test accuracy of unmanipulated models trained following standard practices, cf. Appendix D.1, (*Natural*), and of models trained according to Eq. (3) (*Attack*). We also report, for the latter, the test $F_1$ score computed for the designated explanation of Table 5.

| Dataset | Model | Natural | Attack | |
|---|---|---|---|---|
| | | Acc | Acc | $F_1$ **score** |
| RBGV | GSAT | $100.0_{\pm 0.0}$ | $99.1_{\pm 1.4}$ | $99.7_{\pm 0.2}$ |
| | DIR | $99.8_{\pm 0.3}$ | $99.8_{\pm 0.4}$ | $99.4_{\pm 0.3}$ |
| | SMGNN | $100.0_{\pm 0.0}$ | $100.0_{\pm 0.0}$ | $99.6_{\pm 0.2}$ |
| MNISTsp | GSAT | $94.7_{\pm 0.1}$ | $93.8_{\pm 0.1}$ | $93.6_{\pm 0.4}$ |
| | DIR | $41.8_{\pm 20.6}$ | $94.7_{\pm 0.1}$ | $96.3_{\pm 0.2}$ |
| | SMGNN | $89.9_{\pm 1.3}$ | $95.3_{\pm 0.2}$ | $94.9_{\pm 1.0}$ |
| MUTAG | GSAT | $80.6_{\pm 0.3}$ | $79.6_{\pm 1.0}$ | $95.0_{\pm 0.8}$ |
| | DIR | $76.1_{\pm 2.7}$ | $77.5_{\pm 2.3}$ | $92.6_{\pm 2.6}$ |
| | SMGNN | $79.2_{\pm 2.5}$ | $78.2_{\pm 1.1}$ | $94.9_{\pm 0.8}$ |
| SST2P | GSAT | $84.0_{\pm 0.9}$ | $82.6_{\pm 0.7}$ | $97.2_{\pm 1.1}$ |
| | DIR | $84.3_{\pm 0.6}$ | $82.3_{\pm 0.6}$ | $95.8_{\pm 1.2}$ |
| | SMGNN | $83.1_{\pm 0.6}$ | $82.8_{\pm 1.2}$ | $59.2_{\pm 11.5}$ |

**Consequences for evaluating plausibility.** Our attack can be naturally employed to teach SE-GNNs to output plausible but unfaithful explanations. We test this empirically in Appendix E.1, due to space constraints: in this experiment, the attacked SE-GNNs learn to output highly plausible explanations, i.e., explanations very close to human-defined ground-truth explanations, but are shown to also rely on attributes that do *not* appear in the predicted explanations. This result entails that evaluating the plausibility of explanations (Miao et al., 2022a; Wu et al., 2022; Chen et al., 2024; Tai et al., 2025), i.e., how well they match human expectations, does not reflect their trustworthiness.

## 5 RQ2: CAN UNFAITHFUL EXPLANATIONS GO UNDETECTED?

As we just saw, plausibility metrics cannot detect unfaithful explanations. The question is whether metrics explicitly designed to measure faithfulness, in fact, can. This question is currently unanswered: so far, faithfulness metrics have been validated either by comparing them against ground-truth explanations or by correlating them with rankings induced by progressively randomized explanations (Fang et al., 2023; Zheng et al., 2023; 2025). Neither setup explicitly tests whether they can identify known-unfaithful explanations.

We fill this gap by proposing an empirical benchmark (Section 5.1) that exploits the failure cases outlined in Section 3 for evaluating faithfulness metrics in practice. Overall, our results indicate that

Table 3: **Previous metrics can fail to reject degenerate explanations.**. We report $\text{RejRatio}_{\mathcal{I}}$ for representative metrics from Table 1 and for EST (Definition 1). Cases where no edges could be removed are marked with '-'. Bold entries (underlined) indicate best (second best) results.

| Dataset | Model | $\text{RejRatio}_{\mathcal{I}}$ | | | | | | | |
| --- | --- | --- | --- | --- | --- | --- | --- | --- | --- |
| | | Fid- | Fid+ | Suf | Nec | CF | RFid- | RFid+ | EST (ours) |
| RBGV | SMGNN | $12_{\pm 19}$ | $\underline{20}_{\pm 12}$ | $00_{\pm 00}$ | - | $05_{\pm 01}$ | $00_{\pm 00}$ | - | $\mathbf{48}_{\pm 02}$ |
| | GSAT | $12_{\pm 21}$ | $15_{\pm 16}$ | $03_{\pm 03}$ | - | $\mathbf{50}_{\pm 04}$ | $11_{\pm 09}$ | - | $\underline{49}_{\pm 03}$ |
| | DIR | $32_{\pm 23}$ | $27_{\pm 24}$ | $03_{\pm 03}$ | - | $\underline{39}_{\pm 08}$ | $08_{\pm 06}$ | - | $\mathbf{48}_{\pm 03}$ |
| MNISTsp | SMGNN | $88_{\pm 03}$ | $58_{\pm 04}$ | $99_{\pm 01}$ | $55_{\pm 05}$ | $92_{\pm 01}$ | $\underline{99}_{\pm 00}$ | $75_{\pm 05}$ | $\mathbf{100}_{\pm 00}$ |
| | GSAT | $65_{\pm 09}$ | $38_{\pm 05}$ | $\underline{99}_{\pm 01}$ | $44_{\pm 05}$ | $95_{\pm 02}$ | $\underline{99}_{\pm 01}$ | $61_{\pm 04}$ | $\mathbf{100}_{\pm 00}$ |
| | DIR | $93_{\pm 03}$ | $55_{\pm 07}$ | $\underline{99}_{\pm 01}$ | $54_{\pm 05}$ | $91_{\pm 01}$ | $\underline{99}_{\pm 01}$ | $69_{\pm 07}$ | $\mathbf{100}_{\pm 00}$ |
| MUTAG | SMGNN | $59_{\pm 03}$ | $05_{\pm 04}$ | $\mathbf{99}_{\pm 00}$ | $57_{\pm 04}$ | $55_{\pm 07}$ | $72_{\pm 03}$ | $65_{\pm 04}$ | $\underline{96}_{\pm 01}$ |
| | GSAT | $21_{\pm 21}$ | $05_{\pm 02}$ | $\mathbf{99}_{\pm 00}$ | $53_{\pm 07}$ | $68_{\pm 17}$ | $70_{\pm 14}$ | $61_{\pm 05}$ | $\underline{94}_{\pm 05}$ |
| | DIR | $70_{\pm 13}$ | $04_{\pm 02}$ | $\mathbf{99}_{\pm 00}$ | $54_{\pm 02}$ | $69_{\pm 08}$ | $75_{\pm 07}$ | $62_{\pm 04}$ | $\underline{97}_{\pm 02}$ |
| SST2P | SMGNN | $08_{\pm 02}$ | $\underline{44}_{\pm 05}$ | $14_{\pm 05}$ | - | $23_{\pm 07}$ | $04_{\pm 01}$ | - | $\mathbf{54}_{\pm 04}$ |
| | GSAT | $\underline{51}_{\pm 31}$ | $19_{\pm 14}$ | $07_{\pm 10}$ | - | $37_{\pm 18}$ | $14_{\pm 03}$ | - | $\mathbf{62}_{\pm 15}$ |
| | DIR | $\mathbf{50}_{\pm 02}$ | $09_{\pm 06}$ | $12_{\pm 06}$ | - | $42_{\pm 10}$ | $05_{\pm 01}$ | - | $\underline{49}_{\pm 03}$ |

existing faithfulness metrics can be surprisingly ineffective. To amend this issue, we introduce a simple but more reliable metric (Section 5.2).

## 5.1 BENCHMARK OF FAITHFULNESS METRICS

Section 4 shows that one can manipulate SE-GNNs to output degenerate unfaithful explanations. In this section, we exploit this result to construct a controlled benchmark where faithfulness metrics are evaluated based on how well they can judge known-unfaithful explanations, where these are extracted from the manipulated models of Section 4.

**Experimental setup:** We select a set of representative faithfulness metrics from Table 1, one for each family of perturbation. We also include the Counterfactual Fidelity (CF) (Chen et al., 2024), a metric that does not readily fit into Table 1, as it perturbs *both* the complement and the explanation based on the raw relevance scores, which may not be available (Ma et al., 2025). Details about metric implementations are provided in Appendix D.6.

To find a common evaluation setup across metrics, we adopt a similar setting as Bhattacharjee & von Luxburg (2024) where an auditor is asked to validate a set of explanations given by an external provider, rejecting those that are unfaithful. In particular, given a set $D = \{(G_i, R_i)\}$ of input graphs and associated explanations, we define the rejection ratio for metrics estimating *sufficiency* as:

$$\text{RejRatio}_{\mathcal{I}}(D) = \frac{1}{|D|} \sum_{(G,R) \in D} \max_{G' \in \mathcal{I}} \mathbb{1}\{g(e(G)) \neq g(e(G'))\} \tag{4}$$

$\text{RejRatio}_{\mathcal{I}}$ computes the fraction of instances deemed unfaithful by each metric, where $\mathcal{I}$ is the set of perturbations allowed by the metric in use (cf. Table 1). For *necessity* metrics, the inner condition is replaced with $\mathbb{1}\{g(e(G)) = g(e(G'))\}$. To make the computation tractable, Eq. (4) is evaluated on a budget of perturbations per sample, fixed to 50 across metrics. Details are available in Appendix D.2.

**Results:** Table 3 reports the $\text{RejRatio}_{\mathcal{I}}$ values computed for the attacked models of Section 4 and, for completeness, the raw metrics themselves in Table 6. Since those models are trained to output unfaithful explanations intentionally, *we aim to observe as large rejection ratios as possible*. However, Table 3 shows that some metrics can catastrophically fail to reject these explanations, as shown by the ratios around zero for Suf, RFid-, and CF in RBGV, for Fid- in SST2P, and Fid+ in MUTAG.

In conclusion, these results positively answer **RQ2**: *previous faithfulness metrics can fail to detect the known-unfaithful explanations extracted in Section 4*. To solve the undetectability of unfaithful explanations, we introduce an evaluation metric that is shown to be more robust.

## 5.2 SPOTTING THE FAILURE CASES: A RELIABLE FAITHFULNESS METRIC

We observe that existing metrics can fail because each of them focuses on a restricted set of perturbations, such as edge or complement removals, and for each of them, it is possible to construct unfaithful explanations evading detection under the chosen perturbations. For instance, *sufficiency* metrics that only perturb edges cannot probe models that rely primarily on nodes for their decisions.[3] This is best seen in Example 1, where the explanations $u_{green}$ and $u_{violet}$ will go undetected by any metric perturbing only edges, and $u_{green}$ will also be undetected by any metric testing only complement removals, as feeding Eq. (2) with $G' = u_{green}$ preserves the prediction. This observation generalizes to the case where *sufficiency* and *necessity* are evaluated jointly, as discussed in Appendix F.1.

Building on these observations, we propose the *Extension Sufficiency Test* (EST) metric:

**Definition 1** (EST). *Let $G$ be an input graph with prediction $g(e(G))$ and associated explanation $R = e(G)$, and $d(\cdot)$ a suitable distance measure. Then,* EST *estimates the sufficiency of $R$ as follows:*

$$\mathsf{EST}(R, G) = \max_{R \subseteq G' \subseteq G} d\big(g(e(G)), g(e(G'))\big). \tag{5}$$

EST is a simple but robust faithfulness metric that avoids the previously underlined issues by holistically considering *all* supergraphs $G'$ of $R$ contained in $G$, *regardless of what perturbations are necessary for constructing them from $R$*. Note that EST, similarly to previous metrics (Hase et al., 2021; Zheng et al., 2023), may create an OOD supergraph $G'$ whose prediction differs from that of $G$ just because it is OOD, leading to a large (i.e., worse) EST value regardless of whether $R$ was faithful. However, without access to the model or to the data distribution, it is impossible to know whether $G'$ is OOD or if it was obtained by perturbing the features the model is relying on. For this reason, we prioritize a more conservative metric that, although it may flag faithful explanations as unfaithful, it reliably marks the unfaithful ones as such, minimizing the risks of users being deceived. We provide more details and an algorithm implementing EST in Appendix D.6 and Algorithm 1.

To avoid the computationally prohibitive cost of enumerating all possible supergraphs of $R$ within $G$ in Definition 1, we fix a budget of supergraphs to test in practice. Also, note that EST estimates only the *sufficiency* of explanations; since the classifier $g$ may be sensitive to changes in the explanation even when unfaithful, *necessity* metrics are unsuitable for identifying the failure cases in Section 3 (see Appendix F.2). We additionally leverage existing literature on SE-GNNs to theoretically characterize which family of explanations is deemed unfaithful by EST, by relating Definition 1 with increasingly weaker formal notions of explanations (Azzolin et al., 2025b). This analysis, reported in Appendix F.3, shows that EST provably labels the class of explanations including $u_{green}$ and $u_{violet}$ in Example 1 as unfaithful. We now proceed to analyze how EST fares at the benchmark introduced in Section 5.1.

**Results.** The last column of Table 3 lists the results for EST, which is the only metric consistently achieving the highest – or close to thereof – rejection ratios across every configuration while keeping a relatively low standard deviation. In particular, in all the cases where previous metrics score a rejection ratio around zero, EST rejects at least $\approx 50\%$ of degenerate explanations. This result can be easily further improved by increasing the budget of allowed perturbations. In fact, we show in the appendix (Fig. 2) that the rejection ratio of EST monotonically increases with the budget, reaching $\approx 65\%$. Other metrics, instead, either do not benefit from larger budgets or flatten around $\approx 17\%$. In addition, EST also correctly recognizes non-degenerate explanations that include all relevant information, which are instead erroneously rejected by Suf and CF, as shown in Appendix E.4.

We proceed to analyze how EST behaves in natural settings, and to verify whether the failure cases of Section 3 can in fact emerge even without malicious attacks.

## 6 RQ3: CAN DEGENERATE EXPLANATIONS EMERGE NATURALLY?

Finally, we show that SE-GNNs can output degenerate explanations – similar to those presented in Section 4 – without any explicit manipulation, and that EST is effective in finding them.

In practice, we compute EST's rejection ratios for each SE-GNN across the dataset and complement this with a qualitative analysis of explanation examples, reported in the appendix. This analysis aligns

---

[3]Although GNNs ideally depend on both topology and node features (Bechler-Speicher et al., 2025), a reliable faithfulness metric should remain robust when only one of these is truly relevant.

Table 4: **A natural training of SE-GNNs can result in degenerate explanations.** We report test accuracy, rejection ratios for EST, Fid-, and RFid-, and some case-by-case illustrations of representative explanations. AUCROC is reported only for datasets with known ground truth explanation. This analysis is performed only for well-performing models.

| Dataset | Model | Test Acc | AUCROC | $\text{RejRatio}_{\mathcal{I}}$ | | | Illustration |
|---------|-------|----------|--------|-----|-----|-----|-------------|
| | | | | EST | Fid- | RFid- | |
| RBGV | GSAT | $100.0_{\pm 0.00}$ | - | $59.3_{\pm 09.4}$ | $12.0_{\pm 11.9}$ | $6.2_{\pm 12.4}$ | Fig. 11 |
| | DIR ($K = 1\%$) | $98.0_{\pm 0.3}$ | - | $70.2_{\pm 18.8}$ | $27.7_{\pm 27.3}$ | $30.3_{\pm 16.6}$ | Fig. 12 |
| | SMGNN | $98.9_{\pm 0.01}$ | - | $86.8_{\pm 20.4}$ | $54.1_{\pm 45.1}$ | $77.6_{\pm 19.6}$ | Fig. 13 |
| MNISTsp | GSAT | $94.7_{\pm 0.4}$ | $82.5_{\pm 5.5}$ | $2.4_{\pm 2.0}$ | $1.3_{\pm 1.4}$ | $5.3_{\pm 5.6}$ | Fig. 14 |
| | DIR ($K = 10\%$) | $20.0_{\pm 9.0}$ | - | - | - | - | - |
| | SMGNN | $86.8_{\pm 8.0}$ | $43.2_{\pm 18.5}$ | $99.5_{\pm 0.8}$ | $68.9_{\pm 29.2}$ | $95.5_{\pm 4.7}$ | Fig. 15 |
| MUTAG | GSAT | $80.6_{\pm 3.0}$ | - | $64.5_{\pm 7.7}$ | $50.7_{\pm 9.3}$ | $36.8_{\pm 11.7}$ | Fig. 16 |
| | DIR ($K = 1\%$) | $53.5_{\pm 2.2}$ | - | - | - | - | - |
| | SMGNN | $77.9_{\pm 2.4}$ | - | $75.2_{\pm 9.9}$ | $61.9_{\pm 8.9}$ | $50.3_{\pm 9.1}$ | Fig. 17 |
| SST2P | GSAT | $84.0_{\pm 0.9}$ | - | $0.0_{\pm 0.0}$ | $0.0_{\pm 0.0}$ | $0.0_{\pm 0.0}$ | Fig. 18 |
| | DIR ($K = 10\%$) | $82.5_{\pm 0.9}$ | - | $49.5_{\pm 10.1}$ | $37.8_{\pm 18.9}$ | $14.4_{\pm 1.6}$ | Fig. 19 |
| | SMGNN | $83.0_{\pm 0.6}$ | - | $4.3_{\pm 1.0}$ | $1.7_{\pm 0.8}$ | $4.4_{\pm 1.7}$ | Fig. 20 |

with EST's results: when the rejection ratio is high, explanations highlight little-class-discriminative subgraphs; when rejections are near zero, explanations emphasize class-discriminative subgraphs, often aligning with human expectations.

**Setup:** We train the SE-GNN architectures in Table 2 by following their original protocol and natural losses, tuning their hyperparameters to encourage explanation sparsity. For DIR, the original implementation selects explanations based on a relatively large top-$K$; we train the model using a much smaller $K$ instead, as indicated in the results table. This is useful to show that DIR can still encode label-relevant information inside a small subgraph even when the subgraph has no discriminative power by itself. For each model, we report the accuracy, the AUCROC with the ground truth explanations when these are available, and compare the rejection ratios of EST with those of Fid- and RFid- (as baselines). Further details on the experimental setup are available in Appendix D.3.

**Results:** As shown in Table 4, the results are largely consistent with those for naturally trained models in Table 2, with few exceptions. On RBGV and MNISTsp, SMGNN achieves a modest decrease in accuracy ($\approx 2\%$). For DIR, instead, the small $K$ value hinders learning on MUTAG, and the accuracy on MNISTsp ($\approx 20\%$) confirms DIR's unsuitability for this task, in line with the accuracy reported in Table 2 ($\approx 40 \pm 20\%$). In RBGV and SST2P, reducing DIR's $K$ substantially (from 50% to 1% and from 60% to 10%, respectively) results in only a $\approx 2\%$ drop in accuracy. This is remarkable, as DIR does not fit in Theorem 1, and yet can encode label-relevant information inside degenerate explanations, as illustrated in Fig. 12 and Fig. 19. Interestingly, the behavior of EST is contrasting: rejection ratios are high for certain models but almost zero for others, calling for a more detailed analysis, reported below. A case-by-case discussion for some selected examples is reported in the figures listed in Table 4. Our major findings are as follows:

**i) SE-GNNs can output degenerate explanations:** We provide examples of naturally occurring degenerate explanations in Fig. 12, Fig. 13, and Fig. 15, where SE-GNNs highlight green and violet nodes for RBGV, and background pixels for MNISTsp, respectively. Similar examples are also found for DIR on SST2P, where explanations highlight punctuation and stop words (Fig. 19), and for GSAT and SMGNN on MUTAG, where explanations emphasize individual weakly class-discriminative atoms, rather than meaningful functional groups (Fig. 16, Fig. 17). In all cases, SE-GNNs retain high predictive performance, and EST rejects a substantial fraction of explanations ($\geq 50\%$)

**ii) SE-GNNs can also output non-degenerate explanations:** SE-GNNs do not always fail, and the low rejection ratios ($\approx 2\%$) of GSAT on MNISTsp and SST2P suggest the model indeed outputs *sufficient* explanations. In fact, GSAT correctly extracts the digit-subgraph in MNISTsp, as indicated by the high AUCROC (cf. Fig. 14). In SST2P, however, it simply outputs the entire graph as explanation, see Fig. 18. Furthermore, on SST2P, SMGNN consistently highlights emotion-laden words and achieves around zero rejection ratios (see Fig. 20). Note that these results do not contradict Theorem 1, as it provides existential conditions under which SE-GNNs *can* extract degenerate

unfaithful explanations. In practice, stochastic optimization can still steer the model to select other, non-degenerate explanations. Notwithstanding, this behavior remains beyond the practitioner's control, motivating further investigation into more robust SE-GNNs in future work.

**iii) Faithfulness metrics can fail in the wild:** Fid- and RFid- achieve high rejection ratios in `RBGV`, but exhibit substantial seed-to-seed variability, reducing their reliability. For instance, the examples reported in Fig. 12 and Fig. 13 highlight severe failure cases where these metrics mark most of the explanations composed of green and violet nodes as faithful. Similarly, Fig. 15 and Fig. 19 show that Fid- and RFid- can fail to reject explanations highlighting background pixels or punctuation for `MNISTsp` and `SST2P`, respectively. In contrast, EST achieves more consistent results, and it proved helpful in spotting explanations containing only irrelevant information.

In conclusion, the answer to **RQ3** is positive: *naturally trained SE-GNNs can output degenerate explanations, and popular faithfulness metrics can fail to mark them as such.* Conversely, EST proved effective in distinguishing explanations containing or omitting key relevant features.

## 7    RELATED WORK

Popular faithfulness metrics (see Section 2 and Table 1 for a taxonomy) are typically evaluated by their correlation with *i)* human-defined ground-truth explanations or *ii)* rankings from progressively randomized explanations (Fang et al., 2023; Christiansen et al., 2023; Zheng et al., 2023; 2025), but are rarely tested to flag known-unfaithful explanations as such. To address this, in Section 5.1 we introduce a controlled benchmark where SE-GNNs are trained to output known-unfaithful explanations, and metrics are assessed by the fraction of explanations they correctly reject.

Prior work has shown that explanations can be manipulated in ways that evade detection (Dombrowski et al., 2019; Heo et al., 2019; Slack et al., 2020; 2021), with most studies focusing on post-hoc methods for tabular or image data. Furthermore, similar failure cases to the ones presented in Section 3 can be found in rationalization methods for text (Yu et al., 2019), Neuro-Symbolic architectures (Marconato et al., 2023), and Concept Bottleneck Models (Bortolotti et al., 2025). Tai et al. (2025) has highlighted that when not regularized for sparsity, SE-GNNs extract redundant and potentially unfaithful explanations. This work is complementary to ours, in that we show that when SE-GNNs are strongly regularized for sparsity, their explanations can also be unfaithful. We provide an extended discussion on some of this related work in Appendix C due to space constraints.

## 8    CONCLUSION

We outlined a critical failure case of SE-GNNs whereby they output explanations completely unfaithful to their actual inner workings. We provided a sufficient condition under which several SE-GNNs achieve optimal true risk and demonstrated that this condition is met by explanations with no class-discriminative power, which the model could not have used to achieve high accuracy (Section 3). We then showed that a malicious attacker can exploit this fallacy to deliberately conceal the features the model relies on (Section 4), potentially hiding the use of protected attributes. We also observed that these degenerate explanations can emerge naturally (Section 6), underscoring the need for reliable auditing of explanations. Motivated by this, we proposed a benchmark for faithfulness where faithfulness metrics are tested based on how many known-unfaithful explanations they reject, and showed that popular metrics perform poorly (Section 5.1). Finally, to address this shortcoming, we introduced a new metric that is shown to be more effective (Section 5.2).

**Limitations.** To highlight the counterintuitive aspect of our analysis, Theorem 1 is provided for a restricted but representative setting: explanations composed solely of nodes from an anchor set, which the model cannot use in isolation to achieve high accuracy. Nonetheless, it shows that SE-GNNs can output explanations that do not faithfully represent what the model is actually using for inferring predictions, opening the door to several unfaithful behaviors beyond explanations composed of anchor sets. To improve on this, we provide in Appendix B.2 an extended analysis where we generalize the notion of anchor set to the case of subgraphs, showing how popular SE-GNNs can still prefer uncorrelated subgraphs over other, more informative explanations. In both analysis, we assume the explanation extractor to be *hard*: This is a natural and commonly found desideratum for extracting explanations (Yu et al., 2020; Azzolin et al., 2025b), as it represents an extractor maximally

confident of what is relevant and what is irrelevant; This desideratum is already included in the design of practical SE-GNNs, which employ TopK masking (Wu et al., 2022; Deng & Shen, 2024), Gumbel-softmax trick (Miao et al., 2022a;b), or entropy regularization (Lin et al., 2020; Azzolin et al., 2025b), to push the relevance scores to extremal values. While requiring access to the SE-GNN's training process is a strong assumption for our attack in Section 4, it naturally aligns with several practical Machine Learning-as-a-Service scenarios, where the service provider has full control of the model. Furthermore, despite EST effectively detecting unfaithful explanations omitting key relevant features, it cannot identify unfaithful behaviors arising from redundancy, i.e., explanations containing both truly relevant elements and irrelevant ones. Finally, although our focus is on graph classification, our findings carry over directly to node classification. For example, consider a variant of RBGV in which each node must be classified based on whether it has more red or blue neighbors; In this case, SE-GNNs can still encode the overall count of colored neighbors within green and violet nodes, similarly to Example 1.

## ACKNOWLEDGEMENT

Funded by the European Union. Views and opinions expressed are however those of the author(s) only and do not necessarily reflect those of the European Union or the European Health and Digital Executive Agency (HaDEA). Neither the European Union nor the granting authority can be held responsible for them. Grant Agreement no. 101120763 - TANGO and Grant Agreement no. 101120237 - ELIAS. SM acknowledges the support of FWF and ANR project NanOX-ML (6728).

## ETHICS STATEMENT

All authors have read and approved the ICLR Code of Ethics. As for societal consequences, the aim of this work is to shed light on potential misuse of interpretable-by-design neural models, warning users from blindly trusting explanations, and providing a more reliable tool to audit explanations. It can therefore contribute to the development of more trustworthy models for graph-based data and more reliable certifications of explanation validity.

## REPRODUCIBILITY STATEMENT

The proofs of our theoretical results in Section 3 are provided in Appendix B. Details about the experiments, along with details about datasets and metric implementation, are available in Appendix D. Our code is available at https://github.com/steveazzolin/gnn_deg_expl.

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

## A   USE OF LLMS

LLMs were used to polish the writing, to rephrase sentences, and to debug the code. Our manuscript and our code was first human-generated, and then possibly enhanced by LLMs.

## B   PROOFS

### B.1   PROOF OF THEOREM 1

**Theorem 1.** *Let $\mathcal{D}_{\mathcal{G} \times \mathcal{Y}}$ be a data distribution with deterministic ground truth labeling function $\phi : \mathcal{G} \mapsto \mathcal{Y}$, $e$ be a hard explanation extractor, and $\mathcal{Z} = \{z_y\}_{y \in \mathcal{Y}}$ be an anchor set. Then, there exists an explanation extractor $e(G) := z_{\phi(G)}$ and a classifier $g(z_y) := y$, such that the SE-GNN $g \circ e$ implemented by* GSAT, LRI, CAL, GMT-lin *or* SMGNN *achieves optimal true-risk.*

*Proof.* Recall that an SE-GNN $g \circ e$ is composed of an explanation extractor $e$ and a classifier $g$. Also, recall that $\mathcal{G}$ is the set of graphs and $\mathcal{Y}$ the set of labels with deterministic ground truth labeling function $\phi : \mathcal{G} \mapsto \mathcal{Y}$, and that $\mathcal{Z}$ is an anchor set, i.e., a set of single-node subgraphs $\mathcal{Z} = \{z_i\}_{i \in \mathcal{Y}}$ such that $z_i \subseteq G$, for all instances $G \in \mathcal{G}$ and for all $z_i \in \mathcal{Z}$. Then, we show that selecting as the explanation extractor $e(G) := z_{\phi(G)}$ and as the classifier $g(z_y) := y$ achieves optimal true-risk for GSAT, LRI, CAL, GMT-lin, and SMGNN. At a high level, this shows that whenever $e$ can encode the predicted label inside a single node, then the classifier $g$ implementing a simple mapping from these nodes to labels achieves optimal true risk for several SE-GNNs, even when the selected nodes are uncorrelated with the ground truth task.

We restate below two assumptions relevant to our proof:

**Assumption 1.** *$e$ is a hard explanation extractor, i.e., it can only output relevance scores saturated in $\{0, 1\}$ for* SMGNN *and* CAL, *and $\{r, 1\}$ for* GSAT, LRI, *and* GMT-lin.

**Assumption 2.** *The explanation $R$ cannot be empty, i.e., $|R| > 0$. In other words, and following Assumption 1, $\exists u \in V : p_u = 1$.*

Note that $r$ in Assumption 1 is the parameter controlling the degree of stochasticity in the relevance scores of each node, and it therefore acts as an uninformative baseline value. For example, a value of $p_u = 1$ indicates no stochasticity for node $u$, i.e., maximally relevant, while $p_u = r$ indicates maximum stochasticity, i.e., low relevance (Miao et al., 2022a).

Without any loss of generality, we will consider a binary classification setting, i.e., $|\mathcal{Y}| = 2$, and two distinct nodes, $z_0, z_1$ belonging to the anchor set, that is $z_0, z_1 \in \mathcal{Z}$. We proceed to show that, for each of the following SE-GNN, an explanation extractor selecting $z_0$ for $\phi(G) = 0$ and $z_1$ for $\phi(G) = 1$, paired with a suitable classifier $g$, will yield optimal risk. We will analyze each SE-GNN separately, as follows.

**GSAT & LRI (Miao et al., 2022a;b):** The learning objective of both GSAT and LRI is:

$$\mathcal{L}_{clf}(g, e, G, y) + \lambda_1 \sum_{u \in V} p_u \log(\frac{p_u}{r}) + (1 - p_u) \log(\frac{1 - p_u}{1 - r}) \tag{6}$$

where $V$ is the set of nodes for input graph $G$, $p_u$ is the explanation relevance score for node $u$, $r \in [0, 1]$ is the hyperparameter controlling the uninformative prior below which a node is considered to be non-relevant (Miao et al., 2022a), and $\lambda_1$ is an hyperparameter controlling the relative strength of the regularizer over the cross entropy loss $\mathcal{L}_{clf}$.

We now provide a suitable pair of explanation extractor and classifier achieving an optimal value for Eq. (6). For exposition convenience, instead of considering $e$ as a mapping from graph to subgraphs as described in Section 2, we will unpack it as a function from graph to individual node relevance scores, as follows:

$$p_u := e(G)_u = \begin{cases} 1 & \text{if } u = z_0 \wedge \phi(G) = 0 \\ 1 & \text{if } u = z_1 \wedge \phi(G) = 1 \\ r & \text{otherwise} \end{cases} \qquad g(R) = \begin{cases} 0 & \text{if } p_{z_0} > p_{z_1} \\ 1 & \text{if } p_{z_1} > p_{z_0} \\ 0.5 & \text{otherwise} \end{cases} \tag{7}$$

Since $g$ classifies every sample correctly with maximum confidence, $\mathcal{L}(g, e, G, y)$ equals zero. Also, since each $p_u$ equal to $r$ yields zero loss and $e$ highlights only a single node between $z_0$ and $z_1$ per graph, the second term in Eq. (6) boils down to $-\lambda_1 \log(r)$. By Assumption 1 and Assumption 2, it is not possible to find an assignment of explanation scores with more terms equal to $r$ than the explanation provided by Eq. (7), hence $-\lambda_1 \log(r)$ is the minimum risk attainable.

As a remark, note that the classifier in Eq. (7) can indeed be learned by a linear layer with sigmoid output, receiving as input an embedding of the form $\mathbf{h} = [p_{z_1} \mathbb{1}\{z_1 \in R\}, p_{z_0} \mathbb{1}\{z_0 \in R\}]$, with weights $\mathbf{w} = [\alpha, -\alpha]$ and $b = 0$, for a sufficiently large $\alpha$. Note that $\mathbf{h}$ can be recovered by a classifier implementing a GNN with weighted message passing and sum global readout by simply mapping $z_0$ and $z_1$ to two different one-hot encodings, and multiplying this value by the weight given by the relevance scores. Hence, the classifier expressed in Eq. (7) is easily realizable by practical GNNs.

**GMT-lin (Chen et al., 2024)** We observe that `GMT-lin` differs from `GSAT` just in the number of weighted message passing layers (see Appendix E.1 in Chen et al. (2024)). The proof above applies since the underlying training objective remains the same.

**SMGNN (Azzolin et al., 2025b)** The sparsity-based learning objective of `SMGNN` is:

$$\mathcal{L}_{clf}(g, e, G, y) + \frac{\lambda_1}{|V|} \sum_{u \in V} p_u + \frac{\lambda_2}{|V|} \sum_{u \in V} p_u \log(p_u) + (1 - p_u) \log(1 - p_u) \tag{8}$$

where $\mathcal{L}_{clf}$, $V$, $\lambda_1$, $\lambda_2$, and $p_u$ are specular to those for `GSAT` and `LRI`.

We now provide a suitable pair of explanation extractor and classifier achieving an optimal value for Eq. (8).

$$p_u := e(G) = \begin{cases} 1 & \text{if } u = z_0 \wedge \phi(G) = 0 \\ 1 & \text{if } u = z_1 \wedge \phi(G) = 1 \\ 0 & \text{otherwise} \end{cases} \qquad g(R) = \begin{cases} 0 & \text{if } z_0 \in R \wedge z_1 \notin R \\ 1 & \text{if } z_1 \in R \wedge z_0 \notin R \\ 0.5 & \text{otherwise} \end{cases} \tag{9}$$

Then, by plugging Eq. (9) inside the training objective, we get that $\mathcal{L}_{clf}(g, e, G, y) = 0$ since the model achieves correct highly-confident predictions, the middle term becomes $\lambda_1/|V|$ as only one between $z_0$ and $z_1$ gets a score above zero, and the last term becomes zero as relevance scores are binary in $\{0, 1\}$. Hence, Eq. (8) boils down to $\lambda_1/|V|$. By Assumption 1, the only possible smaller assignment of explanation scores would be giving zero score to every node, which, however, violates Assumption 2. Hence, the explanation given by Eq. (9) is the smallest possible, and therefore $\lambda_1/|V|$ is the smallest risk achievable.

As a remark, note that the classifier in Eq. (9) can indeed be learned by a linear layer with sigmoid output, receiving as input an embedding of the form $\mathbf{h} = [\mathbb{1}\{z_0 \in R\}, \mathbb{1}\{z_1 \in R\}]$ similar to the one-hot encoding defined above for `GSAT`, with weights $\mathbf{w} = [-\alpha, \alpha]$ and $b = 0$, for a sufficiently large $\alpha$. Hence, the classifier expressed in Eq. (9) is easily realizable by practical GNNs.

**CAL (Sui et al., 2022):** `CAL` aims at learning an explanation $R = e(G)$ containing the causal features for the prediction, while delegating spurious features to the complement $C = G \setminus R$, by applying causal interventions.

For doing that, it uses a shared explanation extractor $e$ that predicts node and edge explanation relevance scores, which can be stacked into a node-wise attention matrix $M_X$ and an edge-wise attention matrix $M_A$. The original graph $G = (A, X)$ is therefore separated into $R = (A \odot M_A, X \odot M_X)$ and $C = (A \odot 1 - M_A, X \odot 1 - M_X)$. Then, `CAL` instantiates three separate classifiers: $g_R$ to predict the label from the causal explanation; $g_C$ to predict the label from the spurious complement; and $g$ to predict the label from the *implicit intervened graph* defined below. Note that, in this case, $g$ is a classifier mapping from the global graph readout to labels, i.e., $g : \mathbb{R}^d \mapsto \mathcal{Y}$, with $d$ the size of the embedding. Let $\mathbf{h}_R$ be the graph embedding computed by $g_R$ before the classification head, and $\mathbf{h}_C$ be the same graph embedding computed by $g_C$. Then, `CAL` generates *implicit intervened graph* by simulating causal interventions. In practice, this is achieved by summing the embedding of $\mathbf{h}_{R_i}$ for graph $G_i$ with the embedding $\mathbf{h}_{C_j}$ of graph $G_j$ chosen at random, and by predicting the original label of sample $G_i$ (refer to Section 3.4.3 of Sui et al. (2022) for a detailed description). The overall objective amounts to:

$$\mathcal{L}_{clf}(g_R, e, G_i, y_i) + \lambda_1 \mathsf{KL}\big(\mathcal{U}\{1, |\mathcal{Y}|\}, g_C(C_i)\big) + \lambda_2 \mathcal{L}_{interv}(g, \mathbf{h}_{R_i} + \mathbf{h}_{C_j}, y_i) \tag{10}$$

where $\mathcal{L}_{interv}$ is the standard cross entropy loss defined for implicit intervened graphs with $\mathbf{h}_{R_i} + \mathbf{h}_{C_j}$ the input to $g$, $\lambda_1, \lambda_2$ are hyperparameters controlling the relative strength of the regularizers, and $\mathcal{U}\{1, |\mathcal{Y}|\}$ is a discrete uniform distribution over class labels.

Consider now the same construction as for SMGNN:

$$p_u := e(G) = \begin{cases} 1 & \text{if } u = z_0 \wedge \phi(G) = 0 \\ 1 & \text{if } u = z_1 \wedge \phi(G) = 1 \\ 0 & \text{otherwise} \end{cases} \qquad g_R(R) = \begin{cases} 0 & \text{if } z_0 \in R \wedge z_1 \notin R \\ 1 & \text{if } z_1 \in R \wedge z_0 \notin R \\ 0.5 & \text{otherwise} \end{cases} \qquad (11)$$

Also, since $g_C$ is trained to output maximally uncertain predictions for any input (second term in Eq. (10)), a possible trivial solution would be having $\mathbf{h}_C = \vec{0}$ for any $C$, with an associated constant classifier outputting $1/|\mathcal{Y}|$ for each class, i.e., $g_C(C) := 1/|\mathcal{Y}|$. Note that any $g_C$ trained with weight decay is encouraged to converge to this solution, as no term in Eq. (10) is pushing $\mathbf{h}_C$ to learn a meaningful representation. Together, these observations imply that both the first and second terms in Eq. (10) are zero. Furthermore, since we fixed $\mathbf{h}_C = \vec{0}$, then the last term in Eq. (10) boils down to:

$$\lambda_2 \mathcal{L}_{interv}(g, \mathbf{h}_{R_i}, y_i) \qquad (12)$$

Therefore, the solution where the classification head $g$ equals that of $g_R$ implies that all three terms are zero.

**Final remarks:** Note that our examples are provided assuming the classification head of $g$ is a linear layer. Nonetheless, our examples generalize to the case where multi-layer perceptions are used, as commonly found in practical implementations of SE-GNNs. □

### B.2 EXTENDED ANALYSIS OF THE ANCHOR SET

We provide a generalization of the theoretical analysis in Section 3 to the case where the anchor set is defined in terms of subgraphs rather than single nodes, and where each of these subgraphs is not required to appear in *all* graphs. With this new setting, we will show that the risk attained by several popular SE-GNNs when selecting uninformative subgraphs can be lower than the risk attained by selecting other, more informative explanations $R_y^*$. To keep the discussion intuitive – and in line with previous work studying SE-GNNs from a causal lens (Miao et al., 2022a; Chen et al., 2022; Azzolin et al., 2025a) – we will discuss the specific case where $R_y^*$ is the ground truth explanation for class $y$.[4] Nonetheless, as we will discuss, our insights will equally apply to any other notion of explanation. Let us now extend the anchor set definition provided in Section 3 to the case of subgraphs.

**Definition 2** (Subgraph anchor set). *Let $\mathcal{D}_{\mathcal{G} \times \mathcal{Y}}$ be the data distribution, and $\mathcal{A} = \{\bar{G}_i\}_{i=1}^m$ a set of generic subgraphs $\bar{G}$ that can appear in any graph $G \in \mathcal{G}$, with $m \geq |\mathcal{Y}|$. The partition of $\mathcal{A}$ into $\mathcal{Z}' = \{\{\bar{G}_i^y\}_{i=1}^{m_y}\}_{y \in \mathcal{Y}}$ with $\sum_y m_y = m$ is called a **subgraph anchor set** if:*

1. *(**Per-label coverage**) For every $(G, y)$ in the support of $\mathcal{D}_{\mathcal{G} \times \mathcal{Y}}$ there exists at least one subgraph $\bar{G}_i^y \in G$.*

2. *(**Disjointness of partitions**) The partitions are pairwise disjoint: $\{\bar{G}_i^y\}_i \cap \{\bar{G}_i^{y'}\}_i = \varnothing$ for all $y \neq y'$.*

Note that $\mathcal{A}$ constitutes an arbitrary subset of possible subgraphs appearing in the dataset, with no additional assumptions on the structure of the graphs. We will use $\mathcal{A}$ to represent the possible subgraph explanations that are provided by a generic $e$ learned on $\mathcal{D}_{\mathcal{G} \times \mathcal{Y}}$. Definition 2 allows for $\mathcal{A}$ containing subgraphs $\bar{G}_i$ carrying no information about class labels, meaning they are neither ground truth nor spurious correlation. In fact, anchor sets (as originally defined in Section 3) are a specific case of subgraph anchor sets, where $m = |\mathcal{Y}|$, and where each $\bar{G}_i$ is a single-node subgraph appearing in all graphs.

Importantly, Definition 2 generalizes the original definition of node-wise anchor set (c.f. Section 3) in the following ways: First, it considers subgraphs instead of single nodes. Second, it relaxes the

---

[4]Although $R_y^*$ may depend on the specific graph instance, we consider a fixed subgraph for the ease of the argument. This remains general enough to capture most SE-GNN use-cases commonly found in the literature, like motif-based tasks (Miao et al., 2022a; Chen et al., 2022; Azzolin et al., 2025a).

condition that anchor nodes must be present in all graphs to the case where different subgraphs can appear in different graphs (see *Per-label coverage* in Definition 2).

We now proceed to show that many popular SE-GNNs can achieve a lower risk by picking explanations from the subgraph anchor set rather than the ground truth $R_y^*$. Crucially, this holds even when $\mathcal{A}$ contains only subgraphs with no class-discriminative power, showing how the degenerate explanations discussed in Section 3 can emerge also in the presence of subgraph explanations.

**Theorem 2.** *Let $\mathcal{D}_{\mathcal{G} \times \mathcal{Y}}$ be a data distribution with deterministic ground truth labeling function $\phi$ : $\mathcal{G} \mapsto \mathcal{Y}$ and ground truth explanation $R_y^*$, $e$ be a hard explanation extractor, and $\mathcal{Z}' = \{\{\bar{G}_i^y\}_i\}_{y \in \mathcal{Y}}$ be a subgraph anchor set. If $|R_y^*| \geq \max_i |\bar{G}_i^y|$ for all labels $y$, then there exists a SE-GNN $g \circ e$ such that the true risk of GSAT, LRI, CAL, GMT-lin and SMGNN will be lower or equal for*

$$e(G) := \operatorname*{argmin}_{\bar{G}_i^{\phi(G)} \in G} |\bar{G}_i^{\phi(G)}| \qquad and \qquad g(\bar{G}_i^y) := y$$

*than*

$$e^*(G) := R_y^* \qquad and \qquad g^*(R_y^*) := y.$$

*Proof.* To keep the proof simple, we will use the same setting of Theorem 1 for a boolean classification task, and reuse some intermediate results.

**GSAT, GMT-lin, & LRI:** Let us consider the following pair of explanation extractor and classifier:

$$R := e(G) = \begin{cases} \operatorname*{argmin}_{\bar{G}_i^0 \in G} |\bar{G}_i^0| & \text{if } \phi(G) = 0 \\ \operatorname*{argmin}_{\bar{G}_i^1 \in G} |\bar{G}_i^1| & \text{if } \phi(G) = 1 \end{cases} \qquad g(R) = \begin{cases} 0 & \text{if } R \in \{\bar{G}_i^0\}_i \\ 1 & \text{if } R \in \{\bar{G}_i^1\}_i \\ 0.5 & \text{otherwise} \end{cases} \tag{13}$$

In the equation above, $\operatorname*{argmin}_{\bar{G}_i^0 \in G} |\bar{G}_i^0|$ means that $e$ will select the smallest subgraph $i$ belonging to partition $y = 0$ that is present in the subgraph anchor set for the input sample $G$. The classifier $g$, instead, is just left with inferring to which partition the explanation belongs, and with outputting the corresponding partition index. In addition, to preserve the semantics of Miao et al. (2022a) and in accordance with Assumption 1, we assume that each node in $R$ is given a score of 1, while the rest is given a score of $r$.

Since $g$ replicates the predictions of the ground truth labeling function $\phi$ – hence it classifies every sample correctly with maximum confidence by construction – $\mathcal{L}(g, e, G, y)$ in Eq. (6) equals zero. Also, since each node with score equal to $r$ yields zero loss, and each node with score equal to 1 yields a loss value of $\lambda_1 \log(r)^{-1}$ (see proof of Theorem 1), the true risk is bounded above as follows:

$$\mathbb{E}_{(G,y)\sim\mathcal{D}_{\mathcal{G}\times\mathcal{Y}}}\left[\lambda_1 |e(G)| \log(r)^{-1}\right] \leq \mathbb{E}_{(G,y)\sim\mathcal{D}_{\mathcal{G}\times\mathcal{Y}}}\left[\max_i \lambda_1 |\bar{G}_i^y| \log(r)^{-1}\right] \tag{14}$$

By repeating the same analysis for the following ground truth explanation extractor $e^*$ and ground truth classifier $g^*$:

$$R := e^*(G) = R_y^* \qquad g^*(R) = \begin{cases} 0 & \text{if } R = R_0^* \\ 1 & \text{if } R = R_1^* \\ 0.5 & \text{otherwise} \end{cases}, \tag{15}$$

we get that the true risk when selecting the ground truth explanation $R_y^*$ equals:

$$\mathbb{E}_{(G,y)\sim\mathcal{D}_{\mathcal{G}\times\mathcal{Y}}}\left[\lambda_1 |R_y^*| \log(r)^{-1}\right] \tag{16}$$

It follows that, whenever $|R_y^*| \geq \max_i |\bar{G}_i^y|$ for all labels $y$, the true risk of Eq. (14) will be smaller or equal to that of Eq. (16).

**SMGNN:** The proof works similarly to above. Let us consider the following SE-GNN:

$$R := e(G) = \begin{cases} \operatorname*{argmin}_{\bar{G}_i^0 \in G} |\bar{G}_i^0| & \text{if } \phi(G) = 0 \\ \operatorname*{argmin}_{\bar{G}_i^1 \in G} |\bar{G}_i^1| & \text{if } \phi(G) = 1 \end{cases} \qquad g(R) = \begin{cases} 0 & \text{if } R \in \{\bar{G}_i^0\}_i \\ 1 & \text{if } R \in \{\bar{G}_i^1\}_i \\ 0.5 & \text{otherwise} \end{cases} \tag{17}$$

$\mathcal{L}(g, e, G, y)$ in Eq. (8) is again zero for the reasons above, and since Eq. (8) penalizes the loss by a factor of $\lambda_1/|V|$ for each node in $R$, it reduces to $\lambda_1|e(G)|/|V|$ (see proof of Theorem 1). The true risk is then upper bounded as follows:

$$\mathbb{E}_{(G,y)\sim\mathcal{D}_{\mathcal{G}\times\mathcal{Y}}}\Big[\lambda_1\frac{|e(G)|}{|V|}\Big] \leq \mathbb{E}_{(G,y)\sim\mathcal{D}_{\mathcal{G}\times\mathcal{Y}}}\Big[\max_i \lambda_1\frac{|\bar{G}_i^y|}{|V|}\Big] \tag{18}$$

By repeating the same analysis for the same ground truth explanation extractor $e^*$ and ground truth classifier $g^*$ of Eq. (15), the true risk equals:

$$\mathbb{E}_{(G,y)\sim\mathcal{D}_{\mathcal{G}\times\mathcal{Y}}}\Big[\lambda_1\frac{|R_y^*|}{|V|}\Big] \tag{19}$$

It follows that, whenever $|R_y^*| \geq \max_i |\bar{G}_i^y|$ for all labels $y$, the true risk of Eq. (18) will be smaller or equal to that of Eq. (19).

**CAL:** Exactly as in the original proof of Theorem 1, we can set $\mathbf{h}_C \equiv 0$ and train $g_R$ and $g$ to predict the right label from the explanation only, achieving zero classification and intervention loss in Eq. (10) for both choices of explanation extractor ($e$ or $e^*$). Moreover, we can choose $g_C$ to output the uniform distribution on every input, making the KL term zero as well as in the proof of Theorem 1. Hence, both SE-GNNs attain the same value of the CAL objective, independently of $|R|$. $\qquad\square$

As a remark, note that Theorem 2 does not depend on the specific form of $R_y^*$, but relies only on its cardinality $|R_y^*|$. Therefore, the same result applies to any other notion of explanation $R_y^*$ beyond ground truth explanations, like explanations highlighting spurious correlation (Wu et al., 2022), invariant subgraphs (Chen et al., 2022), or minimal label-preserving subgraphs (Azzolin et al., 2025b). That is, SE-GNNs in Theorem 2 can prefer to highlight smaller uninformative explanations over other reasonable notions of explanations.

## C    EXTENDED RELATED WORK

**Reasoning shortcuts.** Concept bottleneck models (CBMs) introduce an intermediate layer of human-interpretable concepts between inputs and predictions, enabling users to inspect and intervene on a model's reasoning process. While this paradigm promises greater transparency and controllability, recent work has raised concerns about its reliability. In particular, Bortolotti et al. (2024) show that CBMs are vulnerable to *reasoning shortcuts*: instead of leveraging the intended causal relationships between concepts and outcomes, models may create unwanted dependencies and semantic associations in the concept space, thereby undermining the very interpretability of CBMs. While CBMs are expected to learn human-aligned concepts, SE-GNNs are only designed to *declare* which input patterns/features they are relying on, regardless of whether it is the intended pattern or a mere spurious correlation. Therefore, reasoning shortcuts can be seen as an issue of poor alignment between humans and machines, whereas the issue highlighted in Section 3 is poor alignment between the predictive behavior of SE-GNNs and their explanations. In fact, the only *fully reliable* mitigation strategy outlined in Marconato et al. (2023) involves dense human supervision on concepts – a setting that does not apply to SE-GNNs, which are not defined over human-specified concepts.

**Degenerate explanations.** Jethani et al. (2021) was among the first works to highlight that explanation-based architectures can learn to encode predictions in their explanations and to propose amelioration. Nonetheless, Hsia et al. (2024) showed that the solution proposed by Jethani et al. (2021) can be easily hijacked into considering degenerate explanations as legitimate. Then, Puli et al. (2024) proposes an extension to Jethani et al. (2021), encompassing two surrogate models; one to approximate the true label posterior (as in Jethani et al. (2021)), and the other to estimate the Mutual Information between the selected mask and the target label. Their solution, however, does not readily apply to the graph setting. In fact, they propose to learn a classifier $\phi$ that respects the true data distribution by training $\phi$ over any possible randomly generated explanation $R$, where $R$ is sampled IID for each input feature from a Bernoulli distribution. This is impractical for a graph task like counting (cf. RBGV), as randomly removing input nodes can change the ground truth label of the graph, resulting in a contradictory supervisory signal.

Other approaches to contrast degenerate explanations collect annotated explanations to teach the model to rely less on spuriously correlated ones (Yue et al., 2024). Nonetheless, as we show in

Appendix E.1, supervising the SE-GNNs' explanations may not be enough to ensure the faithfulness of explanations.

**Rationalization methods for NLP.** Rationalization methods aim at extracting a rationale for model prediction together with the final explanation, sharing the same spirit as SE-GNNs. They're mostly used for NLP tasks.

Yu et al. (2019) formalized the problem of degeneracy as explanations encoding labels via trivial patterns. However, they did not provide a formal analysis of which rationalization methods can learn such degenerate solutions, nor a sound methodology for identifying when degenerate explanations appear in the wild. On the contrary, Theorem 1 formally unpacks commonly found loss functions of popular SE-GNNs, showing how models are implicitly optimize for such cases. This result hold beyond models promoting conciseness via sparsity, which is instead the main focus of Yu et al. (2019). At high level, Yu et al. (2019) proposes to add a complement predictor, trained adversarially to predict the correct label from information left out of the explanation. Thanks to the adversarial training, the explanation extractor is pushed to not leave any useful information in the complement, thus including all informative features in the explanation. A similar solution was also proposed by Liu et al. (2024a), where the complement predictor is shared with the final classifier. While this may effectively promote non-degenerate explanations, preliminary experiments on a SE-GNN implementing a similar type of mitigation have shown it may not be enough to guarantee non-degeneracy (see `DIR` in Section 6). Additionally, this mitigation promotes the explanation extractor to include potentially redundant and spuriously correlated information inside $R$, resulting in more complex explanations.

A related solution was proposed by Chang et al. (2019), which adopts three different players for each output class trained in a zero-sum game to output class-specific rationales. Nonetheless, this approach scales poorly with the number of classes, and the zero-sum game of the three players is notoriously hard to balance (Mescheder et al., 2018; Farnia & Ozdaglar, 2020).

Yu et al. (2021) investigates the problem of *model interlocking*, where the classifier overfits to little informative explanations at early stages of training, and prevents the explanation extractor from finding more informative ones. This is, however, orthogonal to the degeneracy problem, as degenerate explanations can be *optimal* from a predictive accuracy perspective.

Liu et al. (2022) advocates that instead of relying on complex regularization components, employing a unified encoder for both the explanation extractor and the predictor is enough to avoid degeneracy. Nonetheless, this can serve as a useful inductive bias, but it does not completely rule out the possibility of learning degenerate explanations, as shown in our experiments with `GSAT` of Section 6, which naturally employs shared extractor-classifier (Miao et al., 2022a).

Zhang et al. (2023) propose to optimize for causal non-spuriousness and efficiency during training, which is shown to yield less spuriously correlated explanations. However, this does not tackle degeneracy, as degeneracy is about the model outputting unfaithful explanations, and this issue is not caused by spurious correlations.

Hu & Yu (2024) proposes to employ a guidance rationalization module, trained with soft relevance scores instead of the binary scores employed by the core rationalization module. Then, the core rationalization module is trained to match the predictions and rationales of the guidance module. The idea is that if the guidance module learns non-degenerate explanations thanks to its broader receptive field, then the final core rationalization module will also avoid degenerate explanations. Nonetheless, models trained with soft relevance scores can also fall into degeneracy, as we have shown in our work.

Liu et al. (2024b) proposes to use a standard black-box model as guidance. Then, the explanation extractor is trained to extract explanations for the black box's predictions and optimize the classifier, which receives such explanations as input only. This approach requires however to explain a black-box model in a post-hoc fashion similar to Luo et al. (2020). It thus deviates from our goal of devising an intrinsically ante-hoc self-explainable model.

Liu et al. (2025) proposes an adversarial training where the adversary tries to predict a random class by exploiting any information in the graph (thus not limited to the complement of the main explanation $R$), whereas the explanation extractor and the predictor are trained to make accurate predictions and to predict the adversarial explanation with maximum uncertainty. The proposed method is grounded on the assumption that ground truth rationales for a specific class $y$ appear more

often in samples of class $y$. While being reasonable for text-like input, this might not hold for graphs – for `RBGV` you can construct an infinite number of negative graphs that contain any arbitrary number of red nodes.

In all those cases, to the best of our knowledge, a robust framework to audit these explanation failures remains largely unexplored, and results highlight its highly challenging computational complexity (Bhattacharjee & von Luxburg, 2024). Furthermore, our experiments in Appendix E.1 show that even when explanations are optimized to match human expectations – thus avoiding degeneracy through supervision – the model can still conceal its use of protected attributes, revealing a fundamental misalignment between the explanations and the model's actual decision process. This goes beyond degeneracy alone and towards the broader challenge of explanation-model misalignment.

**Self-explainable GNNs.** In our work, we focused on Self-explainable GNNs (SE-GNNs) that have been introduced to overcome the intrinsic limits of post-hoc methods (Ying et al., 2019; Luo et al., 2020; Yuan et al., 2021; Azzolin et al., 2022; Yuan et al., 2022; Li et al., 2024b). Our focus is on SE-GNNs composed of an explanation extractor and a classifier, jointly trained. While being very general, alternative formulations of SE-GNNs exist, such as prototype-based models (Zhang et al., 2022; Ragno et al., 2022; Dai & Wang, 2021), meta-paths-based (Ferrini et al., 2024), Koopman theory-based (Guerra et al., 2024), Decision Tree-based (Bechler-Speicher et al., 2024b), or other model-specific techniques (Yu et al., 2020; 2022; Giunchiglia et al., 2022; Serra & Niepert, 2022; Spinelli et al., 2023). Furthermore, Müller et al. (2023), Bechler-Speicher et al. (2024a), and Zerio et al. (2025) introduced different families of Self-explainable GNNs that avoid extracting a subgraph explanation altogether, either by distilling the GNN into a Decision Tree, or by relying on learnable shape functions, respectively. Azzolin et al. (2025b), instead, proposed an hybrid approach to extract subgraph-based explanations *together* with other forms of explanations, like rule-based explanations (Armgaan et al., 2024; Pluska et al., 2024; Rissaki et al., 2025).

**Robustness of explanations.** Several works have outlined that GNN explanations are fragile and can be easily broken (Li et al., 2024a;b; 2025). While their analysis mostly pertains to post-hoc explanations, a natural question is whether the problem we outlined in Section 3 can be related to a lack of robustness. For instance, Li et al. (2024a;b; 2025) show that it is possible to adversarially perturb the input graph to drastically change the provided explanation while leaving the prediction intact. We note, however, that this does not necessarily apply to the failure case of Section 3. To illustrate this, we provide two examples of SE-GNNs that output unfaithful, degenerate explanations. One of them is also susceptible to the explanation brittleness phenomenon described in (Li et al., 2024a), whereas the other is not:

- **Degenerate and fragile SE-GNN:** Let us consider a toy binary graph-classification task in which each graph is composed of randomly attached nodes colored either red or blue. A graph is labeled positive if the number of blue nodes exceeds the number of red nodes. Each instance is additionally augmented with some fixed uncolored motifs (4-star, triangle, 6-clique): positive graphs contain the 6-clique and at least one between a 4-star and a triangle, whereas negative graphs contain the 6-clique and may or may not contain any other motifs. We now consider the following subgraph anchor set (see the extended analysis in Definition 2):

$$\mathcal{Z}' = \{\{\bar{G}_0^0 = \text{clique}\}^0, \{\bar{G}_0^1 = \text{star}, \bar{G}_1^1 = \text{triangle}\}^1\}$$

  Note that $\mathcal{Z}'$ contains only motifs with no label-discriminative power by construction. Therefore, any accurate SE-GNN that outputs explanations restricted to $\mathcal{Z}'$ produces unfaithful, degenerate explanations. Similarly to Theorem 2, we provide below a valid construction of such a case:

$$e(G) = \underset{\bar{G}_i^y \in G}{\arg\min} |\bar{G}_i^y| \quad \text{and} \quad g(\bar{G}_i^y) = y$$

  where $y$ is the graph label inferred by $e()$. Consider now a positive graph $G$ that contains the clique and the star, so that the explanation extracted by the model highlights the star. Applying the attack in Li et al. (2024a), we aim to modify the explanation while keeping the predicted label unchanged. A simple input manipulation consists of adding a single edge between any two leaves of the star, thereby creating a triangle. Because the triangle is a smaller motif, the explanation extractor will now prefer selecting it, thus changing the explanation without affecting the prediction.

- **Degenerate and robust SE-GNN:** Example 1 is already an example of a robust, degenerate SE-GNN. This is because the SE-GNN cannot change the predicted explanation unless the

relative numerosity of red and blue is altered, which also results in a change of the graph label. However, this violates the constraints of the attack proposed by Li et al. (2024a), making the attack unfeasible.

These examples together show that SE-GNNs can also be affected by fragile explanations, and it is possible to manipulate their explanations via the attack proposed in Li et al. (2024a;b; 2025). However, the failure case outlined in Section 3 *is not a consequence* of this brittleness.

Li et al. (2025) further considers another type of attack, designed to change the prediction itself. In this case, robustness indeed can play a role in the explanation's faithfulness. For instance, EST (Definition 1) can be seen as measuring the robustness of the model to changes outside of the explanation, and devising an SE-GNN that is robust to such perturbations will naturally lead to non-degenerate explanations (c.f. Appendix F.3). This goal is, however, different from Li et al. (2025), which, instead of training an SE-GNN to be intrinsically robust, builds a *wrapper* around an already trained GNN to make it more robust post hoc. A validation of the most effective strategy is left as an interesting future work.

## D    IMPLEMENTATION DETAILS

We relied on the codebase provided by Gui et al. (2022), which implements `GSAT` and `DIR`. The implementation of `SMGNN` is derived from Azzolin et al. (2025b). Every model uses `ACR` (Barceló et al., 2020) as backbone, where each operation is modified to incorporate the conditioning on explanation relevance scores as follows:

$$h_u^i = COM^i\big(Upd^i(p_u h_u^{i-1}), Aggr(\{\{p_v h_v^{i-1} \mid v \in \mathcal{N}_G(u)\}\}), Read(\{\{p_v h_v^{i-1} \mid v \in V\}\})\big) \tag{20}$$

where $Read$ and $Aggr$ are defined as the sum, $Upd^i$ and $COM^i$ as a 3-layer MLP for `MNISTsp` and `MUTAG`, and as a 2-layer MLP for other datasets. This choice of model backbone was driven by the fact that `ACR` has a precise mapping to the range of tasks it can solve (Barceló et al., 2020), and therefore guides the range of tasks for which the explanation extractor can meet the conditions highlighted in Theorem 1, which nonetheless remains valid for any backbone. In the SE-GNN's classifier, each MLP has the bias term deactivated, discouraging the emergence of *default* predictions (Faber et al., 2021). Also, the final graph global readout implements *Explanation Readout* (Azzolin et al., 2025a), i.e., it weights the final node embeddings based based on their relevance score. Further details about hyperparameters and the dataset are provided below.

### D.1    IMPLEMENTATION DETAILS OF SECTION 4

#### D.1.1    NATURAL TRAINING

When training models without malicious attacks, we stick to the original training procedure of each SE-GNN. In particular, we fix the hyperparameters as follows:

**RBGV:** Each model is trained with a batch size of 64 and for a number of epochs set to 200 with a learning rate of 0.0001 and without weight decay or dropout. Also, we fix the global readout to sum, and use 2 layers for both the explanation extractor and classifier. The embedding dimension is set to 100. For `GSAT`, we set $r = 0.3$ and $\lambda = 0.1$. For `SMGNN`, we set $\lambda_{spars} = 0.4$, $\lambda_{entr} = 0.1$. For `DIR`, we set K to 0.5 and $\lambda = 10$.

**MNISTsp:** Each model is trained with a batch size of 256 and for a number of epochs set to 200 with a learning rate of 0.001 and without weight decay or dropout. Also, we fix the global readout to sum, and use 2 layers for both the explanation extractor and classifier. The embedding dimension is set to 300. For `GSAT`, we set $r = 0.7$ and $\lambda = 0.1$. For `SMGNN`, we set $\lambda_{spars} = 0.01$, $\lambda_{entr} = 0.015$. For `DIR`, we set K to 0.8 and $\lambda = 0.0001$. `SMGNN` and `DIR` further adopt Batch Normalization across GNN layers (Ioffe, 2015).

**MUTAG:** Each model is trained with a batch size of 64 and for a number of epochs set to 100 with a learning rate of 0.001 and without weight decay or dropout. Also, we fix the global readout to sum, and use 2 layers for both the explanation extractor and classifier. The embedding dimension is set to 64. For `GSAT`, we set $r = 0.7$ and $\lambda = 1$. For `SMGNN`, we set $\lambda_{spars} = 0.001$, $\lambda_{entr} = 0.008$. For `DIR`, we set K to 0.5 and $\lambda = 0.0001$.

**SST2P:** Each model is trained with a batch size of 256 and for a number of epochs set to 100 with a learning rate of 0.001 and without weight decay. The dropout ratio is set to 0.3. Also, we fix the global readout to mean, and use 2 layers for both the explanation extractor and classifier. The embedding dimension is set to 64. For `GSAT`, we set $r = 0.7$ and $\lambda = 1$. For `SMGNN`, we set $\lambda_{spars} = 0.1$, $\lambda_{entr} = 0.1$. For `DIR`, we set K equals 0.6 and $\lambda = 10$. In this particular case, we slightly changed the learning rate to 0.0002 and the training batch size to 128, which we found helped with convergence. `SMGNN` further employs Batch Normalization across GNN layers (Ioffe, 2015).

### D.1.2 MALICIOUS TRAINING

Since we aim to maliciously attack the model, we assume full control over the model's training. While this is a limitation of our attack, it fits the scenario in which an external service provider want to conceal the use of protected attributes, highlight a risk in the trustworthiness of the explanations they provide. Regardless, we show in Section 6 that explanations similar to those extracted in Section 4 also appear in natural scenarios, further motivating the study of this weakness. When training models under the attack presented in Section 4, the objective becomes the one illustrated in Eq. (3). Other explanation-regularization losses are deactivated to avoid conflict with the explanation-supervision loss of Eq. (3). The other hyperparameters, including the training batch size, remain unchanged with respect to the ones reported above.

Since the supervision provided by Eq. (3) can be highly class-unbalanced, with many more negatives than positives, we manually reweigh by 10 the contribution from positive nodes for `RBGV`, and by 100 for other datasets. Then, SE-GNNs are trained for a maximum of 1500 epochs, and we define the stopping criterion as a condition on the SE-GNN's loss and accuracy on the training set, as follows:

For `DIR`, we set the threshold on minimum accuracy as 98 for `MNISTsp` and `CPatchMNIST`, 95 for `MUTAG`, and 99 for `SST2P` and `RBGV`. The threshold on minimum classification loss is instead set as 0.01 `MNISTsp`, `CPatchMNIST`, and `RBGV`, 0.08 for `MUTAG`, and 0.015 for `SST2P`. For `GSAT` and `SMGNN`, instead, the threshold on minimum accuracy is set to 95 for `MNISTsp`, `CPatchMNIST`, and `MUTAG`, 96 for `SST2P`, and 99 for `RBGV`. The threshold on minimum classification loss is instead set to 0.08 `MNISTsp`, `CPatchMNIST`, and `MUTAG`, 0.015 for `SST2P`, and to 0.01 for `RBGV`.

Those values were manually defined after inspecting explanations on the validation set. The final model is picked as the last checkpoint when training is stopped.

### D.2 IMPLEMENTATION DETAILS OF SECTION 5

The rejection ratio computed in Eq. (4) provides a more interpretable way to judge the faithfulness of explanations. Typical metrics, in fact, rely on the average change in class probability, which yields scores whose upper-bound depends on the underlying distance function $d$ used, like Total variation or Kullback–Leibler. Some metrics further non-linearly normalize those values to squash them in the fixed $[0, 1]$ range, introducing further sources of variation (Azzolin et al., 2025a). This makes it harder to find a precise rejection criterion separating *good* from *bad* metric values, as it requires defining a threshold on the expected change. To avoid this issue, we opt for the max aggregation in Eq. (4), providing a best-case evaluation.

At a high level, Eq. (4) computes the fraction of input graphs whose prediction can be changed after applying the perturbations induced by each faithfulness metric, i.e., according to $\mathcal{I}$. To allow for some slack, we consider a perturbation as prediction-changing if it makes the classifier's output highly uncertain, e.g., if the predicted probability lies in $[0.4, 0.6]$ for binary classification.

### D.3 IMPLEMENTATION DETAILS OF SECTION 6

In Section 6 we provide a similar natural training of SE-GNNs as Section 4, with the only difference that hyperparameters are optimized to increase the sparsity of explanations, while making sure that the final accuracy is not significantly penalized. All the other aspects of model training are kept unaltered. We detail below the chosen hyperparameters for this analysis, reporting only those that differ from Appendix D.1.1.

**RBGV:** `SMGNN` is trained with $\lambda_{spars} = 0.4$ and $\lambda_{entr} = 1.0$, and with Batch Normalization across GNN layers. `GSAT` is kept the same as in Appendix D.1.1, while `DIR` uses a topK selection with K

equals $1\%$ as discussed in Section 6. This particularly low value is set on purpose and to verify that `DIR` can learn to squash all the relevant information inside a small, potentially unrelated, subgraph.

**MNISTsp:** SMGNN is trained with $\lambda_{spars} = 1$ and $\lambda_{entr} = 1.5$. GSAT is kept the same as in Appendix D.1.1, while `DIR` uses a topK selection with K equals $10\%$. We note that `DIR` reportedly performs poorly on the `MNISTsp` dataset (Wu et al., 2022), and even using larger K does not yield good results.

**MUTAG:** SMGNN is trained with $\lambda_{spars} = 0.1$, $\lambda_{entr} = 0.8$. GSAT is kept the same as in Appendix D.1.1, while `DIR` uses a topK selection with K equals $10\%$. This is the only case where setting a significantly lower K alters performance compared to a larger K.

**SST2P:** GSAT and SMGNN are kept the same as in Appendix D.1.1, while `DIR` uses a topK selection with K equals $10\%$. We noticed that SMGNN trained for random seed number 5 yields all scores squashed to 0, and we removed it from our analysis. Similarly, random seed 3 for `DIR` yielded significantly lower accuracy, and we also removed it.

## D.4 DATASETS

We considered the following datasets in our analysis:

- `RBGV` (ours). Nodes are colored with a one-hot encoding of either red, blue, green, or violet. The task to predict is whether the number of blue nodes is larger than the number of red ones. The topology is randomly generated from a Barabási-Albert distribution Barabási & Albert (1999), and we ensure that green and violet nodes are always disconnected. Each graph contains a total number of red and blue in the range $[0, 100]$, plus a single green and a single violet node. The total dataset size is 5000 graphs, divided with an 80/10/10 split into train, validation, and test sets, respectively.

- `MUTAG` (Debnath et al., 1991), abbreviation for Mutagenicity, is a molecular property prediction dataset, where each molecule is annotated based on its mutagenic effect. The nodes represent atoms and the edges represent chemical bonds. The dataset is composed of 4337 graphs, where negative labels indicate mutagenic molecules and positive non-mutagenic ones.

- `MNISTsp` (Knyazev et al., 2019) converts the popular MNIST image-based digit recognition dataset (Lecun et al., 1998) into graphs by applying a super pixelation algorithm. Nodes are then composed of superpixels, while edges follow the spatial connectivity of those superpixels.

- `CPatchMNIST` (ours) is an extension of `MNISTsp` where we color the top-left and bottom-right pixels with a specific color that is indicative of the final label, as detailed in Table 7. This experiment is detailed in Appendix E.1. Visual examples are shown in Fig. 4.

- `SST2` is a sentiment analysis dataset, from Yuan et al. (2022). The task requires predicting the sentiment polarity of tweets, which are either labeled as negative or positive. Node features are contextual embeddings from a pretrained language model (Yuan et al., 2022).

- `SST2P` (ours) is an extension of `SST2P`, where we force each input sample to contain a single "," and a single "." in the input, such that they are not informative of the final label. To make sure that "," and "." are not correlated with the label, we add them as isolated nodes in the graph. If the graph originally contained such a punctuation, we first remove it. To further avoid the contextual embedding from carrying over some information regarding emotion-laden words, we use fixed embeddings for "," and ".", respectively set to $\vec{1}$ and $\vec{0}$.

### D.4.1 DESIGNATED EXPLANATIONS FOR SECTION 4

In our experiments, we define the following designated explanations $p_u^y$: For `MNISTsp`, these explanations highlight background pixels, similarly to Jethani et al. (2021); For `MUTAG`, these highlight hydrogen (H) and carbon (C) atoms, and plot in Fig. 10 the relative frequencies of these atoms, showing they are not informative of the label; For `SST2P`, the designated explanations highlight "," and ".", similarly to Yu et al. (2019); For `RBGV`, instead, these highlight green and violet nodes, which are clearly uninformative of the underlying task to predict. We report in Table 5 the mapping between classes and designated explanations.

Table 5: Designated explanations are defined as nodes that are unrelated to the task being solved. Yet, in Section 4 we show that SE-GNNs can extract these explanations while accurately solving the task. "H" and "C" stand for hydrogen and carbon atoms, respectively.

| Dataset | Task to predict | Designated Expl. $p_u^y$ |
|---------|-----------------|--------------------------|
| RBGV | If blue $>$ red nodes | Green node for $y = 1$, violet node for $y = 0$ |
| MNISTsp | Digit number | Top-left and bottom-right $y$-th pixel |
| MUTAG | Mutagenicity of molecules | H atoms for $y = 0$, C atoms for $y = 1$ |
| SST2P | Tweet sentiment polarity | ',' for $y = 0$, '.' for $y = 1$ |

### D.5 THRESHOLDING EXPLANATIONS

As discussed in Section 2, SE-GNNs output relevance scores in the interval $[0, 1]$. Then, to extract a discrete subgraph that users can consume, we need to define a selection criterion to decide which nodes are to be considered part of the explanation, and which are not. Following Tai et al. (2025), and in accordance with the semantics each SE-GNN gives to explanation relevance scores it predicts, we define some threshold values for SMGNN and GSAT, as follows.

**For naturally trained models**, we fix the threshold to $0.5$ for SMGNN as Tai et al. (2025), whereas for GSAT we set it to $0.9$. This is done because GSAT's relevance scores are associated with the probability of sampling each node/edge, hence we consider as relevant only those that can be consistently sampled with high probability. The only exception is GSAT trained for RBGV, where the threshold is again set to $0.5$. This is because for RBGV, GSAT is trained to push the sampling probability around $0.3$ (by setting $r = 0.3$), and therefore a probability of $0.5$ is already consistently higher than $0.3$. Occasionally, SMGNN's relevance scores all collapse to values around $0$, or to values such that no node has a value greater than $0.5$. To avoid manually defining case-by-case thresholds when this happens, we apply an instance-wise min-max normalization while sticking to the $0.5$ threshold. In particular, this happened for MNISTsp and MUTAG. For DIR, instead, we keep the topK selection used during training.

**For attacked models**, the threshold is fixed to $0.5$ as SE-GNNs are explicitly trained to extract binary relevance scores. This also applies to DIR, for which we remove nodes inside the topK subgraph having a score less than the threshold. This is because nodes with a score close to $0$ should still be considered irrelevant, even if in the topK.

### D.6 FAITHFULNESS METRICS

Each faithfulness metric defines a set of allowed perturbations indicated with $\mathcal{I}$. Perturbations are either applied to the complement – for *sufficiency* – or to the explanation – for *necessity* metrics. Below, we detail the implementation details for the metrics we empirically tested in Section 5. In all cases, the explanation selection mechanism is fixed and detailed in Appendix D.5.

Fid-: $\mathcal{I}$ is simply defined as the perturbation removing the complement of the explanation from the graph, and feeding the explanation alone back to the model.

Fid+: $\mathcal{I}$ is simply defined as the perturbation removing the explanation from the graph, and feeding the complement of the explanation alone back to the model.

RFid-: $\mathcal{I}$ is defined as random edge removals from the complement, where each edge is removed independently from the others and with a fixed probability. Following Zheng et al. (2023), we fix the probability to $0.9$. Nodes are left untouched.

**RFid+:** $\mathcal{I}$ is defined as random edge removals from the explanation, where each edge is removed independently from the others and with a fixed probability. Following Zheng et al. (2023), we fix the probability to 0.1. Nodes are left untouched.

**Suf:** $\mathcal{I}$ is defined as complement swaps. In particular, given the reference graph for which we compute the metric, we pick a random sample from the same split and with the same label, and we swap their complements. The explanation is randomly attached to the new complement while preserving the total number of edges of the original graph.

**Nec:** $\mathcal{I}$ is defined as random edge removals from the explanation, where each edge is removed independently from the others, and the overall budget of removal is defined dataset-wise and kept fixed for each sample. Following Azzolin et al. (2025a), we fix the budget as 10% of the average graph size of the split of data where the metric is computed. Nodes are left untouched.

**CF:** $\mathcal{I}$ perturbs only the relevance scores, where the perturbation is randomly sampled from a Gaussian distribution with mean and variance estimated instance-wise, and applied to scores inside and outside of the explanation. Therefore, the topology of the perturbed graph is the same as the original one, with perturbed relevance scores. The perturbed graph is then fed to the SE-GNN's classifier $g$ only. The expected semantics is that the classifier should be highly responsive to perturbations. Hence, large changes in the output should be expected for faithful explanations.

**EST:** $\mathcal{I}$ is defined as any possible supergraph of the explanation inside the input graph. Equivalently, given the input graph and its explanations, $\mathcal{I}$ randomly samples perturbations removing nodes and edges, jointly. This can be seen as a more general version of RFid-, where nodes can also be removed.

---

**Algorithm 1** Uniform Sufficiency Test (EST, Definition 1)

**Require:** Input graph $G$, explanation $R$, budget of perturbations $b$, distance function $d$
$\quad\quad \mathcal{S} = []$
1: **for** $i = 1$ to $b$ **do**
2: $\quad$ // Subsample nodes
3: $\quad$ Sample random node weights $\mathbf{w} \sim \text{Unif}(0,1)^{|V(G)|}$
4: $\quad$ Force $\mathbf{w}[v] \leftarrow 1$ for all $v \in V(R)$
5: $\quad$ Define node set $V_i = \{v \in V(G) \mid \mathbf{w}[v] \geq 0.5\}$
6: $\quad$ Construct node-induced subgraph $G_i = G[V_i]$
7:
8: $\quad$ // Subsample edges
9: $\quad$ Sample random edge weights $\mathbf{w} \sim \text{Unif}(0,1)^{|E(G_i)|}$
10: $\quad$ Force $\mathbf{w}[e] \leftarrow 1$ for all $e \in E(R)$
11: $\quad$ Define edge set $E_i = \{e \in E(G_i) \mid \mathbf{w}[e] \geq 0.5\}$
12: $\quad$ Construct edge-induced subgraph $G_i' = G_i[E_i]$
13: $\quad$ Append $G_i'$ to $\mathcal{S}$
14: **end for**
15: **return** $\max_{G' \in \mathcal{S}} d(f(G), f(G'))$

---

**Remark:** Contrary to previous faithfulness metrics, which typically average the contribution from multiple perturbations, EST adopts a worst-case evaluation by taking the maximum change in prediction across perturbations. This stops frequent but irrelevant perturbations from diluting rare but impactful perturbations.

## E ADDITIONAL EXPERIMENTS

### E.1 PLAUSIBLE BUT UNFAITHFUL EXPLANATIONS

In this experiment, we aim to show that it is possible to induce SE-GNNs to output highly plausible but unfaithful explanations. We achieve this by attacking the model following the setting outlined in Section 4, i.e., adding an explicit supervisory signal to control the explanations provided by the explanation extractor. In this case, we supervise the SE-GNN to output only nodes belonging to

Table 6: The Table reports the actual metric value computed with $d$ being Total variation – hence computed without the proxy of $\mathsf{RejRatio}_{\mathcal{I}}$ – for the same experiments as Table 3. The results confirm that EST consistently assigns worse values to unfaithful explanations. Nonetheless, continuous scores make it harder to assess the overall behavior of the model, and when to consider an explanation as unfaithful. ↓ (↑) stands for the lower (the higher) the better.

| Dataset | Model | Fid- ↓ | Fid+ ↑ | Suf ↓ | Nec ↑ | CF ↑ | RFid- ↓ | RFid+ ↑ | EST ↓ |
|---|---|---|---|---|---|---|---|---|---|
| RBGV | SMGNN | 16 ±07 | 44 ±03 | 00 ±00 | - | 02 ±01 | 00 ±00 | - | 47 ±02 |
| | GSAT | 07 ±10 | 50 ±03 | 01 ±00 | - | 49 ±05 | 06 ±06 | - | 46 ±20 |
| | DIR | 20 ±08 | 42 ±06 | 00 ±00 | - | 37 ±10 | 03 ±02 | - | 47 ±03 |
| MNISTsp | SMGNN | 89 ±02 | 49 ±05 | 44 ±05 | 49 ±06 | 89 ±02 | 86 ±02 | 02 ±00 | 99 ±00 |
| | GSAT | 68 ±10 | 61 ±04 | 45 ±06 | 44 ±05 | 90 ±01 | 77 ±03 | 02 ±00 | 99 ±00 |
| | DIR | 88 ±01 | 54 ±06 | 50 ±07 | 52 ±04 | 88 ±01 | 86 ±02 | 03 ±01 | 98 ±01 |
| MUTAG | SMGNN | 33 ±02 | 69 ±04 | 31 ±03 | 13 ±03 | 46 ±04 | 43 ±04 | 06 ±01 | 95 ±01 |
| | GSAT | 13 ±10 | 79 ±09 | 39 ±04 | 17 ±04 | 47 ±09 | 44 ±10 | 08 ±01 | 90 ±05 |
| | DIR | 38 ±06 | 70 ±08 | 35 ±04 | 15 ±02 | 53 ±04 | 49 ±05 | 07 ±01 | 93 ±03 |
| SST2P | SMGNN | 08 ±02 | 41 ±04 | 01 ±00 | - | 21 ±07 | 02 ±01 | - | 52 ±04 |
| | GSAT | 25 ±13 | 45 ±04 | 01 ±00 | - | 31 ±17 | 08 ±02 | - | 58 ±11 |
| | DIR | 26 ±01 | 49 ±01 | 01 ±00 | - | 40 ±12 | 02 ±00 | - | 46 ±03 |

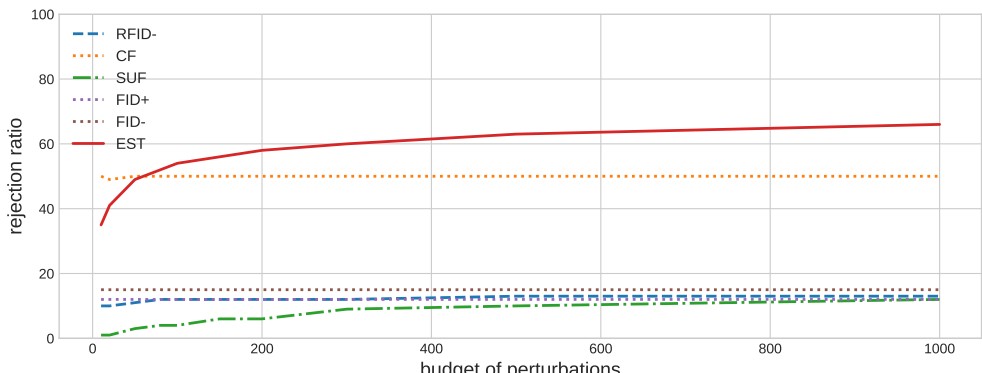

Figure 2: Ablation study on the budget of perturbations to estimate Eq. (4) for GSAT on RBGV (cf. Section 5.2). The plot shows that EST steadily rejects more unfaithful explanations as the budget increases, while other metrics do not substantially improve. Fid- (Fid+) have constant values, a they do not apply any sampling but just erase the complement (the explanation) once. Note that EST samples subgraphs of the complement at random, and thus, the probability of sampling duplicates increases as the budget increases. Future work can investigate more complex sampling strategies to avoid such duplicates.

Table 7: Mapping between colors applied to CPatchMNIST and class label.

| Color | Class label |
|---|---|
| 🔴 | 0 |
| 🟠 | 1 |
| 🟡 | 2 |
| 🟢 | 3 |
| 🔵 | 4 |
| 🟣 | 5 |
| ⚫ | 6 |
| | 7 |
| 🌸 | 8 |
| 🔵 | 9 |

the digit together with its 1-hop neighborhood, which is typically considered to be the ground truth explanation for MNISTsp (Miao et al., 2022a). To simulate the presence of protected attributes, we

color the top-left and bottom-right pixels with a specific color that is indicative of the final label, as detailed in Table 7. The resulting dataset will be referred to as CPatchMNIST. Therefore, for the attack to be successful, we expect the explanation to highlight only nodes pertaining to the digit itself while concealing the use of the protected attribute.

We report in Table 8 the results of the attack, which confirm that it is possible to train SE-GNNs to output highly human-desirable explanations – in particular those matching the human expectations – whilst the model is also relying crucially on other – potentially protected – features. We provide some visual examples of the attack in Fig. 3, and the predicted label for graphs with swapped colors in Fig. 4.

Table 8: Accuracy and test accuracy after either swapping the color of the colored pixels with that of other classes (see Fig. 4), or blacking out the digit, for the experiment in Appendix E.1. Results show that both models are highly dependent on both the digit-related superpixels and the color ones, despite only the former being declared in the explanation.

| Model | Val Acc | Val $F_1$ | Acc color swap | Acc no digit |
|---|---|---|---|---|
| SMGNN | $100.0 _{\pm 0.0}$ | $99.3 _{\pm 0.3}$ | $0.1 _{\pm 0.2}$ | $9.6 _{\pm 0.0}$ |
| GSAT | $99.9 _{\pm 0.1}$ | $99.1 _{\pm 0.1}$ | $3.4 _{\pm 4.2}$ | $8.7 _{\pm 1.9}$ |

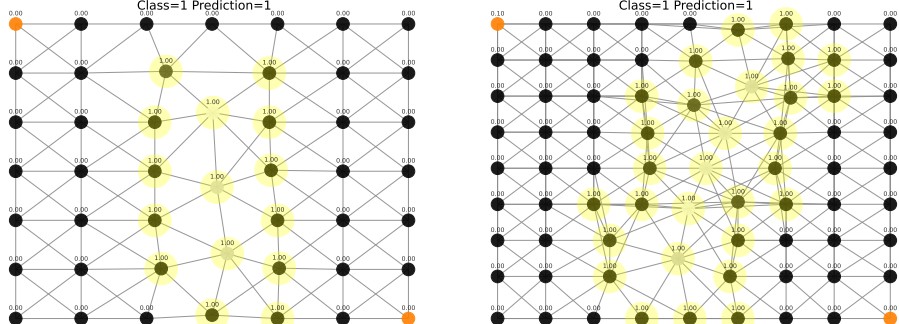

Figure 3: **SE-GNNs can be manipulated to output unfaithful but highly plausible explanations.** We train GSAT (left) and SMGNN (right) to maximize the downstream accuracy while also optimizing for plausibility, that is, outputting the digit and its 1-hop neighbor as an explanation. Nonetheless, both models are heavily relying on the color information, as shown by the severe drop in accuracy when the color information is altered (shown in Table 8), and by the arbitrary manipulation of the predicted label by swapping colors (shown in Fig. 4). The underlying dataset is an extension of MNISTsp, called CPatchMNIST, in which we color the upper-right and bottom-left superpixels with class-discriminative colors that are treated as protected attributes. Details about the dataset and the color coding are in Appendix E.1. The plot shows two graph examples of class 1, where nodes belonging to the explanation are selected based on a threshold value on the relevance score at $0.5$. Numbers above each node represent raw explanation relevance scores. Better seen in digital format.

**Can faithfulness metrics spot plausible but unfaithful explanations?** Similar to the analysis presented in Section 5.1, we investigate whether popular faithfulness metrics can mark these explanations as unfaithful. We adopt the same setting as Section 5.1, and compute rejection ratios for our proposed EST, and for Fid-, and RFid- as baselines. The results in Table 9 show that RFid- can catastrophically fail to mark these explanations as unfaithful. Conversely, Fid- and EST can robustly reject a large number of explanations, albeit with consistent variations across random seeds. Overall, EST achieves the highest ratios in both experiments.

### E.2 EXAMPLES OF ATTACKED SE-GNNS

Taking as an example the GSAT model, we provide for each dataset in Table 2 a few examples of graphs together with the highlighted explanations to showcase the efficacy of the attack. In particular,

Table 9: We report the rejection ratios RejRatio$_\mathcal{I}$ for the plausible but unfaithful explanations discussed in Appendix E.1. EST and Fid- robustly reject a large number of explanations as unfaithful, whereas RFid- fails to mark them as such.

| Dataset | Model | RejRatio$_\mathcal{I}$ | | |
|---------|-------|------|-------|-----|
| | | Fid- | RFid- | EST |
| CPatchMNIST | SMGNN | **50.7** $_{\pm 25.3}$ | $0.0$ $_{\pm 0.0}$ | **50.7** $_{\pm 25.3}$ |
| | GSAT | $\underline{45.3}$ $_{\pm 6.1}$ | $15.7$ $_{\pm 11.6}$ | **54.0** $_{\pm 10.7}$ |

Fig. 5, Fig. 6, Fig. 7, and Fig. 8 shows such examples for RBGV, MNISTsp, MUTAG, and SST2P respectively.

### E.3 FAILURES OF SE-GNN ATTACKS

Taking as an example the GSAT model on MNISTsp, which achieved a slightly lower $F_1$ score in Table 2 compared to other configurations, we provide in Fig. 9 some examples of graphs where the attacked GSAT does not provide the exact ground truth we aim for. This can emerge for several reasons, like the indistinguishability of node embeddings after $L$ layers of message passing (i.e., oversmoothing), or because the explanation extractor failed to infer the correct graph label. Nonetheless, the examples show that even if explanations do not perfectly highlight the ground truth provided in Table 5, they are still confidently highlighting unrelated pixels while hiding the true reason behind the prediction, which is the main goal of our attack.

We further provide in Fig. 21 several examples of explanations for the test split of SMGNN on SST2P, which achieves a low attack $F_1$ score in Table 2 due to OOD samples. Also in this case, although the attack may not generalize well to OOD samples, it still consistently suppresses the relevance score of truly informative tokens, while highlighting degenerate uninformative subgraphs as the most important ones.

### E.4 REJECTION RATIOS FOR SUFFICIENT EXPLANATIONS

In this section, we verify that faithfulness metrics not only reject unfaithful explanations but also recognize faithful ones. Following the attack framework in Section 4, we manipulate SE-GNNs to output explanations that, by construction, include all information the model can use to solve the task. These designated explanations enforce *sufficiency*, though at the cost of potentially including irrelevant features. While enlarging explanations is a trivial way to ensure *sufficiency* (Zheng et al., 2023; Azzolin et al., 2025a) – and does not necessarily yield truly faithful explanations (Tai et al., 2025) – this experiment serves as a sanity check to confirm that *sufficiency* metrics at least recognize *sufficient* explanations. Accordingly, *we expect rejection ratios close to zero across all configurations.*

**Experimental setup:** To ensure the explanation contains all the relevant information the model can use to make the final prediction, we restrict our experiments to RBGV and MNISTsp, where we have full knowledge of how to solve the task. In particular, we train each SE-GNNs with Eq. (3), where $\mathcal{L}_{expl}$ is defined so to induce the following explanations: For RBGV, we extract all red nodes for the negative class, and all blue nodes for the positive class, i.e., $p_u^y = 1$ for any red node in a negative graph, $p_u^y = 1$ for any blue node in a positive graph, and $p_u^y = 0$ otherwise.[5] For MNISTsp, we extract each pixel associated with the digit and its 1-hop neighborhood, i.e., $p_u^y = 1$ for any node $u$ belonging to the digit $y$ itself or to its 1-hop neighborhood, and $p_u^y = 0$ otherwise. Note that these explanations are, in fact, constructed to contain all the features that are known to be related to the task at hand.

The results are shown in Table 10. The final accuracy and the explanation $F_1$ score confirm that the manipulation was successful, and that models are making accurate predictions while outputting our benignly designated explanations. Across both datasets, the only metrics that consistently achieve a rejection ratio around zero are Fid-, RFid-, and EST, with fluctuations in around one case each.

---

[5]For instance, given a graph with three red nodes and two blue nodes, highlighting the three red nodes as the explanation ensures that no perturbation on nodes of the complement can change the class of the graph. Hence, the explanation is *sufficient*.

Among them, EST was also the one achieving the best results in Table 3, confirming its reliability. Suf and CF, instead, reject a high amount of explanations, suggesting that the high rejection ratios obtained in Table 3 were probably conflated by a too aggressive metric.

Table 10: Accuracy, explanation $F_1$ score, and rejection ratios for different metrics computed for models trained to output explanations containing all the relevant features models can use. The experimental setting is described in Appendix E.4 and constitutes a sanity check to verify that metrics with high rejection ratios in Table 3 are not simply rejecting *any* explanation, but are indeed verifying whether the explanation does not include some relevant pattern used by the model.

| Dataset | Model | Test Acc | Test $F_1$ score | Rej_ratio$_\mathcal{I}$ | | | | |
|---|---|---|---|---|---|---|---|---|
| | | | | Fid- | Suf | CF | RFid- | EST |
| RBGV | SMGNN | $100_{\pm 00}$ | $98_{\pm 01}$ | $01_{\pm 01}$ | $76_{\pm 02}$ | $14_{\pm 02}$ | $00_{\pm 00}$ | $01_{\pm 01}$ |
| | GSAT | $100_{\pm 00}$ | $99_{\pm 01}$ | $01_{\pm 01}$ | $78_{\pm 02}$ | $13_{\pm 02}$ | $00_{\pm 00}$ | $01_{\pm 01}$ |
| | DIR | $100_{\pm 00}$ | $98_{\pm 01}$ | $01_{\pm 01}$ | $93_{\pm 01}$ | $57_{\pm 15}$ | $01_{\pm 01}$ | $01_{\pm 01}$ |
| MNISTsp | SMGNN | $90_{\pm 01}$ | $99_{\pm 01}$ | $01_{\pm 01}$ | $75_{\pm 04}$ | $93_{\pm 03}$ | $01_{\pm 01}$ | $01_{\pm 01}$ |
| | GSAT | $93_{\pm 01}$ | $99_{\pm 01}$ | $01_{\pm 01}$ | $35_{\pm 04}$ | $07_{\pm 04}$ | $15_{\pm 18}$ | $05_{\pm 05}$ |
| | DIR | $89_{\pm 01}$ | $99_{\pm 01}$ | $22_{\pm 05}$ | $86_{\pm 03}$ | $93_{\pm 03}$ | $01_{\pm 01}$ | $07_{\pm 02}$ |

## F  ADDITIONAL DISCUSSION

### F.1  EVALUATING *Sufficiency* AND *Necessity* JOINTLY

*Sufficiency* and *necessity* metrics are often evaluated in conjunction, so that an explanation is considered faithful only if it is at the same time both sufficient *and* necessary (Amara et al., 2022; Longa et al., 2024). The remark on previous metrics provided in Section 5.2 generalizes to this case.

Consider the same setting as Example 1 for a negative instance whose explanation is $R = u_{green}$. As previously observed, R is the smallest label-preserving subgraph, as $0 \geq 0$. Then, any Complement and Explanation removal metric, such as Fid- and Fid+, will mark such an explanation as sufficient – as R is label-preserving – and necessary – as removing the green node will make the classifier in Eq. (2) outputting maximally uncertain predictions.

### F.2  ARE NECESSITY METRICS USEFUL FOR SE-GNNS?

In this section, we aim at answering the question of whether metrics estimating the *necessity* of explanations are useful for finding the failure cases of SE-GNNs outlined in Section 3. Let us consider the following representative SE-GNN outputting unfaithful explanations for any arbitrary binary classification task, represented by the boolean classifier $\psi$:

$$e(G) = \begin{cases} u_0 & \text{if } \neg\psi(G) \\ u_1 & \text{if } \psi(G) \end{cases} \qquad g(R) = \begin{cases} 0 & \text{if } R = u_0 \\ 1 & \text{if } R = u_1 \\ 0.5 & \text{otherwise} \end{cases} \qquad (21)$$

where $g$ acts as a simple mapping from explanations to labels, and $u_0$ and $u_1$ can either be label or non-label-preserving subgraphs of the input sample $G$, and are assumed to be non-representative of the true behavior of the model by construction. Note that Eq. (21) matches Example 1 whenever $\psi$ corresponds to $\#blue > \#red$, $u_0 = u_{green}$, and $u_1 = u_{violet}$. Then, any *necessity* metric would apply perturbations to $R$ to induce some shifts in the classifier's output. If such a perturbation succeeds in changing the prediction of the model, the explanation is marked as *necessary*, and thus faithful. Nevertheless, $u_0$ and $u_1$ are in fact relevant for the classifier $g$, which trivially maps $u_0$ and $u_1$ to class labels. This means that removing them will always bring a change in the output class, regardless of how unfaithful $u_0$ and $u_1$ actually are.

### F.3  THEORETICAL ANALYSIS OF EST

We aim to theoretically characterize which family of explanations EST can mark as unfaithful. We do so by relating Definition 1 with the following three formal notions of explanations:

**Definition 3** (Prime Implicant explanation (Azzolin et al., 2025b)). *Let $f$ be a classifier and $G$ be an instance with predicted label $f(G)$, then $R$ is a Prime Implicant explanation for $f(G)$ if:*

1. *$R \subseteq G$.*

2. *For all $R'$, such that $R \subseteq R' \subseteq G$, we have that $f(G) = f(R')$.*

3. *No other $R' \subset R$ satisfies both (1) and (2).*

Prime Implicant explanations have been widely studied in the formal explainability literature, and are considered of *high-quality* as they possess several desirable properties (Marques-Silva & Ignatiev, 2022; Azzolin et al., 2025b; Bassan et al., 2025a;b). In a nutshell, Prime Implicant explanations comprise minimal-sufficient explanations, which are the minimal explanations provably robust to changes in the complement. A related notion of explanations is Minimal explanation, introduced next:

**Definition 4** (Minimal explanation (Azzolin et al., 2025b)). *Let $f$ be a SE-GNN and $G$ be an instance with predicted label $f(G)$, then $R \subseteq G$ is a Minimal explanation for $f(G)$ if:*

1. *$f(G) = f(R)$*

2. *There exists no $R' \subseteq G$ such that $|R'| < |R|$ and $f(G) = f(R')$.*

Minimal explanations have been shown to equal Prime Implicant ones for subgraph-based tasks, but in the other cases they are deemed as *less informative* and potentially ambiguous (see Figure 1 in Azzolin et al. (2025b)). An even weaker notion of explanations is introduced next:

**Definition 5** (Non-label-preserving explanation). *Let $f$ be a SE-GNN and $G$ be an instance with predicted label $f(G)$, then $R \subseteq G$ is a Non-label-preserving explanation for $f(G)$ if $f(G) \neq f(R)$.*

This notion of explanation is clearly weaker than the previous two, as it embraces explanations that do not contain enough information to yield the same prediction as the original sample – meaning they are omitting key relevant features – and allow *any* subgraph to be selected as the explanation. An example of this failure case is given in Example 1: the subgraph highlighted by $R = u_{violet}$ is a non-label preserving subgraph, as despite explaining the positive class, the subgraph has an equal count of reds and blues, yielding the model to predict the negative class. A similar example can also be found for Minimal explanations: the subgraph highlighted by $R = u_{green}$ is a Minimal explanation, as there exists no smaller explanation, and feeding Eq. (2) with it will yield the same prediction (as $0 \geq 0$). Yet, $u_{green}$ is never used by $e$ to infer the label, yielding a misleading explanation that can misguide model debugging (Teso et al., 2023) and scientific discovery (Wong et al., 2024) by falsely suggesting green nodes are linked to the negative class. An additional example is also highlighted in Section 3 of Azzolin et al. (2025b), where Minimal explanations are shown to highlight counter-intuitive subgraphs that do not robustly indicate the model's behavior. In light of the above considerations, Minimal and Non-label-preserving should be regarded as notions that admit degenerate explanations.

We proceed to show how EST fares at marking the previous notions. In the following analysis, we will consider EST fully enumerating all possible extensions in Eq. (5), and consider a perturbation to yield a sufficiently large prediction shift whenever it changes the predicted class.

**Theorem 3.** EST *marks Minimal and Non-label-preserving explanations as unfaithful and Prime-Implicant explanations as faithful.*

*Proof.* We will analyze each explanation family of explanations separately:

1. **Prime Implicant explanations:** If the explanation $R$ is a Prime Implicant explanation, by definition, it holds that:

$$g(e(G)) = g(e(R')) \quad \forall R \subseteq R' \subseteq G. \tag{22}$$

Hence, it follows trivially that Eq. (5) will never encounter a prediction change, hence will mark the explanation as faithful.

2. **Minimal explanations:** We show that if an explanation $R$ fulfills EST (i.e., Eq. (5) equals zero for any $R'$ such that $R \subseteq R' \subseteq G$) and is therefore considered to be faithful, then $R$

can be a Minimal explanation only if it is also a Prime Implicant. In other words, if an explanation passes the EST test, it cannot be *just* a Minimal explanation.

Let us consider an explanation $R$ that passes the EST test and is a Minimal explanation. Then, it holds that:

1. $R \subseteq G$.
2. $g(e(G)) = g(e(R'))$ $\quad \forall R \subseteq R' \subseteq G$.
3. $\nexists R' \subseteq G$ such that $|R'| < |R|$ and $g(e(G)) = g(e(R'))$.

Note that condition 3 implies that no other $R' \subset R$ satisfies both conditions 1 and 2. This is because being label-preserving is a necessary condition for condition 2 to be true. However, condition 3 ensures that there does not exist any smaller label-preserving subgraph than $R$, hence no smaller explanation satisfying condition 2 exists. Therefore, $R$ is the smallest subgraph that satisfies condition 2, hence it is a Prime Implicant explanation.

3. **Non-label-preserving explanation:** Non-label-preserving explanations are, by definition, non-label-preserving subgraphs. Since EST samples perturbed graphs $R \subseteq R' \subseteq G$ uniformly, it can always sample $R' = R$. Hence, evaluating Eq. (5) with $R' = R$ will bring a prediction change, which marks the explanation as unfaithful.

$\square$

In words, this result shows that EST can flag Minimal and Non-label-preserving explanations as unfaithful. By definition, these explanations allow the model's prediction to be altered by changes outside the explanation, leaving room for unfaithful or even malicious behaviors. In contrast, Prime Implicant explanations do not permit prediction changes under complement perturbations, making them more robust. An interesting direction for future work is to investigate whether Prime Implicant explanations themselves can still admit degenerate cases.

## G  FIGURES FROM TABLE 4

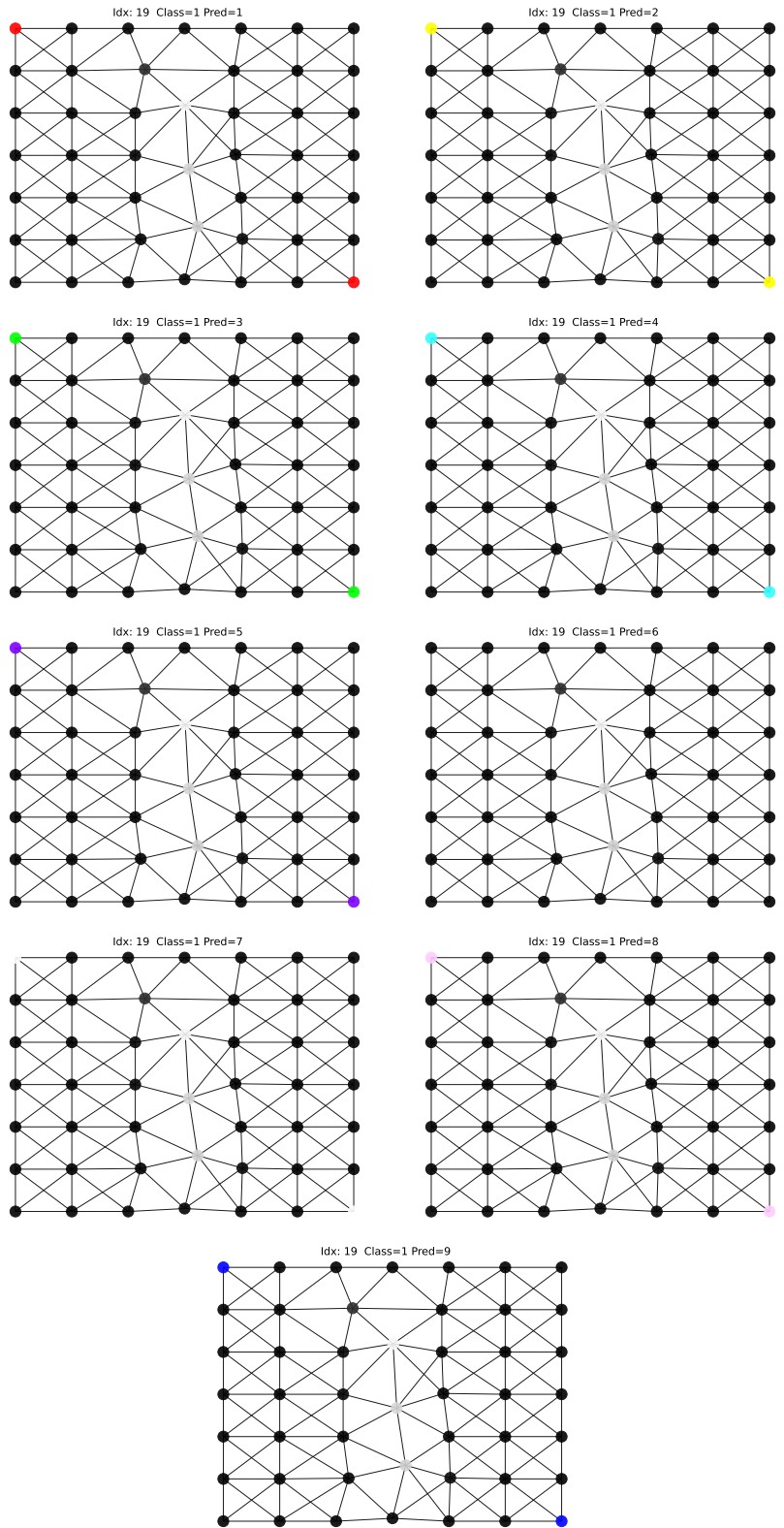

Figure 4: Despite GSAT's explanation declaring that only the digit information is used for the graph in Fig. 3, swapping the color to that of other classes, following the schema of Table 7, results in arbitrary manipulation of the final prediction.

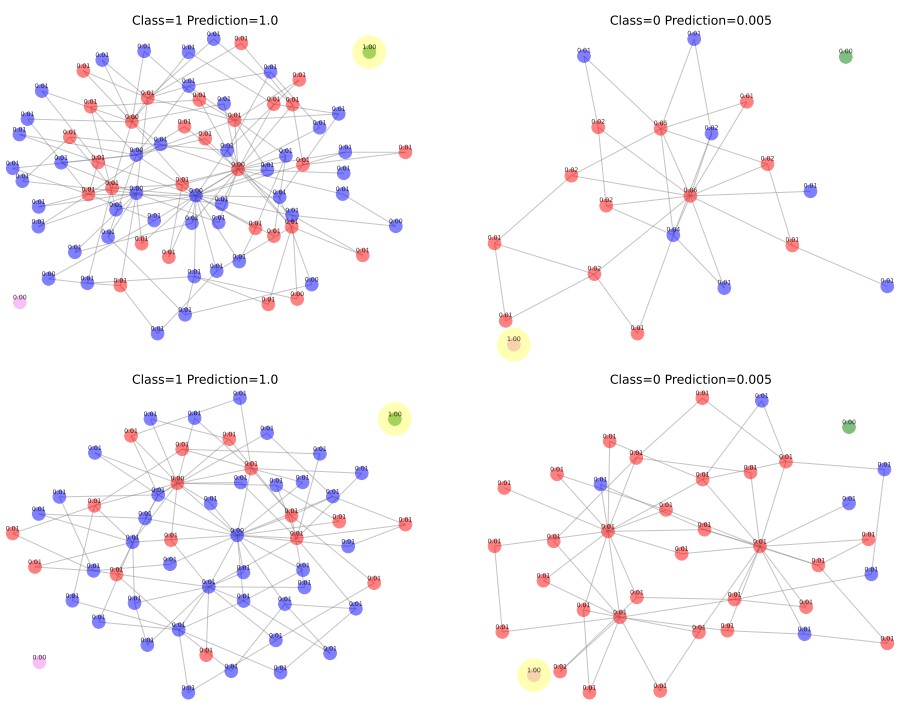

Figure 5: Explanations provided by the attacked `GSAT` on `RBGV`. Numbers above each node represent raw explanation relevance scores, and explanatory nodes are selected based on a $0.5$ threshold. Better seen in digital format.

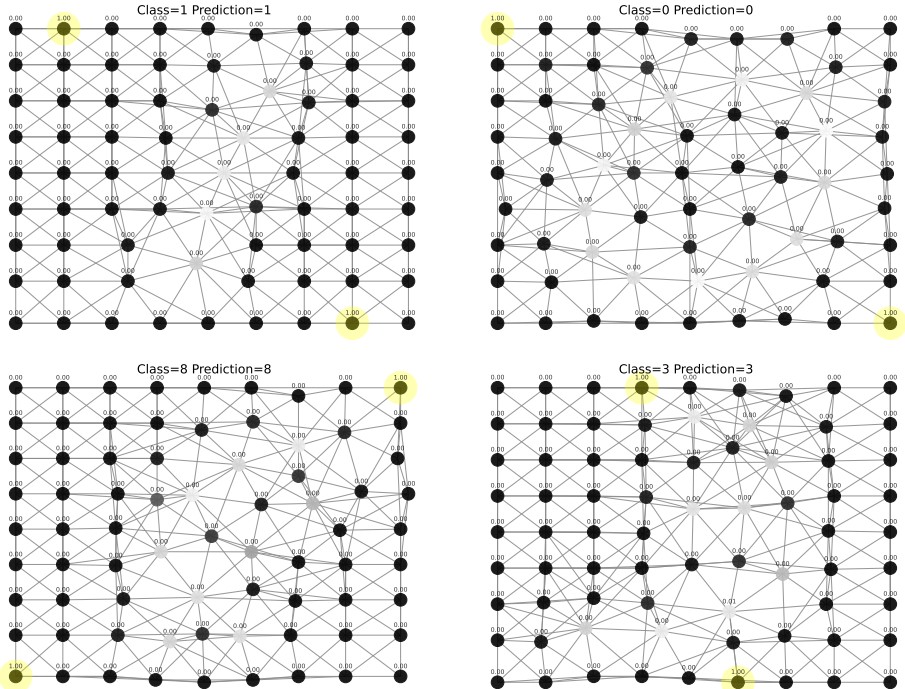

Figure 6: Explanations provided by the attacked `GSAT` on `MNISTsp`. Numbers above each node represent raw explanation relevance scores, and explanatory nodes are selected based on a $0.5$ threshold. Better seen in digital format.

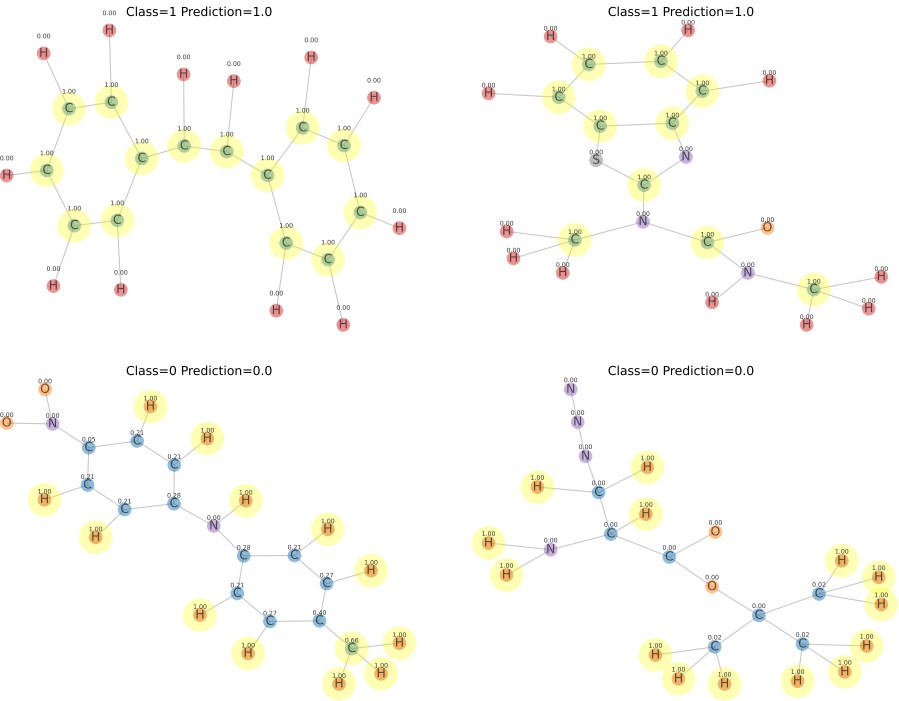

Figure 7: Explanations provided by the attacked GSAT on MUTAG. Numbers above each node represent raw explanation relevance scores, and explanatory nodes are selected based on a 0.5 threshold. Better seen in digital format.

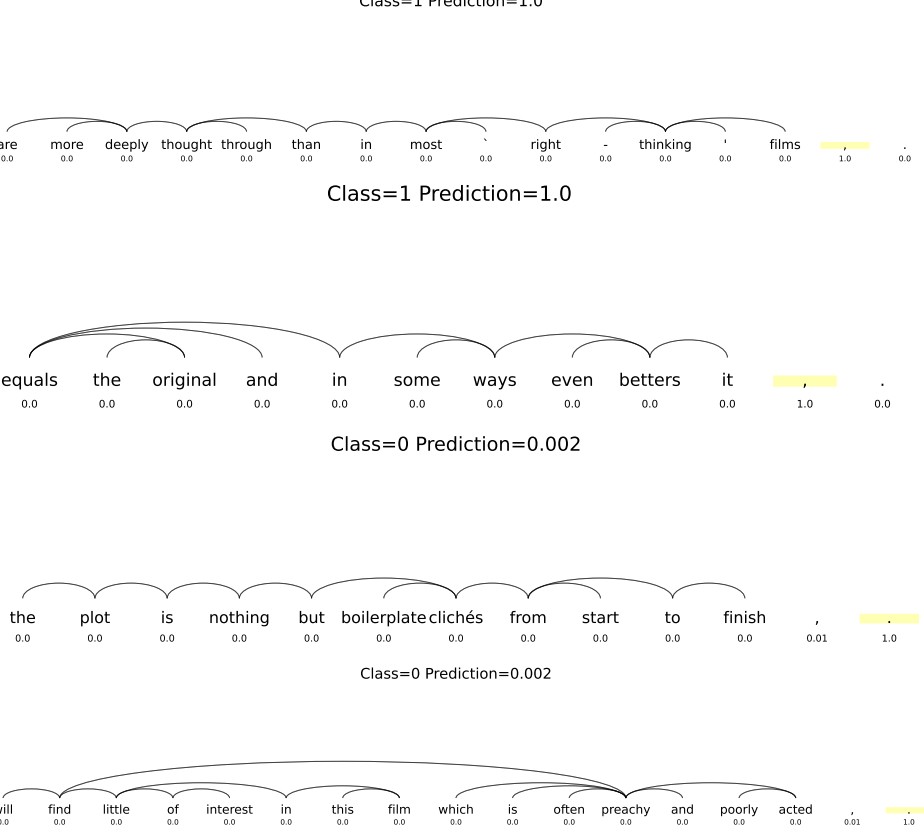

Figure 8: Explanations provided by the attacked `GSAT` on `SST2P`. Numbers below each node represent raw explanation relevance scores, and explanatory nodes are selected based on a $0.5$ threshold. Better seen in digital format.

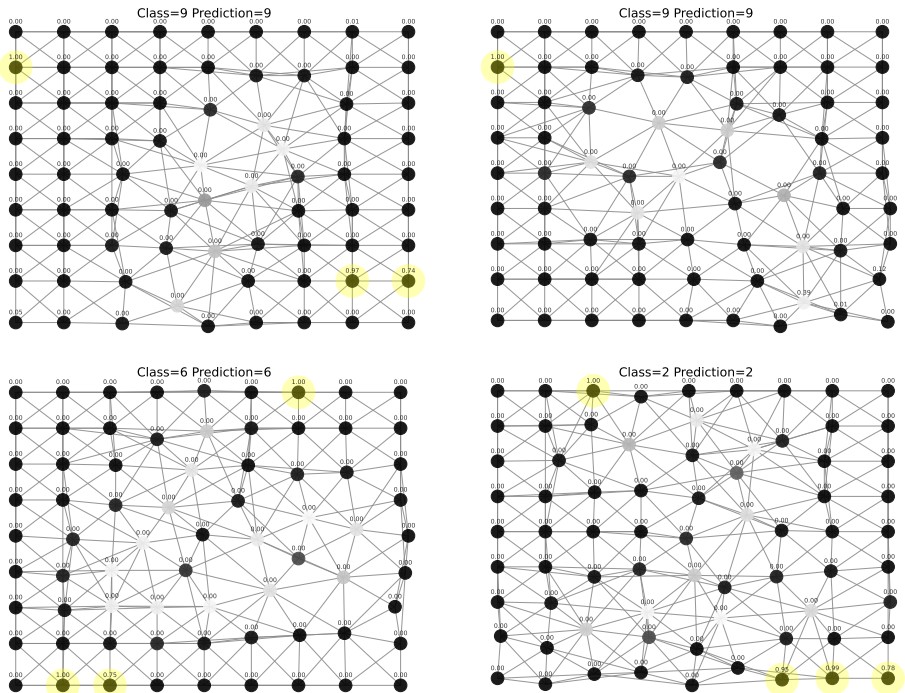

Figure 9: Explanations provided by the attacked `GSAT` on `MNISTsp`. Even if explanations do not perfectly highlight the ground truth provided in Table 5, they still hide the true reason behind the prediction, i.e., the digit in the image, while confidently highlighting apparently unrelated pixels. Numbers above each node represent raw explanation relevance scores, and explanatory nodes are selected based on a $0.5$ threshold. Better seen in digital format.

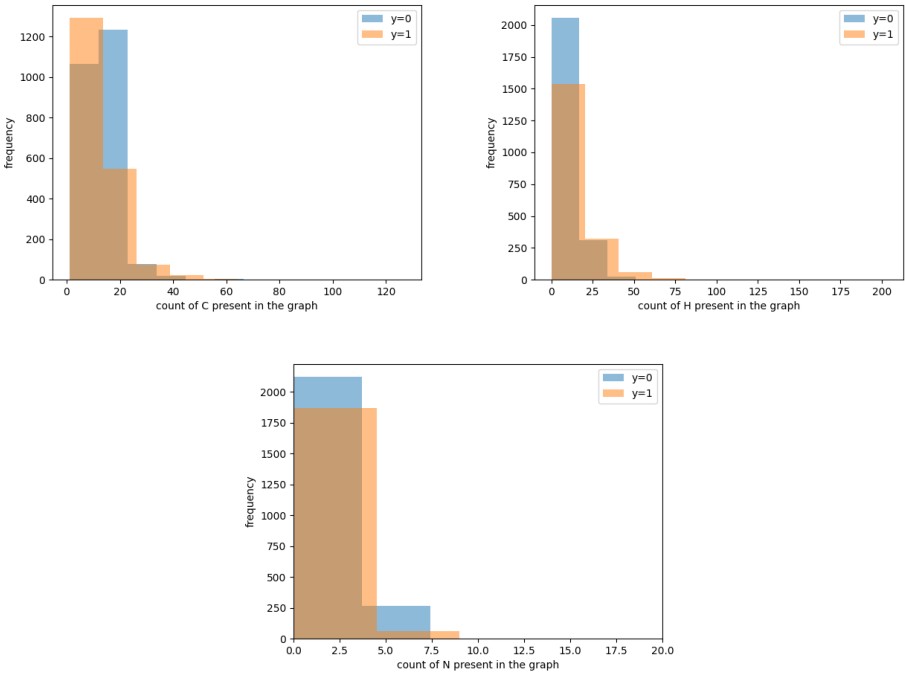

Figure 10: Frequencies of appearance of carbon (C), nitrogen (N), and hydrogen (H) atoms in `MUTAG`, divided per class.

Figure 11: Diagram representing the distribution of explanation relevance score for seed 1 of `GSAT` trained on `RBGV` according to the setup described in Section 6. The four pairs of plots on the left report the distribution of relevance scores, divided by class label and node color (Red, Blue, Green, and Violet), while the last pair on the right reports the boxplot (colored bars) and average count (green triangle) of each color present in the explanation of each sample. Nodes included in the explanation are chosen as those with a score greater than $0.5$. Overall, `GSAT` roughly picks a single red node for $y = 0$ and a single blue node for $y = 1$, corresponding to the smallest label-preserving explanations. When computing rejection ratios for faithfulness metrics, we respectively get a rejection ratio of $58.7\%$ for EST, $0.9\%$ for Fid-, and $0\%$ for RFid-. Although red and blue nodes seem a legitimate explanation for this task, highlighting only one of them is nonetheless still concealing the fact that the model needs to look at all of them to make a correct prediction. Therefore, the low rejection ratios of Fid- and RFid- are considered to be failure cases, as they flagged explanations not containing every element the model is actually using to infer the prediction as faithful. EST, instead, correctly rejects a considerable number of explanations.

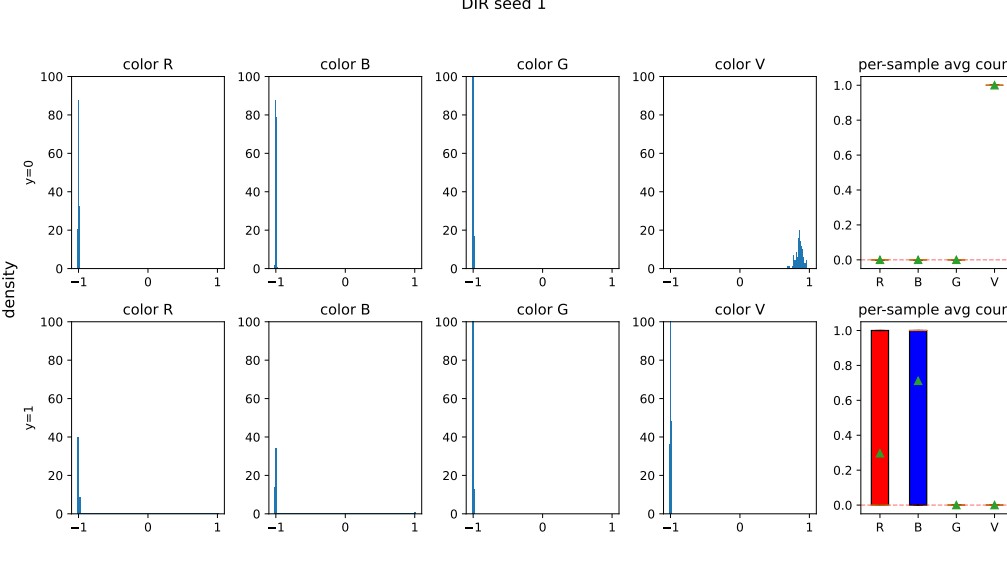

Figure 12: Diagram representing the distribution of explanation relevance score for seed 1 of `DIR` trained on `RBGV` according to the setup described in Section 6. The four pairs of plots on the left report the distribution of relevance scores, divided by class label and node color (Red, Blue, Green, and Violet), while the last pair on the right reports the boxplot (colored bars) and average count (green triangle) of each color present in the explanation of each sample. Nodes included in the explanation are chosen as those having the top-1% relevance scores. Nodes not belonging to the top-1% are assigned a score of $-1$ in the plot. Overall, even with natural training, `DIR` picks the violet node as the explanation for $y = 0$ – similarly to the malicious explanations defined in Section 4 – whereas for $y = 1$ the model is roughly selecting a single blue node. When computing rejection ratios for faithfulness metrics, we respectively get a rejection ratio of $47\%$ for EST, $0\%$ for Fid-, and $22\%$ for RFid-. Those values highlight a catastrophic failure of Fid-, which is reflected by the high standard deviation in Table 4.

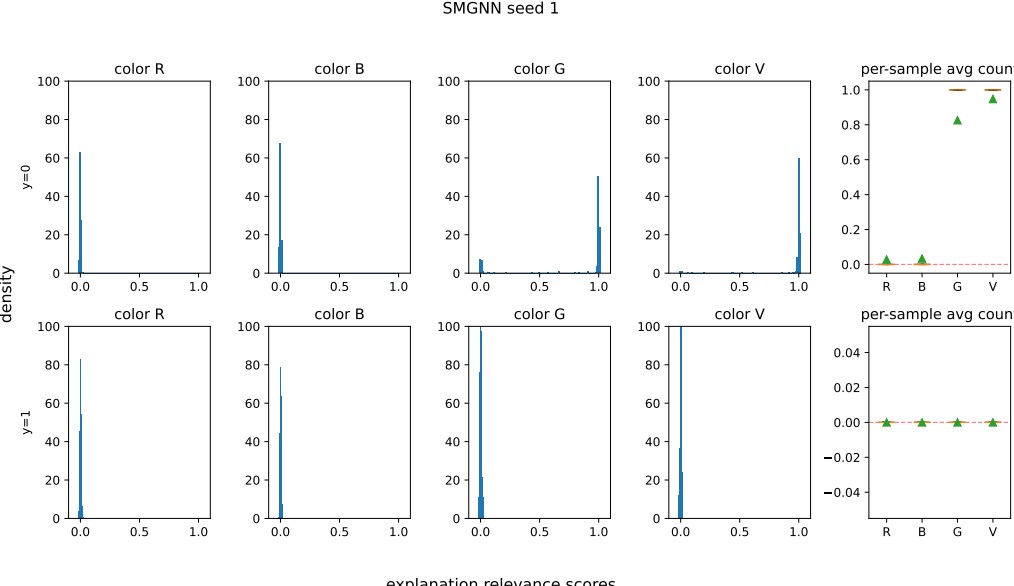

Figure 13: Diagram representing the distribution of explanation relevance score for seed 1 of SMGNN trained on RBGV according to the setup described in Section 6. The four pairs of plots on the left report the distribution of relevance scores, divided by class label and node color (Red, Blue, Green, and Violet), while the last pair on the right reports the boxplot (colored bars) and average count (green triangle) of each color present in the explanation of each sample. Nodes included in the explanation are chosen as those with a score greater than $0.5$. Overall, even with natural training, SMGNN picks green and violet nodes as the explanation for $y = 0$ – similarly to the malicious explanations defined in Section 4 – whereas for $y = 1$ the model is assigning zero relevance to every node. We then compute rejection ratios only for the negative class, getting a value of $99\%$ for EST, $0\%$ for Fid-, and $98\%$ for RFid-. Those values highlight a catastrophic failure of Fid-, which is reflected by the high standard deviation in Table 4.

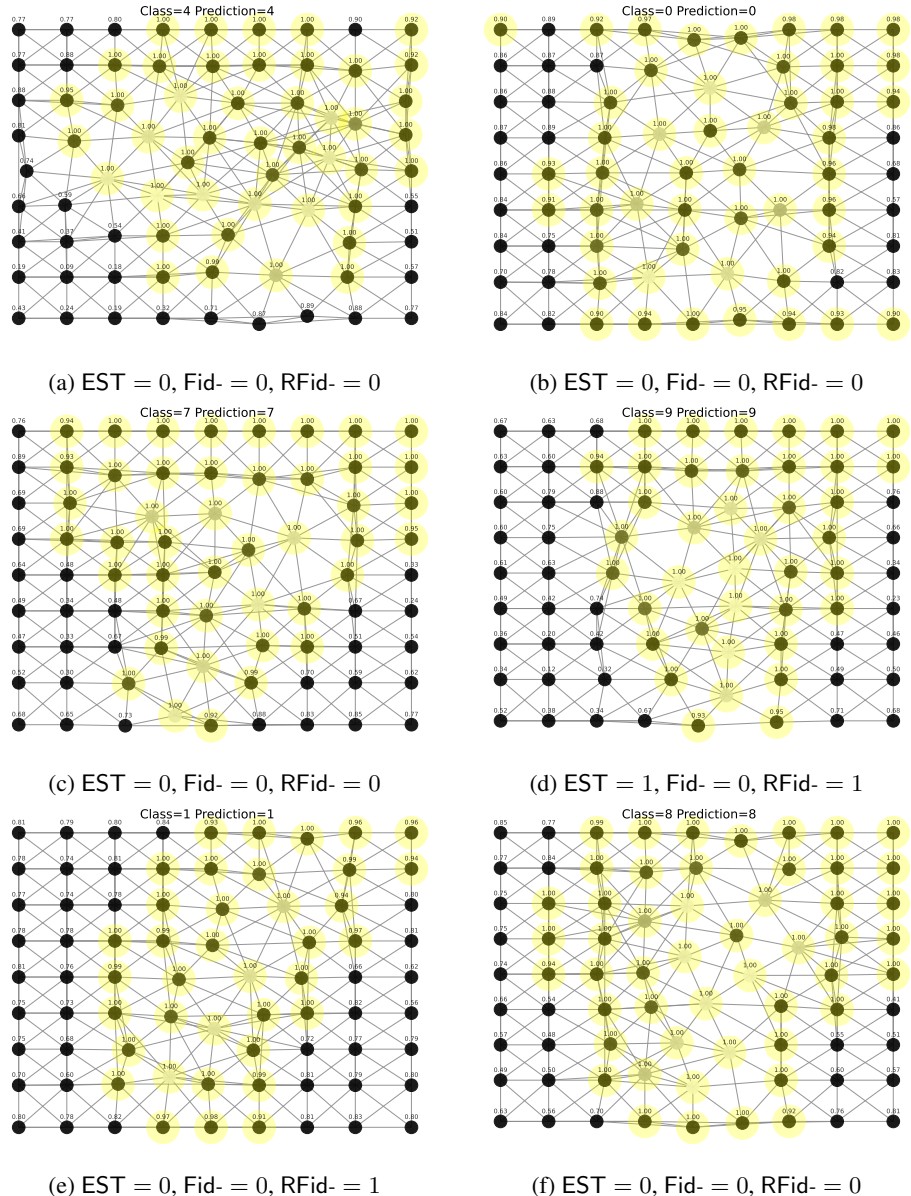

Figure 14: Examples of explanations for seed 4 of GSAT trained on MNISTsp according to the setup described in Section 6. Nodes included in the explanation are chosen as those with a score greater than 0.9. Overall, GSAT extracts explanations highlighting the whole digit with the addition of some background pixels. Therefore, we expect explanations to contain most of the information the model can use to make correct predictions. In fact, the rejection ratios reported below each example confirm that, in most cases, little to no relevant information is left in the complement. In particular, we report the value 1 whenever each metric rejects the explanation, and 0 when not.

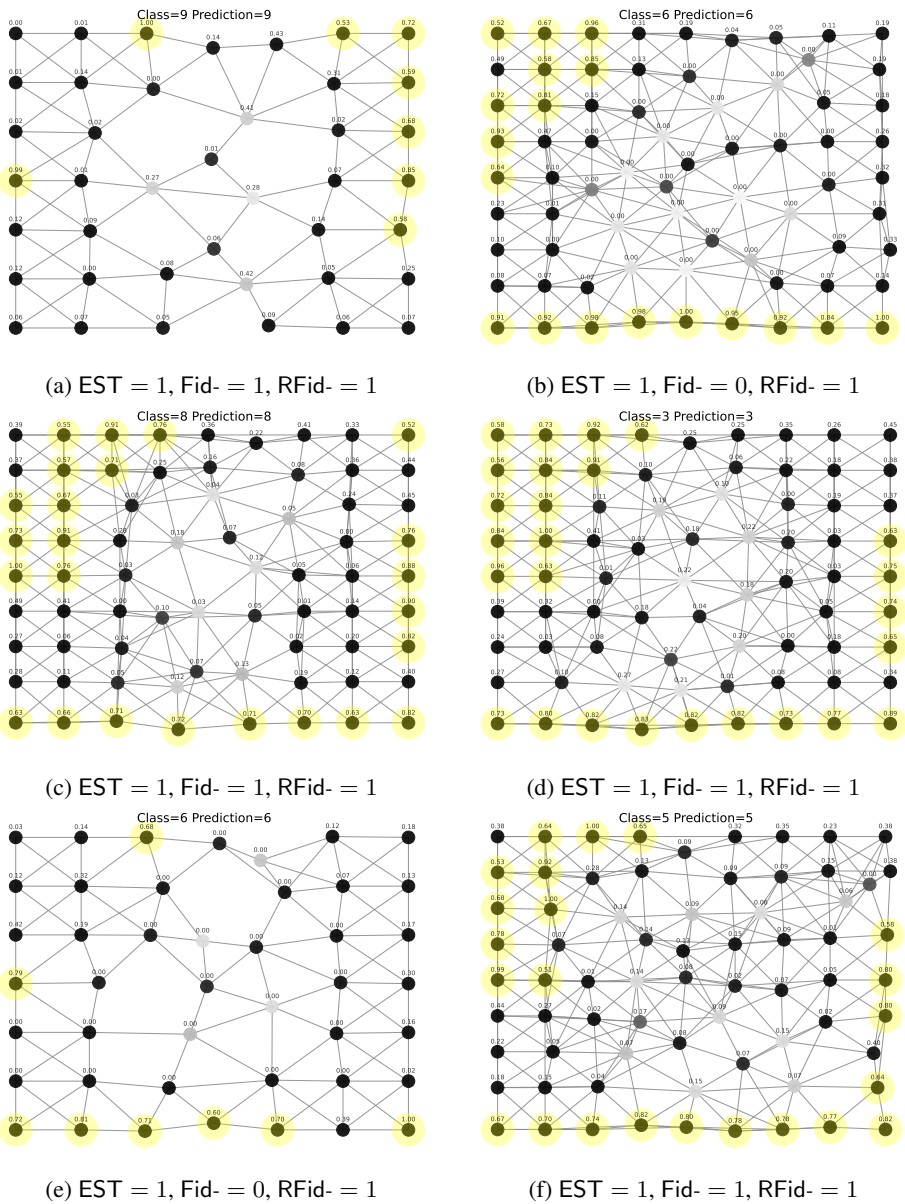

Figure 15: Examples of explanations for seed 2 of `SMGNN` trained on `MNISTsp` according to the setup described in Section 6. Node relevance scores are min-max normalized in the range $[0, 1]$ to ensure a meaningful thresholding across seeds, and the nodes included in the explanation are chosen as those with a score greater than $0.5$. Overall, `SMGNN` extracts explanations highlighting background pixels, hinting to the fact that the model could be extracting Degenerate explanations. In fact, the rejection ratios computed for each example confirm that explanations are marked as unfaithful in most cases. The only exception is Fid-, which marks them as faithful in $2$ cases. Recall that a rejection ratio of $1$ in the plots above means the explanation is rejected, hence considered unfaithful by the metric.

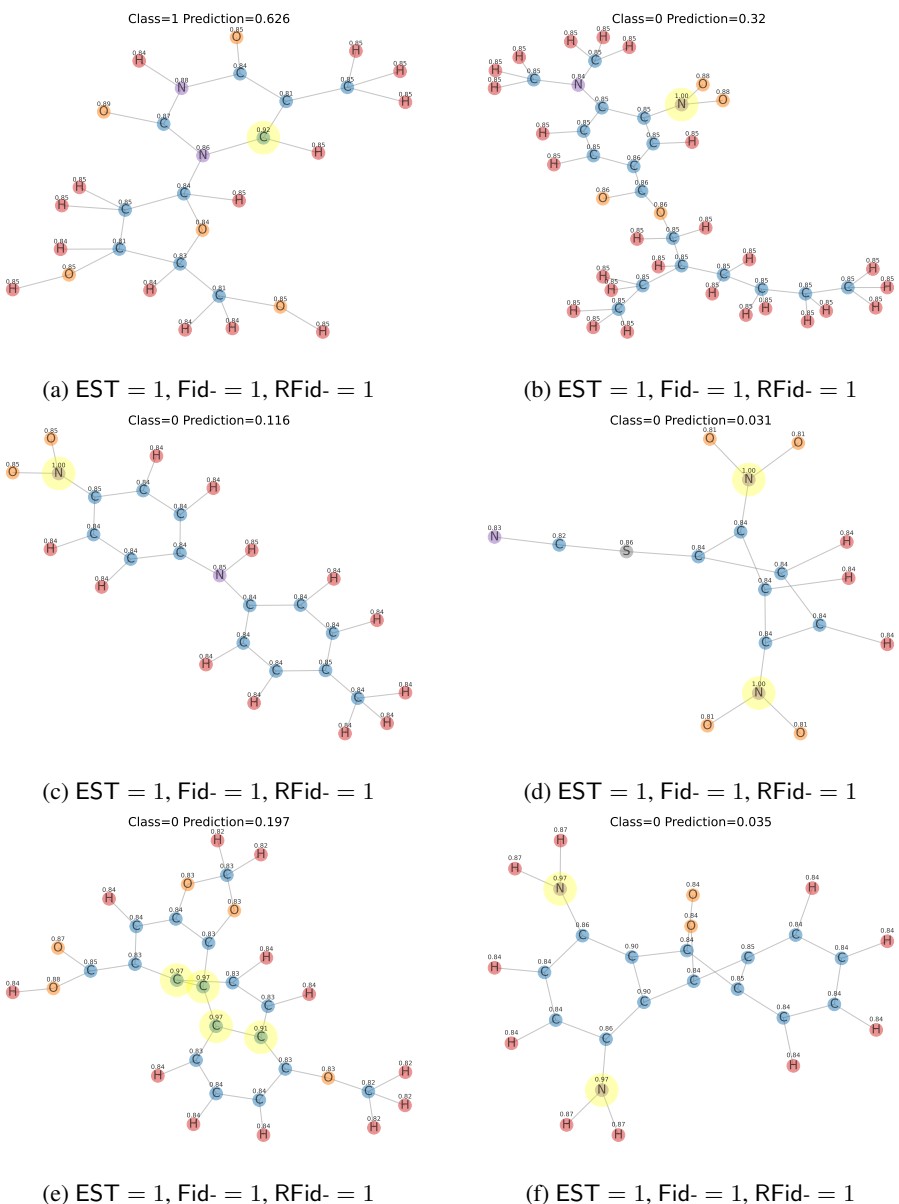

(a) EST = 1, Fid- = 1, RFid- = 1        (b) EST = 1, Fid- = 1, RFid- = 1

(c) EST = 1, Fid- = 1, RFid- = 1        (d) EST = 1, Fid- = 1, RFid- = 1

(e) EST = 1, Fid- = 1, RFid- = 1        (f) EST = 1, Fid- = 1, RFid- = 1

Figure 16: Examples of explanations for seed 1 of GSAT trained on MUTAG according to the setup described in Section 6. Nodes included in the explanation are chosen as those with a score greater than 0.9. Overall, although it is hard to judge whether explanations are faithful or not by visual inspection in this case – as we do not have full knowledge about the task and scores are not maximally sparse – we can still make the following observations: First, explanations tend to highlight atoms that are known to be uninformative of the target label when taken alone, like C and N (cf. Fig. 10); Second, rejection ratios for all metrics are non-negligible (see Table 4), outlining a substantial agreement in judging these explanations as unfaithful. Recall that a rejection ratio of 1 in the plots above means the explanation is rejected, hence considered unfaithful by the metric.

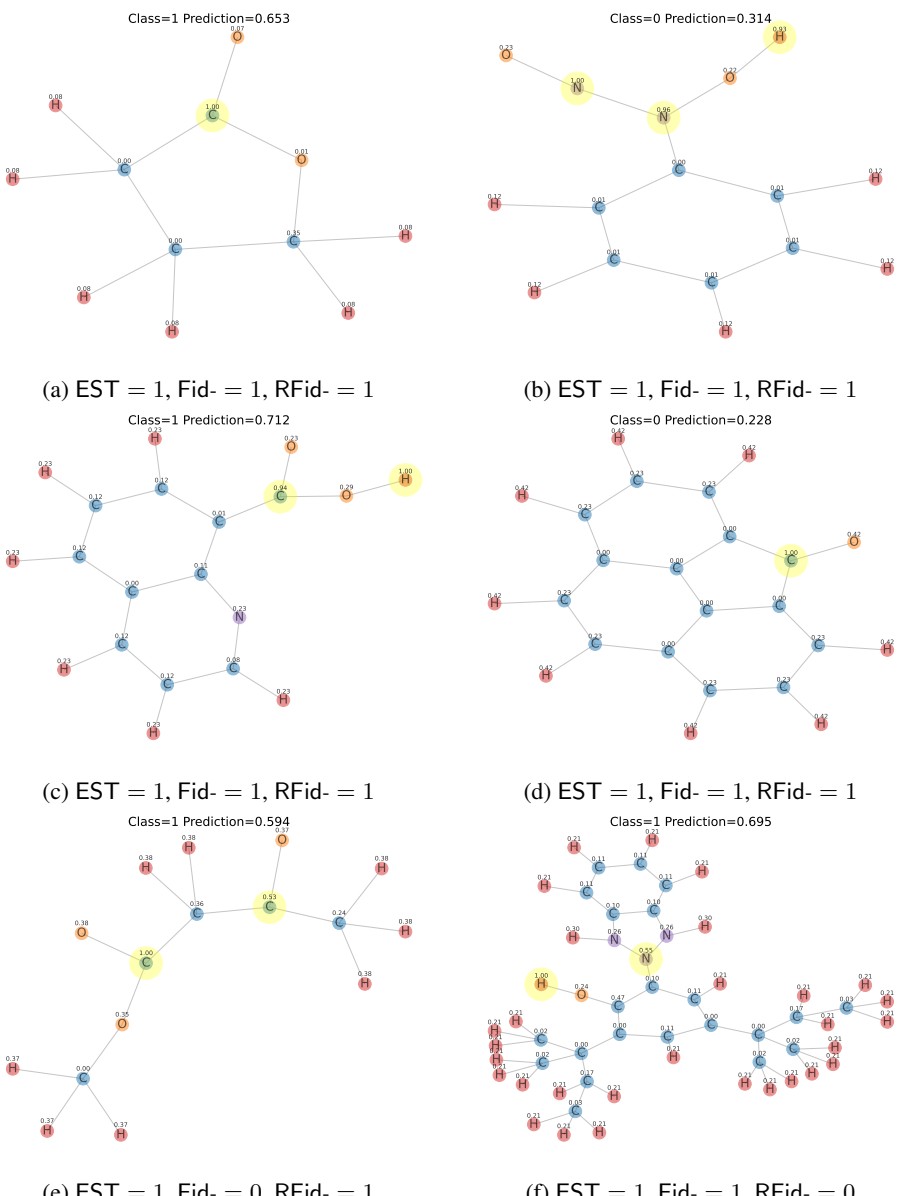

Figure 17: Examples of explanations for seed 3 of SMGNN trained on MUTAG according to the setup described in Section 6. Node relevance scores are min-max normalized in the range $[0, 1]$ to ensure a meaningful thresholding across seeds, and the nodes included in the explanation are chosen as those with a score greater than $0.5$. Overall, although it is hard to judge whether explanations are faithful or not by visual inspection in this case – as we do not have full knowledge about the task and scores are not maximally sparse – we can still make the following observations: First, explanations tend to highlight atoms that are known to be uninformative of the target label when taken alone, like C and H (cf. Fig. 10); Second, the same explanation can appear for different classes, like (a) and (d); Third, rejection ratios for all metrics are substantially high (see Table 4), outlining agreement in judging these explanations as unfaithful. Recall that a rejection ratio of 1 in the plots above means the explanation is rejected, hence considered unfaithful by the metric.

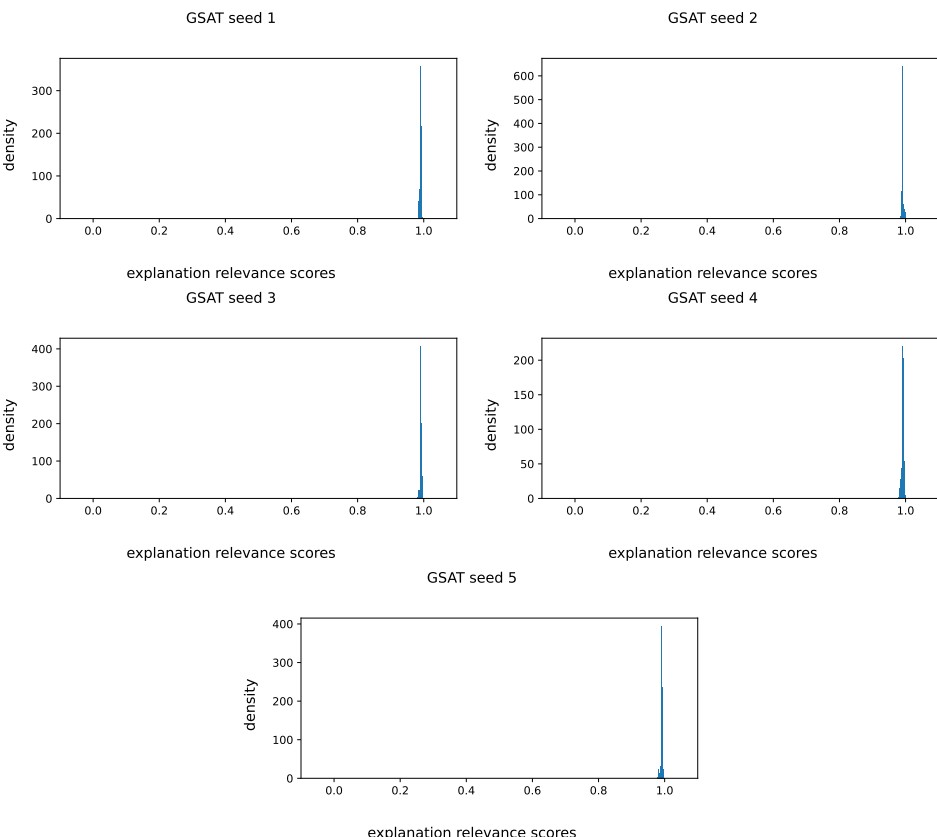

Figure 18: Histograms of explanations scores for different random seeds of GSAT trained on SST2P according to the setup described in Section 6. Overall, relevance scores are all squashed around 1, meaning the model weights equally all nodes. Therefore, by thresholding explanations at 0.9 results in explanations covering the full graph, which makes the explanation trivially *sufficient*. Hence, metrics achieve a rejection ratio of 0%.

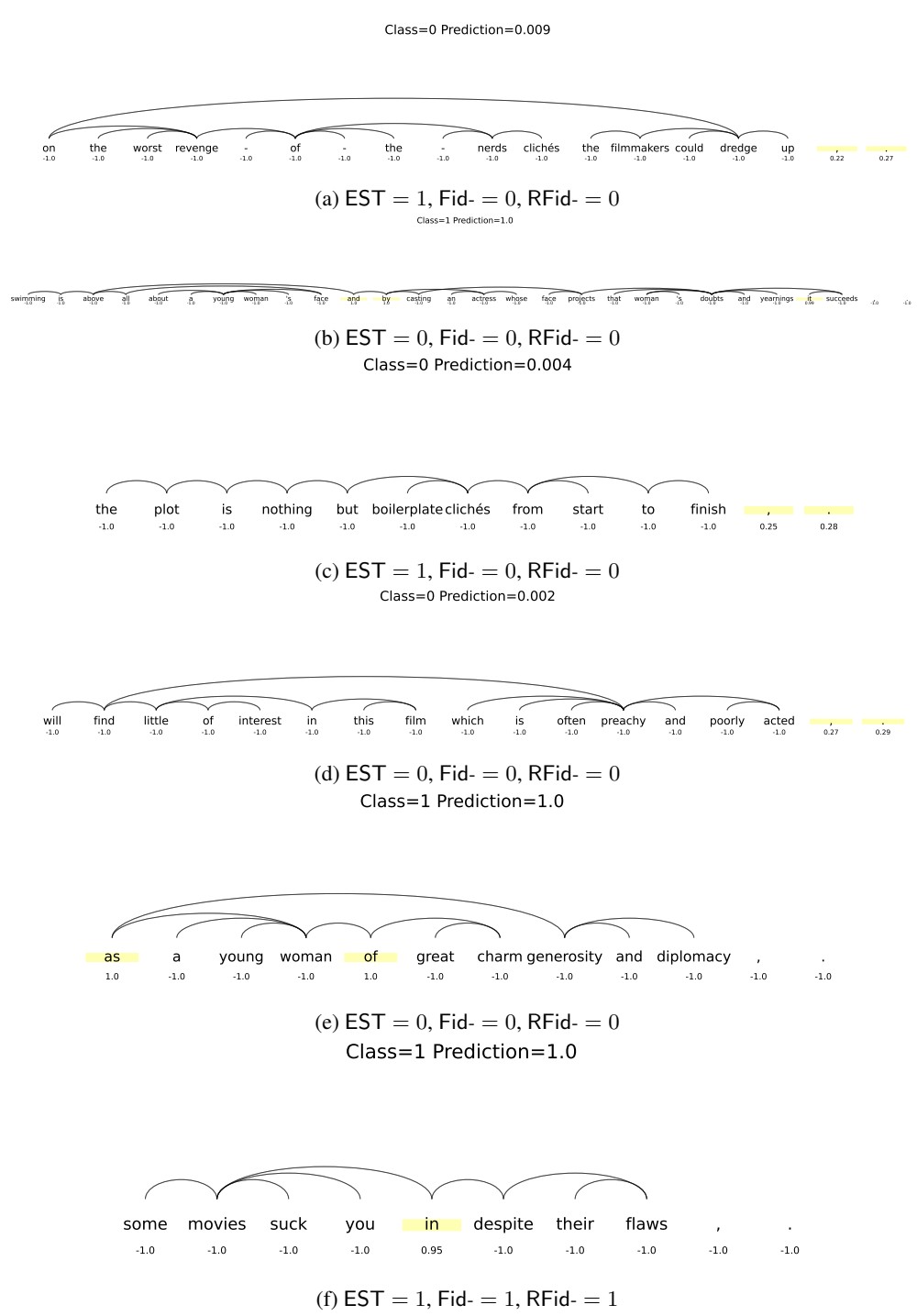

Figure 19: Examples of explanations for seed 4 of `DIR` trained on `SST2P` according to the setup described in Section 6. Nodes included in the explanation are chosen as those with the top $10\%$ relevance scores. Nodes left out of the explanation are given the default score of $-1$. Overall, for $y = 0$ the explanations consistently highlight "," and ".", similarly to the malicious explanations of Section 4. For $y = 1$, instead, explanations often highlight stop words. In addition, as shown by the rejection ratios reported below each example, EST, Fid-, and RFid- can disagree on which explanation to reject. In particular, EST correctly rejects explanations highlighting "," and "." in two cases out of three, whereas the other metrics do not reject any of them. Recall that a rejection ratio of 1 in the plots above means the explanation is rejected, hence considered unfaithful by the metric.

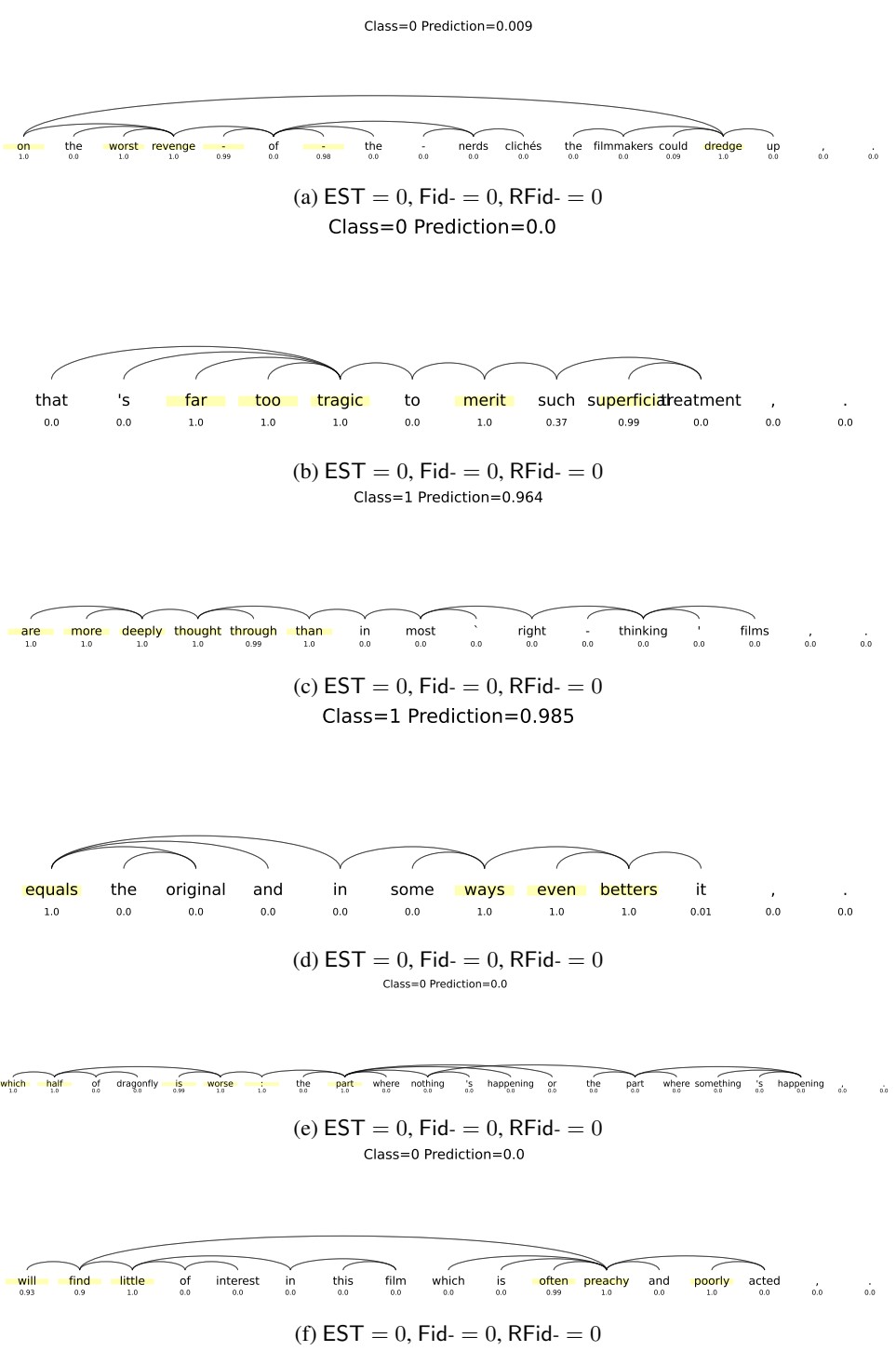

Figure 20: Examples of explanations for seed 1 of `SMGNN` trained on `SST2P` according to the setup described in Section 6. Nodes included in the explanation are chosen as those with a score greater than 0.5. Overall, all explanations highlight words with high emotional content, hinting at the polarity of the sentence. Whilst this is not an indicator of faithful explanations, it is reasonable to at least expect that the model could have used this information to infer the final prediction. Then, the fact that all the tested metrics agree that those explanations are not to be rejected hints that they can be considered as faithful. Recall that a rejection ratio of 1 in the plots above means the explanation is rejected, hence considered unfaithful by the metric.

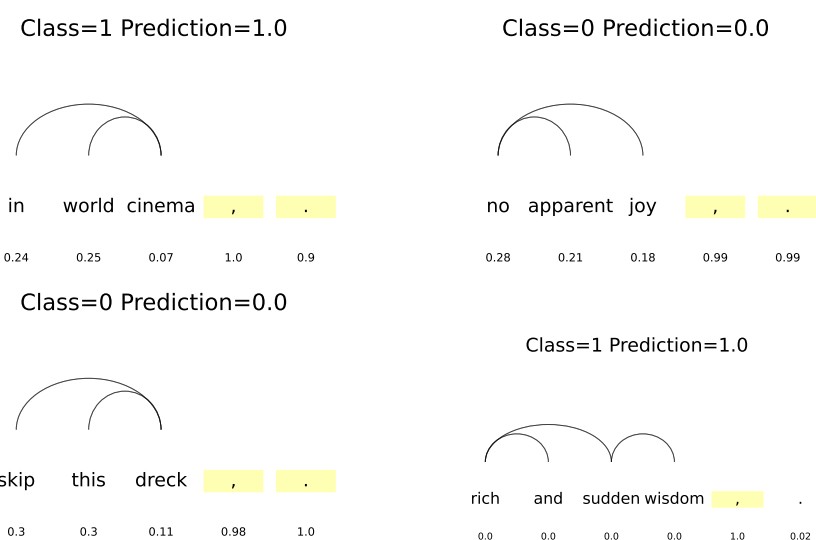

Figure 21: Explanation examples provided by the attacked `SMGNN` on the test set of `SST2P`. As described in Section 4, graphs present in the test split of `SST2P` are sensibly smaller than those of the train and validation splits, and are therefore OOD (Wu et al., 2022). As shown by the relevance scores below each token, the attack does not always generalize to OOD: in three out of four cases, several nodes outside the designated explanation (c.f. Table 5) receive above-zero relevance, which penalizes the $F_1$ score of Table 2. Nonetheless, "," and "." consistently receive near-maximum scores, while the true tokens used by the model are sensibly assigned lower relevance, meaning the attack still succeeded in suppressing the relevance of truly important tokens.

