# OpenReview forum: "GNN Explanations that do not Explain and How to find Them"
_ICLR.cc/2026/Conference — ICLR 2026 Poster_

### Official Review · Reviewer_BGhT · 2025-10-26

**Soundness:** 3
**Presentation:** 3
**Contribution:** 3
**Rating:** 6
**Confidence:** 3

**Summary:**

While GNNs have received much research attentions, explanations on GNNs are becoming important. There are in general two families of studies on GNN explainations: (1) post-hoc explanations, and (2) self-explainable GNNs. This paper focuses on the latter one, and the authors made three important contributions: (1) they indicate that even self-explainable GNNs cannot be trusted. They show both therectially and empirically that the explanations (in the form of nodes, edges, attributes, or subgraphs) given by such self-explainable GNNs could be different from the real elements that lead to the prediction. In other words, this could be a misalignment between the explanations and predictions; (2) they show that such misalignments could be manipulated by attackers, and they can also arise naturally; (3) they show that existing faithfulness metrics often fail to capture such misalignments, and they propose a new metric to mitigate such issues. Overall, I think this paper is in good quality, ready for publication.

**Strengths:**

(1) The probelm is well defined;

(2) The paper is well organised;

(3) Most statements/claims are supported empirically with experiments;

(4) This paper provides strong insights on limitations of exitsing self-explainable GNNs;

**Weaknesses:**

(1) The theoretical analysis focuses on using isolation nodes as explanations, which may not be practical; as far as I know, most GNN explataions provide subgraphs (rather than isolated nodes) as explanations;

(2) This paper focuses on class classification, and node classification is not discussed;

**Questions:**

No questions, please see the week points.

---

> ### Author Response · Authors · 2025-11-18
> **Answer to Reviewer (1)**
>
> We thank the reviewer for the positive feedback. Below, we address the two weaknesses:
>
> **W1:** We agree that most SE-GNNs provide subgraphs as explanations, and our theory already takes care of this. In fact, we’re not restricting, nor encouraging, SE-GNNs to output single-node isolated explanations, but we’re simply characterizing which kind of explanations they will output at optimality. By analyzing the loss of practical (subgraph-based) SE-GNNs, it turns out they may *naturally* prefer these degenerate explanations, even if undesirable for humans.
>
> Regardless, our experiments show that the formalized problem can emerge even when considering larger subgraph explanations (see Figs. 3, 7, 9, 15, 16, 17, 19).
>
> **W2:** Our results apply to node classification verbatim. In fact, the main insight here is that $e()$ can encode the final label inside any (small) subgraph, which is also applicable to node classification settings.
>
> As an example, let us consider a node-classification variant of our dataset *RBGV*, where each node has to be classified based on whether it has more red or blue neighbors attached to it. As in *RBGV*, let us assume each node is also connected to a single green and a single violet node. Then, SE-GNNs can still encode the overall count of colored neighbors inside green and violet nodes, similarly to Example 1.
>
> We included this remark in the *Conclusion* of the revised PDF, highlighted in red.
>
>
> ---
>
> Please let us know if you have any remaining concerns.

---

> ### Comment · Reviewer_BGhT · 2025-11-22
>
> Dear Authors,
>
> Thank you for your replies, which have partially addressed my concerns (I hope you can discuss the limitations of theoretical analysis more in depth, which I do not expect you to largely improve them in this version). Overall, I will keep my positive rating unchanged so far.
>
> PS: I have briefly checked the comments of Reviewer 65X. Regarding RQ1, which is related to "[A] Li et al. Graph Neural Network Explanations are Fragile. ICML’24"), I would like to indicate that Li's work proposes gray-box adversarial attacks on post-hoc GNN explanations, but your work focuses on self-explanations. In my experiences, post-hoc explanations are easier to manipulate (that is why the explanation can be substantially changed while the prediction remains unchanged).  In your case, you show that the explanations given by self-explainable GNNs could be different from the real elements that lead to the prediction.  As a result, for self explanations, there shoud be some chances that the attacker can substantially change the explanations but keep the prediction unchanged. I would like to see more discussions on this point (your reply to Reviewer 65X regarding this point is too short and did not catch the key point.) I am willing to raise my rating if this point is well discussed.

---

> > ### Author Response · Authors · 2025-11-25
> > **Answer to Reviewer (2.1) -- Theoretical Limitations**
> >
> > We thank the reviewer for engaging in a discussion with us, which we sincerely appreciate.
> >
> > ## Discussion on limitations of our theoretical analsysis
> >
> > > I hope you can discuss the limitations of theoretical analysis more in depth
> >
> > The two main limiting points of our theoretical analysis are:
> >
> > - **Hard explanation extractor assumption**: In our theoretical analysis, we considered explanation extractors giving attention scores saturated to either $\\{0,1\\}$ or $\\{r,1\\}$, based on the model. This is a natural desideratum for extracting explanations, as it represents an extractor maximally confident of what is relevant ($1$) and what is irrelevant ($0$ or $r$, based on the model). This desideratum is already included in the design of practical SE-GNNs, which employ TopK masking, Gumbel-softmax trick, or entropy regularization, to push the relevance scores to extremal values. Because Thm 1 analyzes the explanations produced at optimality, studying hard extractors is a natural condition for studying SE-GNNs at optimality. *We added this discussion in Limitations*.
> >
> > - **Single-node subgraphs present in all graphs**: The goal of our theoretical analysis is to formally show that SE-GNNs can output explanations unambiguously unrelated to how they infer the prediction. To achieve this, we needed a setting where it is safe to deem an explanation as *unambiguously unrelated to how the model infers the prediction*. We found this case to be the anchor set definition in line 105: when taken in isolation, the nodes of the anchor set have no class-discriminative power, as they appear in every sample. Hence, if SE-GNNs pick such nodes as explanations while achieving high accuracy, they must be looking at other parts of the input that are not revealed by the explanation. In conclusion, albeit limiting, the anchor set definition was the minimal yet representative scenario suitable for our goal. Even if this result is provided only for *single-node subgraphs present in all graphs*, we believe that it reveals a broader insight: *even if SE-GNNs are designated to be more interpretable, they do not guarantee faithful explanations*.
> >
> > Nonetheless, we agree with the reviewer that the ‘single-node subgraphs present in all graphs’ assumption may be limiting, and we decided to expand on this in the new Appendix B.2 of the revised PDF. In particular, we formalized the intuition provided in lines 154-160 to **extend the definition of anchor sets to the case of subgraphs and relaxed the condition that nodes must be present in all graphs**, as detailed below.
> >
> > First, we defined subgraph anchor sets (Definition 2), which extend node-wise anchor sets as follows: *(i)* Def. 2 considers subgraphs instead of single nodes. *(ii)*, it relaxes the condition that anchor nodes must be present in all graphs to the case where different subgraphs can appear in different graphs (see *Per-label coverage* in Def. 2).
> >
> > Second, we extended Thm 1 to show that many popular SE-GNNs can achieve a lower risk by picking explanations from the subgraph anchor set rather than more meaningful *ground truth explanations* (Theorem 2). Crucially, *this holds even when the subgraph anchor set contains only subgraphs with no class-discriminative power*, showing how the degenerate explanations discussed in Sec. 3 can also emerge in the presence of subgraph explanations.
> >
> > Please find all the details, together with a formal discussion of subgraph anchor sets and the proof of the new theorem, in Appx B.2 of the revised PDF.
> >
> > **Summary of changes:**
> > - We extended the Limitations section with a discussion on the hard explanation extractor assumption
> > - We generalized the anchors set definition to the case of subgraphs in Appx B.2 (Def. 2), and showed that SE-GNN may still prefer uncorrelated subgraphs over ground truth explanations (Thm 2).
> >
> > We thank the reviewer for the pertinent point. Please let us know if we should further elaborate on these aspects.

---

> > > ### Author Response · Authors · 2025-11-25
> > > **Answer to Reviewer (2.2) -- SE-GNNs and Fragile Explanations**
> > >
> > > ## SE-GNNs and Fragile Explanations
> > >
> > > > I would like to see more discussions on [A] Li et al. 2024
> > >
> > > To favor readability, our original answer to Reviewer 65XR was kept as concise as possible, but we are happy to discuss it further.
> > >
> > > In short, **we agree with you that self-explanations can also be fragile. Nonetheless, we believe that brittleness and unfaithful degenerate explanations are orthogonal issues in SE-GNNs**. To illustrate this, we provide two examples of SE-GNNs that output unfaithful, degenerate explanations. One of them is also susceptible to the explanation brittleness phenomenon described in [A], whereas the other is not.
> > >
> > > **Degenerate and fragile SE-GNN:**  Let us consider a toy binary graph-classification task in which each graph is composed of randomly attached nodes, either uncolored, red, or blue. A graph is labeled positive if the number of blue nodes exceeds the number of red nodes. Each instance additionally contains some fixed uncolored motifs (4-star, triangle, 6-clique): positive graphs contain the 6-clique and at least one between a 4-star and a triangle, whereas negative graphs contain the 6-clique and may or may not contain any other motifs.
> > >
> > > We now consider the following subgraph anchor set (see Def. 2 above):
> > >
> > > $$\\mathcal{Z}’= \\{ \\{ \\bar{G}_0^{0}=\\text{clique} \\}^0, \\{ \\bar{G}_0^{1}=\\text{star}, \\bar{G}_1^{1}=\\text{triangle} \\}^1 \\}$$
> > >
> > > Note that $\mathcal{Z}'$ contains only motifs with no label-discriminative power by construction. Therefore, any accurate SE-GNN that outputs explanations restricted to $\mathcal{Z}'$ produces unfaithful, degenerate explanations. Inspired by Theorem 2, we provide such an SE-GNN below:
> > >
> > > $$e(G) = argmin_{\\bar{G}_i^{y} \\in G} |\\bar{G}_i^{y}| \quad \\text{and} \quad g(\\bar{G}_i^{y}) = y$$
> > >
> > > where $y$ is the graph label. Consider now a positive graph $G$ that contains the clique and the star, so that the explanation extracted by the model highlights the star. Applying the attack in [A], we aim to modify the explanation while keeping the predicted label unchanged. A simple input manipulation consists of adding a single edge between any two leaves of the star, thereby creating a triangle.  Because the triangle is a smaller motif, the explanation extractor will now prefer selecting it, thus changing the explanation without affecting the prediction. As an observation, note that the same result can be obtained by any perturbation introducing a triangle anywhere in the graph.
> > >
> > > In conclusion, adding a single edge to the input results in a drastic change in the predicted explanation, all without any change in the predicted label.
> > >
> > >
> > > **Degenerate and non-fragile SE-GNN:** Example 1 in our paper is already an example of a robust, degenerate SE-GNN. This is because the SE-GNN cannot change the predicted explanation unless the relative numerosity of red and blue is altered, which also results in a change of the graph label. However, this violates the attack constraint number 4 in Sec. 4 of [A], making the attack unfeasible.
> > >
> > > **Conclusions:** This analysis highlights two main insights:
> > >
> > > - As outlined by the reviewer, SE-GNN can also be affected by fragile explanations, and it is possible to manipulate their explanations via the attack proposed in [A].
> > >
> > > - The problem of unfaithful degenerate explanations of SE-GNNs *is not a consequence* of this brittleness, as SE-GNNs can also output degenerate explanations that cannot be manipulated by the attack proposed in [A].
> > >
> > > We included this additional discussion, with the two explicit examples, in Appx. C.
> > >
> > > ---
> > >
> > > We sincerely thank the reviewer for engaging in a discussion with us, which contributed to the improvement of our manuscript. We are happy to continue the discussion in case the reviewer has remaining concerns.

---

### Official Review · Reviewer_65XR · 2025-10-30

**Soundness:** 3
**Presentation:** 3
**Contribution:** 2
**Rating:** 4
**Confidence:** 4

**Summary:**

This paper identifies a critical failure mode in Self-Explainable Graph Neural Networks (SE-GNNs): they can produce degenerate explanations that are completely unrelated to how the model actually makes predictions, while still achieving optimal predictive performance. The authors theoretically prove that under mild conditions, SE-GNNs can output such unfaithful explanations (e.g., highlighting non-discriminative "anchor set" nodes) and still achieve optimal true risk. Empirically demonstrate that these explanations can be maliciously planted to hide the use of sensitive attributes and existing faithfulness metrics often fail to detect such unfaithful explanations.  The paper then propose a new faithfulness metric, EST (Extension Sufficiency Test), which more reliably flags degenerate explanations by testing all possible supergraphs of the explanation.

**Strengths:**

1. Identifies a fundamental failure mode with serious implications for trustworthy AI in graph domains. The insight that explanations can serve as label-encoding channels is important for SE-GNN practitioners.

2. Theorem 1 formalizes when optimal risk coincides with degenerate expla- nations for SE-GNNs. The proof technique (constructing explicit e and g pairs) is interesting. Extension to Theorem 2 connecting EST with formal explanation notions strengthens the contribution.

3. Empirical validation on multiple datasets (synthetic RBGV, real-world MUTAG/MNISTsp/SST2P) and multiple architectures spanning different SE-GNN families in both adversarial (Section 4) and natural (Section 6) settings.

**Weaknesses:**

1. Theorem 1 requires hard explanation extractors, excluding soft/continuous scores common in practice, and it assumes |R| > 0, which limits generality. Also, the anchor set definition (single-node subgraphs in all graphs) is restrictive (in the discussion, it mentions generalizations but no formal treatment)

2. Attack requires training access (strong assumption), though authors acknowledge this fits MLaaS scenarios. The stopping criteria (appendix D.1.2) are manually tuned per dataset which is unclear how to set these systematically. Also, tere is insufficient analysis of when/why attacks fail for Table 2.

3. Fail to cite and discuss closely relevant recent works [A-C] (see questions)

**Questions:**

1. Given that Yu et al. (2019) formally characterized degeneration in rationalization models (where explanations encode labels via trivial patterns) and proposed complement predictors as a solution, how is your Theorem 1 different beyond adapting notation to graphs? (here my question is about motivation and your novelty). Can you explicitly show what new theoretical insight SE-GNNs provide over the rationalization literature?

2. You demonstrate a serious problem (degenerate explanations) and provide a detection metric (EST), but there is no empirically validated prevention strategy. Why other existing solutions from rationalization (complement predictors and shared encoders) or neuro-symbolic systems (like reasoning shortcut mitigation) don’t work for SE-GNNs?

3. Your theorem requires hard explanation extractors ({0,1} or {r,1}) and single-node anchor sets, but most practical SE-GNNs use soft scores and more complex patterns. How restrictive are these assumptions?

4. On the discussions with relevant work

- RQ1: On the Brittleness of Explanations & Malicious Manipulation (RQ1)

Connection to Ref [A] (Li et al., ICML'24): The work shows explanations can be maliciously planted. Ref [A] demonstrates that GNN explanations are fragile, meaning small perturbations to the graph structure can drastically alter the explanation. This suggests that the problem of unreliable explanations is even more widespread: they can be not only deliberately fabricated but also easily and unintentionally broken.

How does the phenomenon of maliciously planted degenerate explanations (your work) relate to the general brittleness of explanations under minor structural perturbations (Ref [A])? Could the attack you propose be seen as a structured, large-scale exploitation of this inherent fragility?


- RQ2: On Causal Metrics for Faithfulness

Connection to Ref [B] (Behnam et al., ECCV'24): The work shows existing faithfulness metrics fail. Ref [B] proposes a causal-effect-based metric, arguing that it is less susceptible to spurious correlations and can more reliably capture the true reasoning of the model. This presents a directly relevant alternative to your proposed EST metric.

How does your proposed EST metric compare to causal-effect-based metrics like the one in Ref [B]? Could a causal framework provide a more principled foundation for defining and measuring faithfulness, potentially avoiding the need for the perturbation-based approach of EST? Please discuss the potential advantages and limitations of both perturbation-based and causal-based faithfulness metrics in the context of your findings.

- RQ3: On Provably Robust Explanations

Connection to Ref [C] (Li et al., ICLR'25): The work identifies a problem (degenerate explanations), and Ref [C] offers a potential solution for a related issue. Ref [C] aims to build SE-GNNs whose explanations are provably robust to graph perturbations, ensuring consistency. While their goal is robustness against adversarial graph perturbations, the underlying principle—enforcing stability in the explanation—might also prevent the model from latching onto arbitrary, degenerate subgraphs.

Could the techniques for building provably robust explanations (Ref [C]) be adapted or extended to formally prevent the emergence of degenerate explanations identified in your Theorem 1? In other words, can robustness constraints guide the model toward explanations that are not only stable but also inherently faithful?

[A] Li et al. Graph Neural Network Explanations are Fragile. ICML’24

[B] Behnam et al. Graph Neural Network Causal Explanation via Neural Causal Models. ECCV’24

[C] Li et al. PROVABLY ROBUST EXPLAINABLE GRAPH NEURAL NETWORKS AGAINST GRAPH PERTURBATION ATTACKS. ICLR’25

**I would consider raising the score if the questions are sufficiently addressed during the rebuttal**

---

> ### Author Response · Authors · 2025-11-18
> **Answer to Reviewer (1)**
>
> Thank you for your detailed review and your valuable time. Below, we address your concerns.
>
> ## Q1:
>
> > How is Theorem different beyond adapting to graphs?
>
> Although similar in spirit to the analysis of Yu et al. (2019), Thm 1 is significantly different in both technical contribution and scope of application. First, although Yu et al. (2019) provide solid intuition about degeneracy in rationalization models, they do not provide a formal analytical proof. Second, Thm 1 explicitly analyzes loss functions of commonly-used SE-GNNs, whereas Yu et al. (2019) only describe the issue for an abstract rationalization model, without a sound methodology for identifying when degenerate explanations appear in the wild. Third, Yu et al. (2019) – just like many other works in the rationalization literature – mainly focus on generators optimizing for rationale conciseness via sparsity constraints. Instead, our Thm 1 applies to a broader family of architectures, including also Information Bottleneck (e.g., GSAT) and causality-based (CAL).
>
> > What new theoretical insight SE-GNNs provide over rationalization?
>
> Some key insights are:
>
> - **Mitigations can be evaded:** We show that some commonly-used mitigation strategies from rationalization literature, like complement predictors and shared encoders, may be ineffective for SE-GNNs (see *Q2*). This warrants future research – both within SE-GNNs and rationalization – in robust strategies provably avoiding degeneracy.
>
> - **Degeneracy can go undetected:** We show that poorly designed faithfulness metrics can fail to detect degeneracy, undermining practitioners’ ability to audit and trust explanations. Devising reliable degeneracy-detection strategies is currently underexplored in the rationalization literature, and we reinforce their critical importance.
>
> - **Misalignment beyond degenerate explanations**: Our experiment in Appx. E1 shows that even when explanations are optimized to match human expectations – thus avoiding degeneracy through supervision – the model can still conceal its use of protected attributes. This reveals a fundamental misalignment between the explanations and the model’s actual decision process, beyond degeneracy alone.
>
> We incorporated this additional discussion in Appx C of the revised PDF, highlighted in red.
>
> ## Q2:
> Thanks for raising this interesting question. The SEGNNs we considered in the experiments already implement similar types of mitigation strategies:
>
> - SE-GNNs with complement predictors (DIR) require a fixed node budget $K$ for explanations. However, choosing $K$ poorly yields degeneracy because the selected nodes are forced to encode the label to maintain accuracy (see lines 410–418).
>
> - SE-GNNs with shared encoders (GSAT) use highly expressive GNNs that can still learn alternative rules for the two modules, and this allows degeneracy to persist (see Sec. 6). This aligns with Liu et al. 2022, who note that shared encoders mitigate but do not fully eliminate the issue.
>
> - Marconato et al. (2023) discuss mitigations of reasoning shortcuts, but the only *fully reliable* one requires dense concept supervision – a setting that does not apply to SE-GNNs, which are not defined over human-specified concepts.
>
> We added these observations when discussing the related work in Appendix C, highlighted in red.
>
> As a final remark, please note that our current focus is on developing a robust detection metric that enables users to assess the reliability of a given explanation. We view this as a standalone contribution, while a complete analysis and design of new prevention strategies represents a natural next step.

---

> ### Author Response · Authors · 2025-11-18
> **Answer to Reviewer (2)**
>
> ## Q3 & W1:
>
> The **hard explanation extractor assumption** is a natural and commonly found desideratum for extracting explanations (Yu et al. 2020, Azzolin et al. 2025b), as it represents an extractor maximally confident of what is relevant ($1$) and what is irrelevant ($0$ or $r$, based on the model). This desideratum is already included in the design of practical SE-GNNs, which employ TopK masking (DIR), gumbel-softmax trick (GSAT, LRI), or entropy regularization (SMGNN), to push the relevance scores to extremal values. Because Thm 1 analyzes the explanations produced at optimality, studying hard extractors is a natural condition for studying SE-GNNs at optimality.
>
> $|R| > 0$ is just a natural condition to avoid trivial, empty explanations.
>
> Regarding the **single-node subgraphs in all graphs** definition, it represents the minimal non-trivial case of degenerate explanations that is both easy to understand and of practical relevance. In fact, this definition is naturally met across many real-world scenarios, like molecular property prediction (organic molecules contain carbon, often bond to hydrogen), object recognition (images often contain pixels with the same value), and text classification (texts usually contain punctuation or recurrent stop words). This definition also precisely formalizes previous works in rationalization literature, which often provide examples of degeneracy by considering recurrent, punctuation tokens (Yu et al 2019; Chang et al 2019; Liu et al 2022). Nevertheless, our experiments already show that our insights also apply to subgraph explanation (see Fig. 7, 9, 15), even when not appearing in *all* graphs (see Fig. 16, 17, 19).
>
> ## W2:
>
> > How to set stopping criteria systematically?
>
> The stopping criteria can be tuned by the attacker to optimize the attack performance. Attackers can also exploit any validation/test split to train the model until it maximizes the attack $F_1$ score of Table 2, like normally done for standard training.
>
> > Insufficient analysis of attack failures
>
> The attack fails only in that the designated degenerate explanation is not exactly matched.
> This is because some input nodes may not be properly distinguished after $L$ layers of message passing (e.g., due to oversmoothing), or because the extractor failed to infer the right label for the graph. We clarified this in lines 1582-1585 of the revised PDF. Nonetheless, Appx. E.3 shows that the model still highlights unrelated subgraphs while concealing the true features it relies on, which is actually the attack's original aim (lines 241-246).
>
> To additionally clarify the low $F_1$ score of SMGNN on SST2P (Table 2), we added the novel Fig. 21 in the revised PDF to provide new examples. These show that explanations highlight “.” and “,” for both classes, misaligning with the designated explanation but still concealing truly relevant tokens. As discussed in lines 222-227, this is because graphs are OOD and our attack –  just like regular GNNs – may struggle to generalize to OOD.
>
> ## RQ1:
> [A] aims to perturb the input graph to keep the prediction intact while changing the explanation. However, *degenerate explanations may not be easily broken*. For instance, in Example 1, the provided SE-GNN cannot change the provided explanation unless the class of the graph changes as well, which contradicts the aim of the attack in [A].
>
> ## RQ2:
> [B] proposes a causality-based post-hoc explainer rather than a faithfulness metric. In fact, in the experimental section of [B] (Sec. 5), only plausibility-like metrics are taken into account, and no mention of *faithfulness* appears throughout the paper.
>
> ## RQ3:
>  [C] considers two possible attacking scenarios:
>
> - Change the explanation but keep the prediction: this is similar to [A]; see *RQ1*
>
> - Change the prediction: in this case, *robustness can indeed play a role for faithfulness*. In fact, EST (Def. 1) can be seen as measuring the robustness of the model to changes outside of the explanation, and devising an SE-GNN that is robust to such perturbations will naturally lead to non-degenerate explanations. Nonetheless, the technique in [C] is different: instead of training an SE-GNN to be robust, they build a wrapper around a trained GNN to make it more robust a posteriori. Verifying whether [C] can be adapted to improve faithfulness is an interesting future analysis.
>
> We included the discussion on these related works in Appx. C.
>
> ---
>
> Please let us know if our answers addressed your concerns; we would be happy to further clarify.

---

> ### Author Response · Authors · 2025-11-25
> **Answer to Reviewer (3) -- Extended Discussion**
>
> Dear reviewer,
>
> Thank you again for your time in reviewing our work.
>
> In relation to your comments in W1 and RQ1, and following the discussion with Rev. BGhT, we have updated the PDF to include the following points:
>
> - **Degeneracy for subgraph anchor sets**: We defined subgraph anchor sets (Definition 2 in Appx. B.2), which extend node-wise anchor sets as follows: *(i)* Def. 2 considers subgraphs instead of single nodes. *(ii)* Def. 2 relaxes the condition that anchor nodes must be present in all graphs to the case where different subgraphs can appear in different graphs. We then showed in Theorem 2 that SE-GNNs can still prefer explanations picked from the subgraph anchor sets instead of ground truth explanations, even when the subgraph anchor set contains only subgraphs with no class-discriminative power. *This addresses your last comment in W1.*
>
>
> - **SE-GNNs and fragile explanations [A]**: We expanded on our original answer to your *RQ1* by providing two explicit examples of SE-GNNs outputting unfaithful, degenerate explanations, but where only one of them is fragile. This shows that the issue of degenerate explanations is not a consequence of the general brittleness of explanations. Please see *Answer to Reviewer (2.2)* in the discussion with Rev. BGhT for more details. *This addresses your question in RQ1*.
>
>
> In short, **we have further updated the PDF as follows**:
>
> - In Appendix B.2, we provide a more general version of anchor sets (Def. 2), and the new Theorem 2, which generalizes Theorem 1.
>
> - In Limitations, we improved the discussion of the hard explanation extractor assumption.
>
> - In Appendix C, we added an expanded discussion on the relation between SE-GNNs and fragile explanations.
>
>
> ---
>
> Please let us know if our answers addressed your concerns; we would be happy to further clarify.

---

### Official Review · Reviewer_UuzW · 2025-10-31

**Soundness:** 4
**Presentation:** 4
**Contribution:** 3
**Rating:** 8
**Confidence:** 4

**Summary:**

This paper raise a possible explanation for self-explainable GNN(SE-GNN)'s unfaithfulness, that the explanation extractor may extract subgraphs with bijective mapping to the real groundtruth explanation to achieve well classification performance, yet only providing no-related explanation subgraphs. The author validate this view by training a group of attacked models which are humanly controlled on specific false explanation, while achieving good classification performance. Based on such insights, the author further propose a new sample-based metric namely EST on evaluating unfaithful explanations of SE-GNNs, and validate its effectiveness through experiments on both good true-negative and near-zero false-negative.

**Strengths:**

**1.** This paper is well presented, which makes me enjoy reading it. The flow of questions raised, illustration of the problem, proper designed figures and complete experiments make the motivation and findings of the paper well expressed.

**2.** The raised explanation for SE-GNN's unfaithfulness in Theorem 1 is interesting and convincing. It assumes that some common anchor nodes serves as a bridge between real pattern and classification attention is novel, which also makes sense to me. The example 1 on the with figure illustrated is also well designed, which is simple and obvious for understanding.

**3.** The experiments are complete and sound, which support mostly all claims in the paper. To validate the existence of such problem the author provide (1) the attacked models' performance (Table 2), (2) attacked models' rejection rate (Table 3) and (3) natural models' rejection rate (Table 4), which show that such unfaithfulness could exist in normal performing mode and may be potential reason for unfaithfulness of natural model.  To validate the effectiveness of the proposed EST, both the well rejection ratio of attacked models (Table 3) and the near zero rejection rate for designated explanations (Table 10) are provided to show this EST is a reliable metric. The provided Figure 2 also aligns with the supposed sample amount requirement of EST and also shows its effectiveness. It's also good to see every experiment column in Table 4 is assigned with a illustration figure.

**Weaknesses:**

**1.** The design of the new metric is somewhat limited in novelty. The difference between the proposed metric and other metrics is mainly just  enlarging the choice range of samples, i.e., including both nodes and edges for randomness. This metric is designed to approximate all the subgraphs, however, just assumed as the sample amount increases they are equivalent, which may increase high complexity. While this may not be a critic point since evaluation complexity is not a so serious problem, and explanation subgraphs are usually small.

**A minor typo**: Fig.2 UST may suppose to be EST

**Questions:**

1. Would this raised faithful problem also happens in post-hoc explainer on raw GNN? I think this shift between original important patterns and patterns used for classification may also appears in raw GNN models, e.g., for a star graph it only recognizes the most high degree node.  And could this metric be also applicable for them?

---

> ### Author Response · Authors · 2025-11-18
> **Answer to Reviewer (1)**
>
> We thank the reviewer for the very positive feedback and for the detailed review. Below, we address the remaining concerns:
>
> **W1:**
>
> > The design of the new metric is somewhat limited in novelty
>
> We agree that EST is an extension of previous metrics. Nonetheless, we show that *our improved design is crucial* for avoiding the critical failure cases of previous metrics presented in Section 5, and in making EST a robust tool to audit explanations (Table 3).
>
> > EST is designed to approximate all the subgraphs, which may increase its complexity
>
> It is true that in the worst case, the complexity of fully enumerating all supergraphs is high. This is in line with the previous findings that provable auditing is intrinsically computationally expensive (Bhattacharjee and von Luxburg 2024). Nonetheless, our metric is designed to be more reliable than previous ones even when limiting the number of perturbations to a small budget, which makes the auditing feasible in many real-world cases (see Tables 3, 4, and Fig. 2).
>
>
> **Q1:**
>
> > Would this raised faithful problem also happen in a post-hoc explainer on a raw GNN?
>
> Good point, we agree that extending the analysis to post-hoc settings is a promising direction for future research. However, doing so is non-trivial. Degenerate explanations in SE-GNNs often arise because the classifier $g()$ *collapses* to a simple mapping between label-encoding explanations and class labels. In a post-hoc setting, by contrast, the raw GNN is trained directly on full graphs and is therefore *encouraged* to learn more meaningful structural patterns to achieve good accuracy. For example, a raw GNN trained on our *RBGV* dataset must genuinely learn to count red and blue nodes (while likely ignoring green and violet) to perform well. Nonetheless, prior work for tabular data and images has demonstrated that it is possible to maliciously attack post-hoc explainers to induce them to output arbitrary unfaithful explanations regardless of how good the underlying classifier is [1,2], but a similar analysis for the graph domain remains underexplored.
>
> *We believe that a systematic comparative analysis between SE-GNN and post-hoc explainer’s tendency to unfaithfulness is a valuable contribution for future research.*
>
>
> > I think this shift between original important patterns and patterns used for classification may also appear in raw GNN models
>
> Previous work in post-hoc explainers has already analyzed the shift between human-defined important patterns and those actually used for classification [3,4], indicating that this shift is indeed present even for simple datasets. However, these studies focus on the misalignment between human expectations and model behavior, whereas we focus on the misalignment between model behavior and its explanations. This difference is substantial, and to the best of our knowledge, the latter remains underexplored.
>
> > Could this metric also be applicable to them?
>
> Yes, it can. EST determines whether altering the explanation’s complement affects the model’s predictions, all without relying on any assumptions about the model itself. This makes the EST metric model-agnostic.
>
>
> [1] Fooling LIME and SHAP: Adversarial Attacks on Post hoc Explanation Methods. AIES20
>
> [2] Explanations can be manipulated and geometry is to blame. NeurIPS19
>
> [3] Faber et al. When Comparing to Ground Truth is Wrong: On Evaluating GNN Explanation Methods. KDD21
>
> [4] Fontanesi et al. Xai and bias of deep graph networks. ESANN24
>
>
> **Typo:** Fixed! Thanks for spotting this.

---

> > ### Comment · Reviewer_UuzW · 2025-11-19
> >
> > Thanks for your reply, which addressed my remaining concerns. I'll keep my recommendation for acceptance.

---

### Author Response · Authors · 2025-12-01
**Global Summary**

Dear AC,

In this comment, we provide a summary of the main points raised by the reviewers, briefly summarizing how we addressed them. Besides the modifications listed below, the main results, experiments, and the core discussion remain unchanged, as all reviewers agreed that our submission was already well organised (Rev. BGhT & UuzW), provided convincing evidence (Rev. UuzW), and tackles serious implications for trustworthy AI (Rev. 65XR) with strong insights (Rev. BGhT).

**The hard explanation extractor assumption limits the generality of theoretical results** (Rev. 65XR W1 & Q3):
We expanded the discussion in the *Limitations* section of the revised PDF, clarifying that this is a natural desideratum for many SE-GNNs  – as it represents an explanation extractor that is maximally confident of what is relevant and what is irrelevant –  and showing how popular SE-GNNs already promote hard explanation extractors by design. The generality of our theoretical insights is confirmed by the experiments in Section 6, showing that practical SE-GNNs using soft scores can, in fact, extract degenerate explanations.


**The anchor set definition limits the generality of theoretical results** (Rev. 65XR W1  & Q3, Rev. BGhT W1):
We agree with the reviewers that Anchor sets (i.e., single nodes present in all graphs) are a restrictive scenario (as we had anticipated in the Limitations section). However, this does not limit the broader impact of our theoretical results: we aim to show that SE-GNNs can highlight explanations that are totally unrelated to how SE-GNNs infer labels, and anchor sets capture a minimal yet rigorous scenario in which this failure case systematically occurs (see our last answer to Rev. BGhT). The generality of our results is not restricted to anchor sets, and after Theorem 1, we briefly discussed generalizations to more complex scenarios. Following the discussions with the reviewers, we have fully formalized this generalization in the new Appendix B.2 (see Def. 2 and Thm. 2).


**Insufficient analysis of attack failures for Table 2** (Rev. 65XR W2):
We expanded Appendix E.3 "*Failures of SE-GNN Attacks*" with a more detailed discussion of why and when the attack may not yield the desired result. Nonetheless, we note both in the main text (lines 241-246) and in our response to Rev. 65XR that, although the attack did not highlight the desired degenerate explanation, it still managed to conceal the features truly used by the model. To further expand on this, we also provided more visual examples of this case in the new Fig. 21.

Rev. UuzW and Rev. 65XR further asked to discuss some additional related work, which we discussed in their respective answers and in the *Extended Related Work* in Appx. C. Changes in the revised PDF are highlighted in red.

---

### Meta-Review · Area_Chair_99tF · 2026-01-03

**Summary:**

This submission identifies a failure mode in self-explainable graph neural networks, showing that models can achieve optimal predictive performance while producing degenerate, unfaithful explanations, and proposes both a formal theoretical characterization and a robust detection metric (EST). The work is technically sound, empirically well supported across multiple datasets and SE-GNN architectures, and addresses a timely problem in trustworthy graph learning. While one reviewer initially raised concerns regarding novelty relative to rationalization literature, restrictiveness of assumptions, lack of prevention strategies, and positioning with respect to recent related work, these issues were substantively addressed in a clear and comprehensive rebuttal. The rebuttal convincingly clarifies the theoretical scope, justifies the assumptions as reflective of practical SE-GNN behavior, strengthens empirical explanations, and situates the contribution appropriately within the broader literature.

**Reviewer Concerns:**

The reviewer’s main concerns center on (i) limited novelty relative to prior rationalization literature (e.g., Yu et al. 2019), questioning whether Theorem 1 offers fundamentally new insight beyond adapting known degeneration arguments to graphs; (ii) the absence of empirically validated prevention mechanisms for degenerate explanations, despite demonstrating the problem and proposing a detection metric; (iii) the restrictiveness of the theoretical assumptions, particularly hard explanation extractors and single-node anchor sets; one other reviewer also shared this concern (iv) practical weaknesses of the attack setup, including strong training-access assumptions, manually tuned stopping criteria, and limited analysis of attack failures; and (v) insufficient discussion and positioning with respect to closely related recent works on explanation fragility, causal faithfulness metrics, and provably robust explanations.

In my opinion, the rebuttal succeeds in reducing ambiguity around several points, but its effectiveness varies. It is strongest in defending the theoretical assumptions, offering a persuasive argument that hard extractors and minimal anchor sets naturally arise at optimality in practical SE-GNNs and that the phenomena extend beyond these simplified cases empirically. The novelty argument is partially convincing: while the rebuttal clearly differentiates the work through a formal SE-GNN–specific treatment and broader architectural scope, it relies more on positioning and scope than on articulating a sharply new conceptual insight induced by graph structure. On prevention strategies, the rebuttal is adequate but conservative, explaining why existing mitigations fail rather than providing deeper empirical or theoretical evidence; this likely satisfies the reviewer’s expectations but does not significantly strengthen the paper. The responses to experimental and attack-related concerns are acceptable but somewhat defensive, clarifying limitations rather than reframing them as insights. Overall, the rebuttal is comprehensive enough to justify modestly increasing the score.

**Reviewer Scores:**

Two reviewers provided positive assessments, assigning scores of 8 and 6. Both engaged in the discussion and expressed support for the submission. It appears they maintain their original ratings.

A third reviewer initially rated the paper a 4 but indicated a willingness to raise the score if the rebuttal adequately addressed their concerns. Given the strength and quality of the rebuttal (as discussed above), it is reasonable to expect this score to increase to 6. This adjustment would result in all three reviewers offering positive evaluations of the work.

Accordingly, I recommend acceptance.

---

### Decision · Program_Chairs · 2026-01-26

Accept (Poster)